# Future changes in the mean and variability of extreme rainfall indices over the Guinea Coast and role of the Atlantic equatorial mode

Koffi Worou[1], Thierry Fichefet[1], and Hugues Goosse[1]

[1]Earth and Climate Research Centre (TECLIM), Earth and Life Institute (ELI), Université catholique de Louvain (UCLouvain), Louvain-la-Neuve, Belgium

**Correspondence:** Koffi Worou (koffi.worou@uclouvain.be)

**Abstract.** The occurrence of climate extremes could have dramatic impacts on various sectors such as agriculture, water supply, and energy production. This study aims to understand part of the variability in the extreme rainfall indices over Guinea Coast that can be related to the Atlantic equatorial mode (AEM) whose positive phases are associated with an increase in the intensity and frequency of rainfall events. We use six extreme indices computed from six observed rainfall databases and historical and SSP5-8.5 simulations from 24 general circulation models (GCMs) that participate in the sixth phase of the Coupled Model Intercomparison Project (CMIP6) to study changes in extreme rainfall events over Guinea Coast during July–September. Under present-day conditions, we found that current GCMs clearly overestimate the frequency of wet events and the maximum number of consecutive wet days. The magnitude of the other extreme indices simulated is within the range of the observations which, moreover, present a large spread. Our results confirm existing studies. However, less attention has been paid to the evaluation of the modelled rainfall extremes associated with the AEM under different climate conditions, while the variability of the AEM is expected to decrease in the future with a potentially significant impact on the extreme events. Here, we use six (one) observed rainfall (sea surface temperature) data and 24 GCMs outputs to investigate the present-day, near-term, mid-term, and long-term future links between the AEM and the extreme rainfall events over Guinea Coast. The biases in the extreme rainfall responses to the AEM are subject to a large spread across the different models and observations. For the long-term future (2080-2099), less frequent and more intense rainfall events are projected. As an illustration, the multimodel ensemble median (EnsMedian) maximum rainfall during five consecutive wet days (RX5day) would be $21\%$ higher than under present-day conditions. Moreover, the variability of the majority of the extreme indices over Guinea Coast is projected to increase ($48\%$ for RX5day in the long-term future). By contrast, the decreased variability of the AEM in a warmer climate leads to a reduced magnitude of the rainfall extreme responses associated with AEM over Guinea Coast. While under present-day conditions the AEM explains $18\%$ of the RX5day variance in the EnsMedian, this value is reduced to $8\%$ at the end of $21^{\text{st}}$ century. As a consequence, in absolute, there is a projected increase in the total variability of most of the extreme rainfall indices, but the contribution of the AEM to this variability weakens in a warmer future climate.

# 1 Introduction

Severe flash flood events were reported over West and Central Africa during the last decades, affecting millions of people,
destroying houses, buildings and left dozens of deaths (United Nations Office for the Coordination of Humanitarian Affairs
(OCHA), 2012, 2021). Some of these climatic hazards were caused by severely abnormal rainfall, which often contribute to
increasing the rivers levels, favouring flooding of the surrounding areas (Elagib et al., 2021; Fofana et al., 2022). A combination
of different factors could also lead to high impact weather events, such as the very heavy rainfall that occurred in Ouagadougou
on the 1st of September 2009 (Engel et al., 2017; Lafore et al., 2017; Beucher et al., 2019), where than one third of the
annual rainfall was recorded within $24\,\mathrm{hours}$, and reached $263\,\mathrm{mm}$. In addition, the large-scale sea surface temperature (SST)
conditions during this event showed an anomalous warming of the Mediterranean Sea, a negative North Atlantic Oscillation-
like pattern and a pronounced Atlantic cold tongue. This enhanced cold tongue corresponds to a negative phase of the Atlantic
equatorial mode (AEM), whose impact on the occurrence of extreme rainfall events over Guinea Coast is the focus of the
current study (Zebiak, 1993; Losada et al., 2010a, b).

During the last decades, the Guinea Coast has experienced changes in its rainfall characteristics, with conclusions that
depend on the season considered, the period of study, the source of observations and the extreme rainfall index used (Bichet
and Diedhiou, 2018; New et al., 2006; Odoulami and Akinsanola, 2017; Kpanou et al., 2020; Dike et al., 2020). It is important
to highlight that these changes are not homogeneous over the whole region. Through the period 1961-2000, the observations
indicate a decrease in the annual rainfall in southern Nigeria, with, an increase in the intensity of the daily rainfall and the
annual maximum 1 day rainfall (New et al., 2006). Odoulami and Akinsanola (2017) showed, however, a significant negative
trend in the June-September mean daily rainfall over the Guinea Coast during the 1998-2013 period. Nevertheless, they found
a decrease in the frequency of rainfall events higher than the 95th percentile. This result is opposite to Kpanou et al. (2020),
who, on an annual base, found an increasing trend in the number of days with rainfall exceeding the 95th percentile over
coastal areas in the southwestern Côte d'Ivoire, Togo, and Bénin. Odoulami and Akinsanola (2017) also observed an overall
upward and insignificant trend in the number of dry days over that area and a significant decreasing trend in the number of
wet days. The Sixth Assessment Report (AR6) of the Intergovernmental Panel on Climate Change (IPCC) stated that there is
a low confidence in the observed heavy precipitation trend over West Africa for the last decades, as well as in the contribution
of human influence to that trend (Seneviratne et al., 2021).

Several studies have shown that climate General Circulation Models (GCMs) participating in the sixth phase of the Coupled
Models Intercomparison Project (CMIP6) simulate reasonably well the spatial distribution of the rainfall extreme indices in
West Africa, with however different levels of magnitude (Faye and Akinsanola, 2021; Klutse et al., 2021). Moreover, Regional
Climate Models (RCMs) forced with CMIP5 GCMs outputs (CORDEX project) show an added value in simulating the spatial
distribution of the present-day extreme indices over West Africa (Akinsanola et al., 2020). In addition these models project
an enhancement of extreme rainfall events over Guinea Coast in the future, under the Representative Concentration Pathway
(RCP) 4.5 and 8.5 forcing scenarios (Akinsanola and Zhou, 2019). Furthermore, Akinsanola et al. (2020) demonstrated from
RCP4.5 and RCP8.5 projections with RCMs an increase in the West African rainfall variability, which is associated with

enhanced mean rainfall and rainfall extremes in the Guinea Coast. This increase in variability was explained by the change in the water vapor in a warmer world following the Clausius Clapeyron equation (Akinsanola et al., 2020).

The first mode of covariability between the sea surface temperature in the tropical Atlantic and the rainfall over West Africa during the boreal summer indicates a strong connection between the eastern equatorial Atlantic SST and the Guinea Coast rainfall, and explains 31% of the total covariability (Polo et al., 2008). These local SST changes are related to the Atlantic equatorial mode, also named Atlantic zonal mode (Zebiak, 1993), whose variability occurs on interannual timescales. Positive phases of the AEM are characterized by above normal SST conditions in the eastern equatorial Atlantic. The enhanced low-level wind convergence over the warm oceanic area is accompanied by an ascent of moist air, which is then advected by the low-level circulation toward the Guinea Coast. This provides favourable conditions for the rainfall occurrence over this region (Polo et al., 2008; Rodríguez-Fonseca et al., 2015; Schubert et al., 2016; Lübbecke et al., 2018; Worou et al., 2020). Worou et al. (2022) have demonstrated that the covariability between the Guinea Coast rainfall and the AEM is maximum in JAS, in the simulations performed within CMIP6 models (June-September in observations), under present-day conditions. Over the 1901-2016 period, in observations, the eastern equatorial Atlantic positive correlation with the rainfall over the Guinea Coast was clear during the boreal summer (Losada et al., 2012; Diatta and Fink, 2014; Worou et al., 2020). Nevertheless, the variability of the AEM is projected to decrease under future global warming (Worou et al., 2022; Crespo et al., 2022; Yang et al., 2022). This would potentially reduce the role of the AEM in the Guinea Coast rainfall variability. Moreover, Diatta et al. (2020) showed that the Guinea Coast rainfall extremes are strongly correlated with the AEM index. Atiah et al. (2020) also showed an increase (a decrease) in the total annual wet day rainfall, heavy rainfall and very heavy rainfall over Ghana during warmer (colder) sea surface conditions in the eastern equatorial Atlantic. Therefore, it is important to understand and quantify the projected potential changes in the extreme rainfall events in Guinea Coast that are associated with the AEM.

Previous studies suggested a link between the AEM and the Guinea Coast extreme rainfall variability in observations. However, this link in the current GCMs and the impact of the projected reduction in the AEM variability on the extreme rainfall events over Guinea Coast have not yet been assessed. This study aims to further explore the relationship suggested in previous studies between the AEM and the extreme rainfall events over the Guinea Coast under present-day and future climate conditions by using outputs from CMIP6 models and to determine how those changes in AEM contribute to the simulated changes in rainfall extremes. The Section 2 is dedicated to the data used in this work and the methods adopted. Section 3 describes the present-day statistics of the rainfall extremes over the Guinea Coast, and the near-term, mid-term and long-term future changes in the mean and variability of these extremes. In Section 4, the impact of the AEM on the Guinea Coast rainfall extremes under present-day conditions is evaluated, as well as the future changes in the extreme indices responses related to the AEM. We conclude with Section 5, where our findings are summarized.

## 2   Datasets and methods

The present study focuses on the impact of the Atlantic equatorial mode on the rainfall extreme events over the Guinea Coast under different climate conditions. Four 20-year periods are considered: the present-day (1995 - 2014), the near-term future

(2021 - 2040), the mid-term future (2041 - 2060) and the long-term future (2080 - 2099) periods. The choice of these periods is based on the definition provided by IPCC (2021) (see their Table SPM.1).

## 2.1 CMIP6 data

Daily rainfall and monthly SST data from 24 climate models participating in CMIP6 were retrieved from one of the Earth System Grid Federation (ESGF) portals (e.g. https://esgf-node.llnl.gov/search/cmip6/, last access: 15 June 2022) and analysed.
The analysis of the present-day period is based on the historical simulations, which covered the 1850-2014 period and were forced with observed natural and anthropogenic forcings (Eyring et al., 2016). The analyses for the future periods rely on the shared socioeconomic pathway with a high greenhouse gas emission (SSP5-8.5, O'Neill et al., 2016). These simulations are started from the year 2015. We choose the SSP5-8.5 scenario to get the clearest signal of climate change.

The majority of the studies analysing the internal modes of variability in the tropical Atlantic only used one member for
each available model (e.g., Kucharski and Joshi, 2017; Richter and Tokinaga, 2020; Worou et al., 2022; Crespo et al., 2022; Yang et al., 2022). We rely here on the same approach and use one realization for each model in our study, assuming that all realizations are equivalent for the analyses we performed. It could be interesting to test the limitations of this hypothesis using the models that provide a relatively large ensemble for both historical and future simulations. However, this is out of the scope of this present study, and we expect that the differences between ensemble members of a specific model are smaller than
the differences between models for the diagnostics analysed here. Table 1 provides a list of the different models used, their corresponding ensemble member, and their resolution.

## 2.2 Observations

To evaluate the performance of the models in simulating different aspects of the extreme indices, and to take into account uncertainties in observations, we use daily rainfall data from six different sources: CHIRPS, ARCv2, PERSIANN-CCS-CDR,
REGEN LongTermStns, TAMSAT v3, and GPCC-FDD-v2022. These data are described in Table 2. We show in Fig. S1 three additional observed rainfall data: GSMAP, IMERG and GPCP (Kubota et al., 2007; Huffman et al., 2015; Becker et al., 2013). These data are not considered in our current work, as they do not cover the present-day period (1995-2014). They show, however, a consistent annual cycle of the extreme rainfall indices with the other observational rainfall data (Fig. 1). A comparison between some of those datasets is also given in Sanogo et al. (2022); Ageet et al. (2022).
Observed monthly SST data are derived from the Hadley Centre Global Sea Ice and Sea Surface Temperature (HadISST). This dataset is available at a spatial resolution of 1° (Rayner et al., 2003) and covers the 1870-2022 period. We do not include additional observed SST data, as we consider that this is not a critical point for our analyses.

## 2.3 Choice of the season

This study is focused on the July-September season (JAS), following Worou et al. (2022). This season contributes to $46\%$ of the
120 total annual rainfall over the Guinea Coast. It is also dominated by the monsoon system of West Africa, which is characterized

**Table 1.** List of the analysed 24 CMIP6 models in this study, their ensemble member considered in the historical and SSP5-8.5 simulations, and their resolutions. In the variant label for each model member, "r", "i", "p" and "f" represent the realization index, the initialization method, the physics, and the forcing, respectively. Note that in the SSP5-8.5 outputs, the parent variant label does not necessarily correspond to the variant label. We then read thoroughly metadata in future simulation outputs and associate them to their corresponding parents, from which they were branched. For example, the daily precipitation file from the SSP5-8.5 simulation r2i1p1f1 performed with CESM2 was branched on the historical r11i1p1f1 output.

| CMIP6 model | Historical member | SSP5-8.5 member | Resolution ($°$lat$\times°$lon) |
| --- | --- | --- | --- |
| ACCESS-CM2 | r1i1p1f1 | r1i1p1f1 | 1.25 x 1.88 |
| ACCESS-ESM1-5 | r1i1p1f1 | r1i1p1f1 | 1.25 x 1.88 |
| CESM2 | r11i1p1f1 | r2i1p1f1 | 0.94 x 1.25 |
| CESM2-WACCM | r1i1p1f1 | r1i1p1f1 | 0.94 x 1.25 |
| CNRM-CM6-1 | r1i1p1f2 | r1i1p1f2 | 1.40 x 1.41 |
| CNRM-CM6-1-HR | r1i1p1f2 | r1i1p1f2 | 0.50 x 0.50 |
| CNRM-ESM2-1 | r1i1p1f2 | r1i1p1f2 | 1.40 x 1.41 |
| CanESM5 | r1i1p1f1 | r1i1p1f1 | 2.79 x 2.81 |
| EC-Earth3 | r1i1p1f1 | r1i1p1f1 | 0.70 x 0.70 |
| EC-Earth3-Veg | r1i1p1f1 | r1i1p1f1 | 0.70 x 0.70 |
| GFDL-ESM4 | r1i1p1f1 | r1i1p1f1 | 1.00 x 1.25 |
| HadGEM3-GC31-LL | r1i1p1f3 | r1i1p1f3 | 1.25 x 1.88 |
| INM-CM4-8 | r1i1p1f1 | r1i1p1f1 | 1.50 x 2.00 |
| INM-CM5-0 | r1i1p1f1 | r1i1p1f1 | 1.50 x 2.00 |
| IPSL-CM6A-LR | r1i1p1f1 | r1i1p1f1 | 1.27 x 2.50 |
| KACE-1-0-G | r1i1p1f1 | r2i1p1f1 | 1.25 x 1.88 |
| MIROC-ES2L | r1i1p1f2 | r1i1p1f2 | 2.79 x 2.81 |
| MIROC6 | r1i1p1f1 | r1i1p1f1 | 1.40 x 1.41 |
| MPI-ESM1-2-HR | r1i1p1f1 | r1i1p1f1 | 0.94 x 0.94 |
| MPI-ESM1-2-LR | r1i1p1f1 | r1i1p1f1 | 1.87 x 1.88 |
| MRI-ESM2-0 | r1i1p1f1 | r1i1p1f1 | 1.12 x 1.13 |
| NorESM2-LM | r1i1p1f1 | r1i1p1f1 | 1.89 x 2.50 |
| NorESM2-MM | r1i1p1f1 | r1i1p1f1 | 0.94 x 1.25 |
| UKESM1-0-LL | r1i1p1f2 | r1i1p1f2 | 1.25 x 1.88 |

**Table 2.** List of the six observed rainfall data considered in this study.

| Data | Title | Availability | Resolution (°lat×°lon) | Reference |
|------|-------|--------------|------------------------|-----------|
| CHIRPS | Climate Hazards Group Infrared Precipitation with Station Data | 1981-2023 | 0.25 x 0.25 | Funk et al. (2014) |
| ARCv2 | African Rainfall Climatology Version 2 | 1983-2023 | 0.1 x 0.1 | Novella and Thiaw (2013) |
| PERSIANN-CCS-CDR | Precipitation Estimation from Remotely Sensed Information using Artificial Neural Networks-Cloud Classification System-Climate Data Record | 1983 - 2023 | 0.04 x 0.04 | Sadeghi et al. (2021) |
| REGEN LongTermStns | Rainfall Estimates on a Gridded Network based on long-term station data v1-2019 | 1950-2016 | 1 x 1 | Contractor et al. (2020) |
| TAMSAT v3 | Tropical Applications of Meteorology using SATellite data and ground-based observations, version 3.0 | 1983-2023 | 0.0375 x 0.0375 | Maidment et al. (2017) |
| GPCC-FDD-v2022 | Global Precipitation Climatology Centre, Full Data Daily Version 2022 | 1982-2020 | 1 x 1 | Markus et al. (2022) |

by an abrupt shift of the rainfall band from the coastal areas in the end of June to the Sahel region. Sanogo et al. (2022) found a strong coupling between the West African monsoon rainfall and the occurrence of extreme precipitation events, mainly during the boreal summer. Note that although mesoscale convective systems have an important contribution to the Guinea Coast rainfall during April-May-June and September-October-November (Maranan et al., 2018), the annual cycle of the very wet day index (Table 3) across different observations indicates the highest values during July, August, September and October (Fig. S1 (b)). The total wet day precipitation (PRCPTOT) and the contribution of the monthly rainfall to the annual rainfall (ANNPCT) also show their highest values over these months (Fig. S1 (a),(c)). As we do not focus our analysis on specific contributions of mesoscale convective systems to the rainfall over Guinea Coast, we will keep the JAS season as the focus of our study.

## 2.4 Methods

### 2.4.1 Definition of the AEM index

Before computing the AEM index, all the monthly observations and models SST data are remapped on the HadISST grid, at $1°$ of resolution, by using a bilinear interpolation method with the climate data operator routine (CDO, https://code.mpimet.mpg.de/projects/cdo). The index of the AEM is defined over the Atlantic Niño 3 region (ATL3, Zebiak, 1993) which extends between $20°W$ - $0°E$ and $3°S$ - $3°N$. There are models with a low resolution that only have a few grid points within the ATL3 region. If the AEM index was computed on each model grid, this would imply to use different regions for different models. This motivates the choice of a common grid of $1°$ of resolution, which requires interpolating the data. A similar procedure has been applied in Kucharski and Joshi (2017); Worou et al. (2022) for instance.

Over every 20-year study period, the monthly SST data is linearly detrended at each grid point to remove any drift due to climate change. The resulting monthly SST anomalies are then averaged over the ATL3 region. Next, we compute the seasonal (JAS) mean of the obtained area-averaged monthly SST anomalies. This gives one value per year for the AEM index. The AEM index time series are then standardized, by dividing them by their standard deviation over the entire period considered.

### 2.4.2 Computation of the extreme rainfall indices

First, all the daily observational and model rainfall data are remapped on the grid of the model which has the lowest resolution. In our study, this corresponds to the grid of MIROC-ES2L model, with a resolution of $2.8°$. We use a first order conservative remapping method with the CDO routine to perform the remapping of the rainfall data. We define the Guinea Coast region as the area extending between $18°W$ - $15°E$, and $4°S$ - $10°N$. There are 32 grid points inside this area on the MIROC-ES2L grid (Fig. S2).

In this work, we analyse a set of six extreme rainfall indices defined by the Expert Team on Climate Change Detection and Indices (ETCCDI) (http://etccdi.pacificclimate.org/indices_def.shtml, last access: 16 June 2022):

– the SDII (simple daily intensity index), which describes the intensity of wet rainfall events;

– the R95p (very wet day), which describes the intensity of rainfall events exceeding the 95th percentile;

– the CWD (consecutive wet days), which describes the maximum duration of a wet event;

– the RX5day (maximum 5 days precipitation), which is the intensity of an event over a duration of five days;

– the FRQW (frequency of wet days), which describes the frequency of wet events;

– R20mm (very heavy precipitation days), which is a measure of the frequency of rainfall events exceeding $20\,mm$, and which could have a high socioeconomic impact.

The details of these indices are provided in Table 3. The indices recommended by the ETCCDI are widely used to monitor and detect changes in drought and wet conditions over different regions. Some applications can be found in several studies such as New et al. (2006); Sillmann et al. (2013a, b); Mouhamed et al. (2013); Diedhiou et al. (2018); Faye and Akinsanola (2021); Delhaye et al. (2022), among others. We computed these indices in two ways. Firstly, we computed monthly indices over the whole period of study, and perform the monthly averages to get the mean annual cycle. In a second way, we considered for each year in the period of study, daily rainfall values over the JAS season (92 values at each grid point). We obtain one value per year for each extreme index. The corresponding series provide the information needed to study the seasonal climatology and interannual variability of the extreme rainfall indices.

### 2.4.3 Links between the AEM and the extreme rainfall indices

The analysis of the JAS rainfall extreme patterns associated with the JAS AEM is completed through linear regressions of the extreme anomalies at each grid point onto the standardized JAS AEM SST index. The patterns are computed for each GCM. The regression patterns in the observations are computed by regressing the rainfall indices from the six different observational data onto the standardized AEM index from HADISST. In both cases, we will consider the ensemble median (EnsMedian) of the models and the EnsMedian of the observations to resume the common characteristics. To identify the most robust changes among the members of those observations and models' EnsMedians, we will only consider a two-third sign-agreement approach

**Table 3.** List and definition of the six rainfall extreme indices selected for this study. These indices are based on the definition provided by the ETCCDI.

| Index Label | Index Name | Index Definition | Index Unit |
|---|---|---|---|
| SDII | Simple daily intensity index | For each year, compute the average of wet days daily rainfall in a month/season. A wet day is defined as a day when the rainfall is greater or equal to $1\,mm$ | $mm \cdot day^{-1}$ |
| R20mm | Very heavy precipitation days | For each year, count the number of days in the month/season when the daily rainfall is greater or equal to $20\,mm$ | $days$ |
| R95p | Very wet days | Let PR95 be the 95 percentile of the wet-day daily rainfall timeseries in the month/season over the 1995-2014 period. For each year, sum of rainfall over days in the month/season, when the rainfall amount is greater or equal to PR95 | $mm$ |
| RX5day | Maximum 5 days precipitation | For each year, compute the maximum of the rainfall sum over 5 consecutive days in the month/season. | $mm$ |
| CWD | Consecutive wet days | For each year, compute the largest number of consecutive wet days in the month/season. A wet day is defined as a day when the rainfall is greater or equal to $1\,mm$ | $days$ |
| FRQW | Frequency of wet days | For each year, compute the number of wet days (when the rainfall is greater or equal to $1\,mm$) in the month/season. The result is divided by the total number of days in the month/season, and multiplied by 100 | $\%$ |

among the set of data to show the robust signals related to AEM: at a grid point, a regression coefficient value is robust if more than 66% of the models agree on the sign of the multimodel EnsMedian (Rehfeld et al., 2020). Similarly, the robustness of the models' EnsMedian change in the regression patterns between two periods will be based on the two-third sign agreement approach: at a grid point, the change in the regression slope between two periods is robust if at least two-third of the models agree on the sign of the EnsMedian.

### 2.4.4   Robustness of the mean state changes

We use the methodology suggested in the Cross-Chapter Box Atlas 1 of the IPCC AR6 (Gutiérrez et al., 2021; Dosio et al., 2021) to disentangle the forced changes signal from the internal variability. We will mainly indicate regions where the forced response (the EnsMedian change in the climatology) is robust, or conflicting or non-robust. A change in the mean state is considered as robust if two thirds of the models show changes greater than the interannual variability (IAV) and 80% of the models agree on the sign of the change. The IAV is computed for each model following the equation:

$$IAV = \sqrt{\frac{2}{20}} \times 1.645 \times \sigma \tag{1}$$

where $\sigma$ is the standard deviation of the linearly detrended variable for the present-day period (1995-2014). The factor $\sqrt{2/20}$ takes into account the variability of the difference between two 20-year periods. The factor 1.645 considers a confidence interval of 90% for the change signal to exceed the IAV. A change is considered non-robust if less than two-third of the models

present a change greater than the IAV. Finally, a change signal is conflicting if more than two-third of the models project a change greater than the IAV, and less than 80% agree on the sign of the change.

Additionally, we test another approach used in Monerie et al. (2017) (but we keep the IPCC approach), to determine the
similarities in the robustness of the mean state changes. In this approach, a signal-to-noise ratio (SNR) is defined as:

$$SNR = \frac{\overline{\Delta X}}{\sigma \Delta X} \tag{2}$$

where:

  – $\Delta X$ is the change in a variable $X$ in one model,

  – $\overline{\Delta X}$ is the multimodel ensemble mean (EnsMean) of the change in a variable $X$, which is termed the forced signal.

  – $\sigma \Delta X$ is the spread of the changes (the standard deviation) among the different models, which is due to the internal variability and the different responses of the GCMs.

Therefore, regions where the SNR is greater than one are supposed to experience a robust change. Qualitatively, both approaches give a similar result for the robust long-term changes in the extreme indices, except for R95p (Fig. S4). In the IPCC approach, the projected long-term changes in R95p are non-robust, whereas the second approach shows robust changes over a
large area of West Africa. Moreover, the IPCC approach gives more information on grid points where the change is not robust or conflicting, whereas the second approach only provides information on the robust forced changes. Hereafter, we consider the IPCC approach.

### 2.4.5  Performance metrics for the models

The evaluation of the models' performance in representing the spatial distribution of the different extreme indices relative to
observations is based on four metrics which have been applied in different studies (e.g., Akinsanola and Zhou, 2019; Faye and Akinsanola, 2021; Akinsanola et al., 2021; Li et al., 2021) :

  – the percentage of bias (%BIAS)

$$\%BIAS = 100 \times \frac{\sum_{i=1}^{n}(M_i - O_i)}{\sum_{i=1}^{n} O_i} \tag{3}$$

  – the normalized root mean square error (NRMSE)

$$NRMSE = \frac{\sqrt{\frac{1}{n}\sum_{i=1}^{n}(M_i - O_i)^2}}{\frac{1}{n}\sum_{i=1}^{n} O_i} \tag{4}$$

  – the pattern correlation coefficient (PCC)

$$PCC(M,O) = \frac{Cov(M,O)}{\sqrt{Var(M)Var(O)}} \tag{5}$$

- a variant of the Taylor skill score (Taylor, 2001)

$$TSS = \frac{4(1+PCC)^2}{\left(\frac{\sigma_{cmip6}}{\sigma_{observation}} + \frac{\sigma_{observation}}{\sigma_{cmip6}}\right)^2 (1+PCC_0)^2} \tag{6}$$

where $M$ and $O$ are model and observation values, respectively, $i$ is the index of a grid point, $n$ is the number of grid points over which the data are compared, $Cov$ and $Var$ are the covariance and variance, respectively, $PCC_0$ is the maximum correlation reachable, set to 1 in our case, and $\sigma_{cmip6}$ and $\sigma_{observation}$ are the standard deviations of the model and observation patterns (mean state pattern or teleconnection pattern) over the Guinea Coast, respectively. TSS and PCC (%BIAS and NRMSE) values are close to one (zero) for a very good representation of the observations by the model. The TSS is close to zero for no match between the model and observation. Moreover, one can choose to penalize models with low spatial correlation or low spatial variability by modifying the power of the different terms in the TSS (e.g., equations 4 and 5 in Taylor, 2001). In our case, both the variability and correlation terms are to the same power. We do not choose to reward models that better represent the spatial pattern correlation than the spatial variability and inversely.

## 3 Present and future rainfall extremes over the Guinea Coast

### 3.1 Annual cycle of extreme rainfall indices over Guinea Coast

The mean annual cycle of the Guinea Coast extreme rainfall indices is shown in Figure 1. In the GCMs, the magnitudes of those indices increase in general from January, reach a maximum during the boreal summer and decrease afterward. In the observations, a similar behaviour is present in the indices SDII, R20mm, R95p and RX5day (Fig. 1 (a)-(d)). It is not the case for the observed CWD and FRQW. The CWD increases during the first months of the year, reaches a plateau in May (between 10 and 11 days for the EnsMedian) until July. Then it increases again and reaches a maximum in September (14 days for the EnsMedian). Interestingly, in the observations, the FRQW presents a bimodal structure in the observations, with its highest values in May and September (76 % and 79 % for the EnsMedian, respectively).

In the observations, the uncertainties in the SDII, R20mm, R95p and RX5day are larger for the boreal spring, summer and fall, compared to their values during the boreal winter. The GCMs spread for SDII is comparable to the observations' spread in JAS, despite the lower models' EnsMedian value compared to observations' EnsMedian. For example, in July, the SDII 10th - 90th percentile values ranges between 7.5 and $13 \, \mathrm{mm \cdot day^{-1}}$. For R20mm, R95p and RX5day, the models' spreads are larger than the spreads in the observations.

In JAS, there is a difference of one day between the R20mm observations and models EnsMedians. The 10th - 90th percentile values range between 1.7 and $5 \, \mathrm{days}$ for the models. Interestingly, in July and August, both models and observations EnsMedians have the same value, $42 \, \mathrm{mm}$. The maximum difference among them in the annual cycle is of the order $8.5 \, \mathrm{mm}$. In August, the 10th (90th) percentile value of R95p is $35(79) \, \mathrm{mm}$ for the models, against $33(57) \, \mathrm{mm}$ in observations.

For the RX5day, the maximum difference between the models and observations EnsMedians is $20 \, \mathrm{mm}$ (found in June). The maximum EnsMedian value in observations (models) is reached in August, $82(74) \, \mathrm{mm}$.

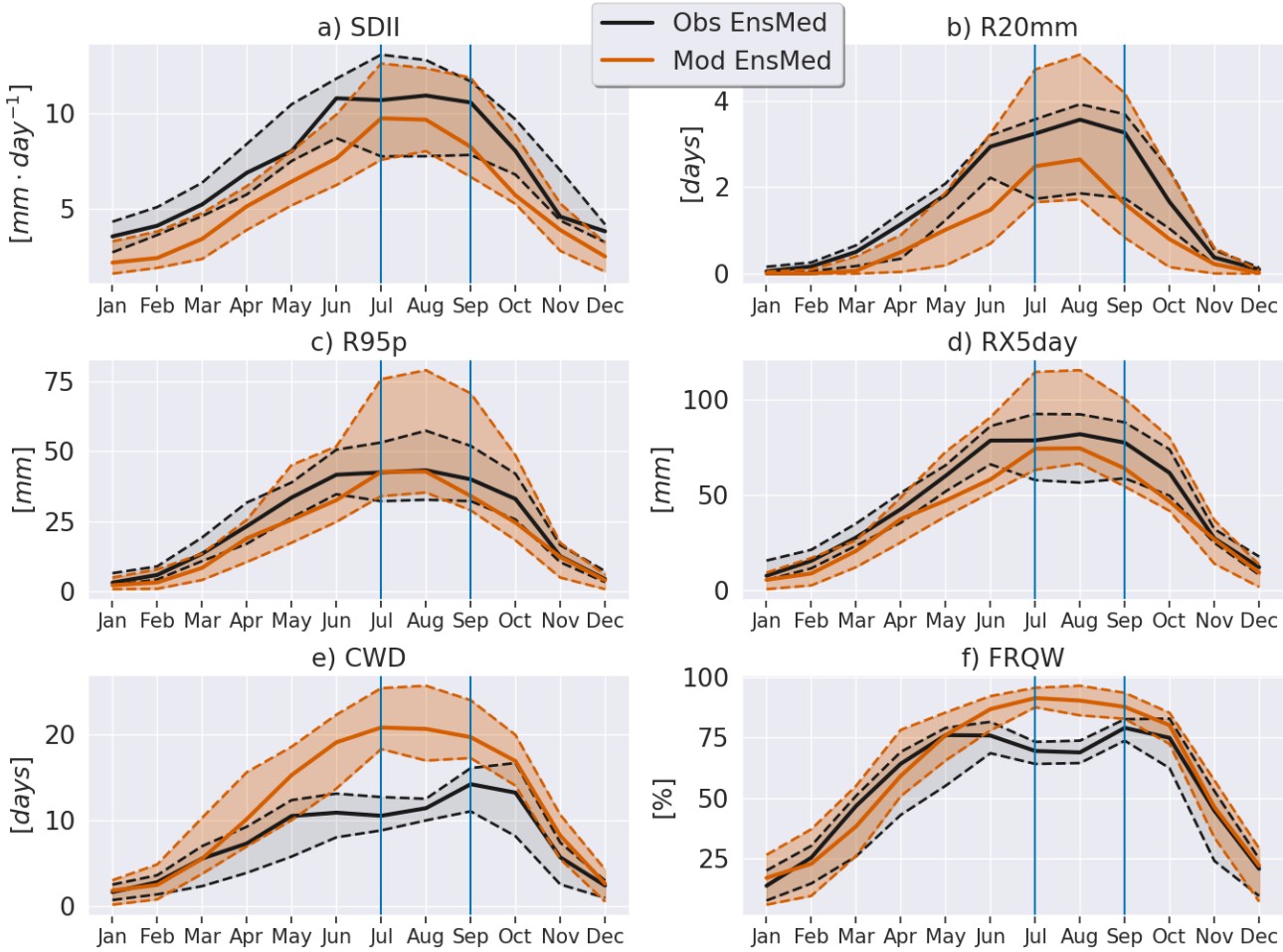

**Figure 1.** Annual cycle of the six extreme rainfall indices over the Guinea Coast during the 1995 – 2014 period. The black (orange) curves indicate the median values of the 6 (24) observations (models). The gray (orange) shading indicates the 10th - 90th percentile range for 6 (24) observations (GCMs). These percentiles are marked with the dashed curves (in black for observations, and in orange for models). The two vertical lines indicate the JAS season.

Finally, the models simulate too many consecutive wet events compared to observations, mainly during the boreal summer. There is no overlapping of the spread of the models and observations in JAS, for CWD and FRQW. The models EnsMedian CWD reaches 20 days in July, which is two-times the value of the observations EnsMedian (Fig. 1 (e)). Remarkably, the spread of the FRQW is reduced in JAS, compared to the other seasons, in both models and observations. The models EnsMedian FRQW value in July is 91%, against 69% for the observations EnsMedian (Fig. 1 (f)). Noteworthy, we found no clear linkages between the JAS CWD and FRQW biases and the JAS long-term projected mean changes in the extreme rainfall indices (not shown).

In summary, there is a large variability among the different observed datasets, in the annual cycle for most of the extreme indices. The uncertainties in the observed SDII are comparable to the uncertainties in the models, although the models EnsMedian indicate weaker intensities compared to the observations' EnsMedian. For the other indices, the spread of the models is higher than the spread of the observations.

## 3.2 Mean JAS rainfall extremes in Guinea Coast under present-day conditions

First, we present the observed and simulated JAS spatial distribution of extreme rainfall indices over West Africa in the present-day period (Fig. 2). The EnsMedian JAS spatial distribution of the indices in the six observational data is shown in contours in Fig. 2 (and in colours in Fig. S5). Most of the indices show maximum values over the Guinean Highlands (in the western Guinea Coast) and the Cameroon mountains (east of $5\,°$E), and moderate values in the center Guinea Coast (between $7.5\,°$W and $5\,°$E). From $10\,°$N to the north, there is a gradual decrease in the wet indices. Focusing on Guinea Coast, the simple daily intensity index (SDII) ranges between $6\,(6)$ and $10\,(8)\,\mathrm{mm}\cdot\mathrm{day}^{-1}$ over the center of the region in the models (observations) EnsMedian (Fig. 2 (a)). Figure 3 (a) indicates a robust EnsMedian wet bias of $1$ to $2\,\mathrm{mm}\cdot\mathrm{day}^{-1}$ in the SDII over the center of Guinea Coast, and non-robust dry biases elsewhere. Over the Sahel, there is a homogenous dry bias stretching from the west to the east, with robust values exceeding $2\,\mathrm{mm}\cdot\mathrm{day}^{-1}$ in some locations.

The R20mm index presents a non-robust EnsMedian overestimation of $1$ to $2\,\mathrm{days}$ over the center Guinea Coast (Fig. 3 (b)). The R20mm bias is, however, negative and robust over the eastern Cameroon and the western Sahel. For the R95p and RX5day variables, the EnsMedian biases are positive and robust among the different observations, mainly over the center Guinea Coast. The bias values can reach $50\,\mathrm{mm}$ and $30\,\mathrm{mm}$ for R95p and RX5day, respectively (Fig. 3 (c)-(d)). For each of these two variables, Sahel models EnsMedian exhibits weaker values compared to the observations, and this bias is robust north of $15\,°$N in general.

The CWD and FRQW summer values in the northern hemisphere are uniformly overestimated by the GCMs EnsMedian over the Guinea Coast, and this bias is coherent among the different observations (Fig. 3 (e)-(f)) . The spatial distribution of CWD biases indicates values between $10$ and $30\,\mathrm{days}$ in the center Guinea Coast, and these values exceed $30\,\mathrm{days}$ east of $7.5\,°$W. The CWD biases are weaker over Sahel (between $0$ and $-5\,\mathrm{days}$). The FRQW positive biases range between $10$ and $40\,\%$ over Guinea Coast, and exceed $40\,\%$ along the coast (between $7.5\,°$W and $0\,°$E). Over the Sahel, FRQW is uniformly underestimated.

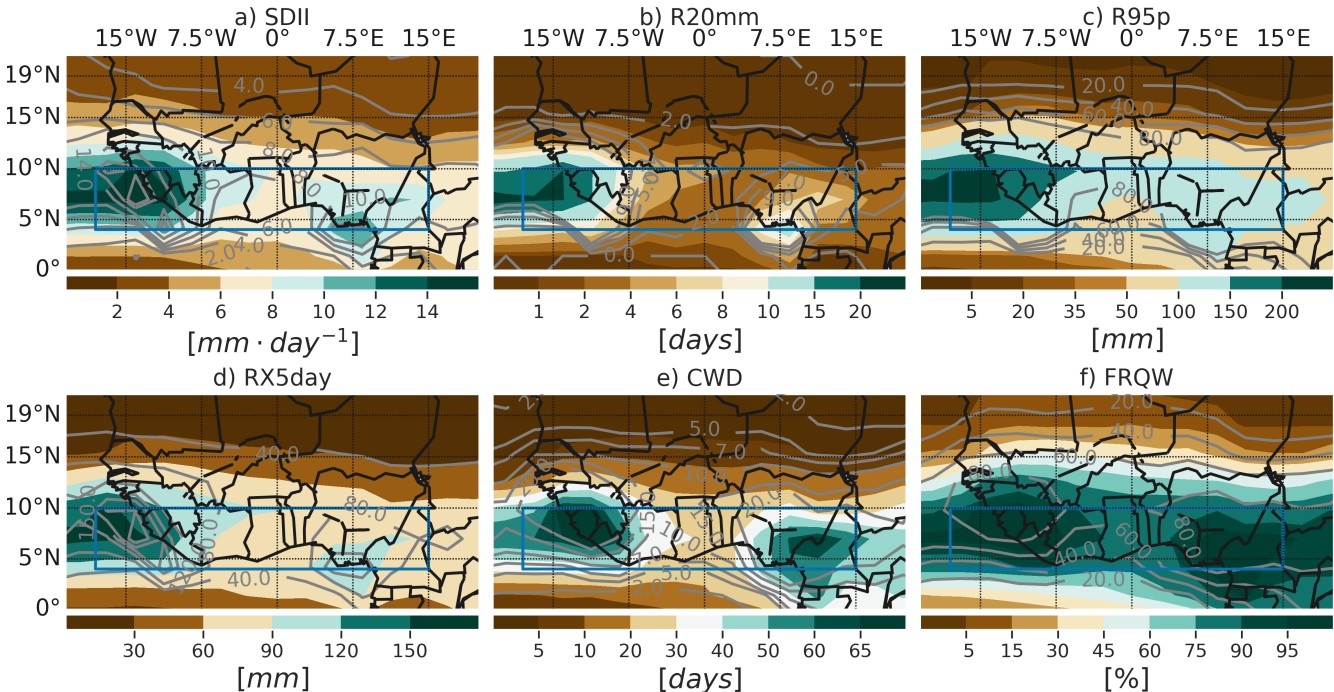

**Figure 2.** EnsMedian distribution of the mean JAS extreme rainfall indices over the 1995-2014 period from 24 CMIP6 models. The blue rectangles indicate the Guinea Coast region. Contours in gray show the observations EnsMedian values.

Figure 4 presents a summary of the models' performances in simulating the extreme characteristics over Guinea Coast. It shows that the relative bias (in %) is higher in the CWD than in the other variables. Compared to the observations, all the CWD values in the models are higher in magnitude, with a %BIAS median value of $108\%$ (Fig. 4 (a)). This means, in other words, that the CWD values in the GCMs are twice as high as in the observations. This can also be seen in Fig. 4 (b), where the EnsMedian value of the NRMSE is $1.3$, which is the highest NRMSE value compared to the other extreme rainfall indices. This overestimation of the CWD by the GCMs over the coastal regions has already been reported by Faye and Akinsanola (2021). Despite the substantial differences in magnitude compared to the observations, the CWD spatial correlation between models and observations gives a median value of $0.5$. The corresponding TSS amounts to $0.4$.

Compared to the other extreme indices, the models present a consistent positive bias in the JAS FRQW (Fig. 4 (a)). The EnsMedian %BIAS amounts to $23\%$, which also confirms results in Fig. 1 (f). Likewise, the spread of the NRMSE is small, and its median value is 0.3 (Fig. 4 (b)). However, there is a large spread in the spatial correlations between the mean JAS values in the models and the values in observations (Fig. 4 (c)). The PCC and TSS values of the FRQW present similar characteristics with median values of $0.4$ and $0.3$, respectively.

R95p and RX5day are indices of intensity and duration of extreme rainfall events. The performance statistics of these indices indicate no clear representation compared to the different observations in term of %BIAS (Fig. 4 (a)). The median values of

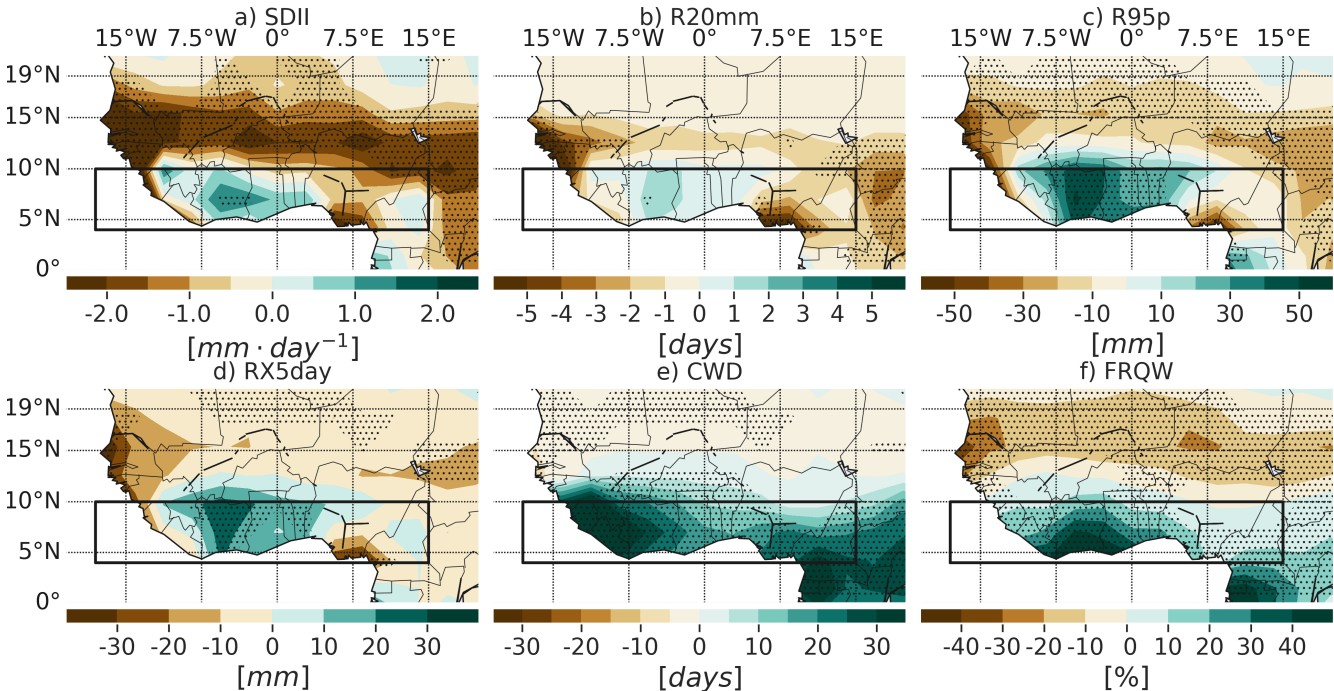

**Figure 3.** JAS mean biases of the 6 rainfall-based extreme indices over the 1995-2014 period. The biases (model minus observation) for each model are computed relative to each of the 6 observations. In total, there are 6 observations × 24 models = 144 different biases. The stippling indicates regions where two-third of the biases agree on the sign of the EnsMedian of the 144 biases. The black rectangles indicate the Guinea Coast region.

this latter are $2.7\,\%$ and $-2.4\,\%$ for R95p and RX5day, respectively. Although the sign of the bias is not coherent among the different observations, more than $75\,\%$ of the NRMSE are below $0.63\,(0.52\,)$ for R95p (RX5day) (Fig. 4 (b)). The EnsMedian values of PCC and TSS are $0.6\,(0.6\,)$ and $0.4\,(0.5\,)$, respectively for R95p (RX5day) (Fig. 4 (c)-(d)).

Finally, more than half of the model's biases in SDII and R20mm (over Guinea Coast) are negative, relative to the observations (Fig. 4 (a)). The EnsMedian value of these two variables amounts to $-11\,\%$ and $-20\,\%$, respectively. As in the case of R95p and RX5day, there are more than $25\,\%$ cases where the SDII and R20mm are overestimated. Moreover, $75\,\%$ of the models present an NRMSE lower than $0.5\,(1\,)$, for the SDII (R20mm) index. The EnsMedian value of the PCC and TSS are $0.6$ and $0.5$ for SDII, respectively. For R20mm, the EnsMedian value of PCC and TSS corresponds to $0.7$ and $0.6\,$, respectively,

and they are the highest values compared to the other extreme indices. It is important to note that the area defining the Guinea Coast is large, and the sign of the biases is not uniform over the region for all the variables. Thus, the area average over the region is subject to compensating effects.

In summary, over Guinea Coast, it rains too frequently and during too many consecutive days in the GCMs, compared to the observations. However, the intensity of daily wet events is weaker, and the number of very heavy precipitation days (with

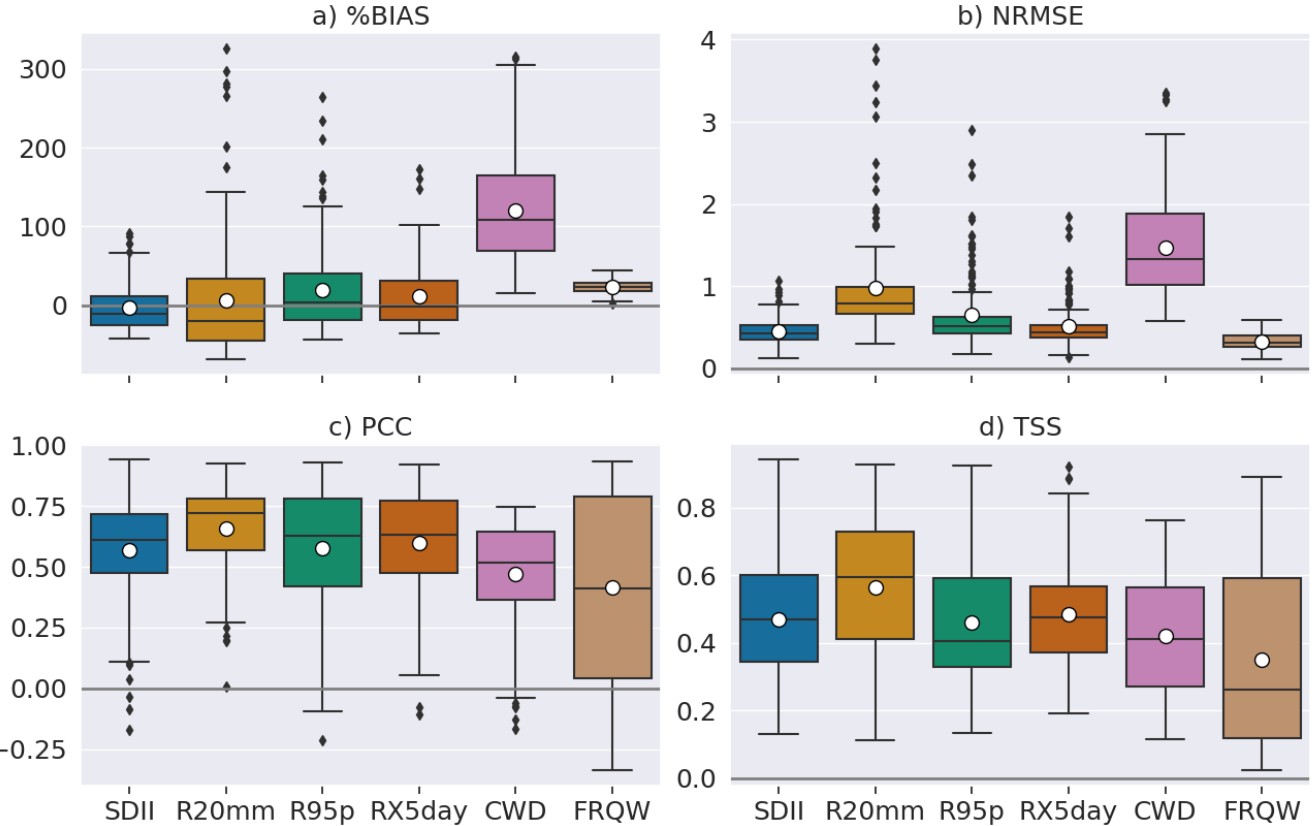

**Figure 4.** Performance of the 24 GCMs relative to the 6 rainfall observations, in representing the spatial distribution of the rainfall extreme indices over the Guinea Coast. Each boxplot indicates the distribution of $24 \times 6 = 144$ different values. The median (mean) values of the statistics are represented by the horizontal bar (white circle). Outliers are plotted individually with the black markers.

an intensity above $20\,\mathrm{mm}$) is lower than observed. In contrast, very wet day intensity (with rainfall exceeding 95[th] percentile) is higher than observed. The R20mm, according to the TSS metric has the highest EnsMedian values, and can be considered as the index which is better represented in the GCMs. Moreover, the biases in the indices can be as large as the spread of the observations, mainly for R95p, CWD, and FRQW (Fig. S3). The performance metrics for the individual models across the six different observations are shown in Figs. S6 - S9. Next, we will describe the changes of the rainfall extremes over the Guinea Coast under global warming, as projected by the climate models. All the indices will be studied, regardless of their good or poor representation by the GCMs, but the biases identified here will be accounted for in our discussion.

### 3.3 Changes in the mean and variability of the JAS extreme rainfall indices over Guinea Coast

Figure 5 displays the projected EnsMedian long-term changes (2080-2099 minus 1995-2014) in the rainfall extreme indices over West Africa. The spatial patterns of the changes (relative to the present-day period) in the extreme rainfall indices over

West Africa are similar for the near-term, mid-term and long-term periods, with, in general, a gradual increase in magnitude over the three future periods. However, the near-term and mid-term changes (Figs. S10-S11) are often less robust than the long-term changes.

In particular, over the majority of West Africa, the forced change in the different extreme indices does not emerge from the IAV in the near-term future period, as also shown by Monerie et al. (2017). These authors found that the interannual forced
changes in the central (western) Sahel in August-October (June-August) becomes stronger than the IAV only from the 2060s (2080s). In agreement with Monerie et al. (2017), we found an increase in the SDII over the central to eastern regions of Guinea Coast and Sahel, which becomes stronger than the IAV from 2040s (Fig. S11 (a) and 5 (a)). Our results also show over the western areas of the Sahel, a long-term decrease in the SDII, which is stronger than the IAV, but with less agreement on the sign of the change among the models.

Analysing the various indices specifically, we found that the changes in the R95p are not robust over Guinea Coast, even at the end of the 21$^{st}$ century. This suggests a strong influence of IAV on the intensity of rainfall events exceeding the 95$^{th}$ percentile. We note that the long-term projection of these events is robust over the easternmost Sahelian region (Fig. 5 (c)). The R20mm is projected to increase over several areas of West Africa in the long-term future: easternmost Sahel and Guinea Coast, central Guinea Coast (Fig. 5 (b)), which is consistent with Akinsanola and Zhou (2019).

The FRQW index exhibits a robust decrease of $5\%$ over Guinea Coast and over western Sahel (Fig. 5 (f)). Additionally, a robust decrease in the CWD is projected in the westernmost and easternmost region of the Guinea Coast. Over the Sahel, only the decrease in the western areas is robust for the CWD index (Fig. 5 (e)). These changes in CWD are consistent with the findings of Klutse et al. (2018) under global warming of $1.5\,°C$ and $2\,°C$. Consistently, Wainwright et al. (2021) found an increase (a decrease) in the mean length of dry (wet) spells over Guinea Coast during the wet season in a future warmer
climate.

There is also a robust increase in RX5day over the Guinea Coast and the central to eastern Sahel (Fig. 5 (d)). These results show a tendency to less frequent and more intense rainfall over the Guinea Coast in a warmer climate, which happens over shorter duration. Our results are in accordance with Dosio et al. (2021), who found an increase in the mean SDII, and a decrease in the number of wet days in the June-August long-term projections for the Guinea Coast.

Figure 6 displays the average of the change in mean and standard deviation of the JAS extreme rainfall indices over the Guinea Coast for the 2021-2040, 2041-2060 and 2080-2099 periods, relative to 1995-2014. There is a tendency toward an increase in magnitude of SDII, R20mm, R95p and RX5day averages over Guinea Coast, which is gradual from the near-term to the long-term future periods (Figs. 6 (a)-(d)). The variability in these variables also exhibits a gradual increase with time in the future periods. Specifically, the changes in the mean and standard deviation of the R95p are of the same order of magnitude.

Consistent with the previous discussion, there is projected decrease in the CWD and FRQW indices over Guinea Coast (Figs. 6 (e)-(f)). The variability of the CWD will also decrease over the three future periods, with higher magnitudes in the long-term period compared to the others. For the FRQW, however, no change in the variability is projected in the near-term period, while there is a slight increase from the mid-term to the long-term future periods.

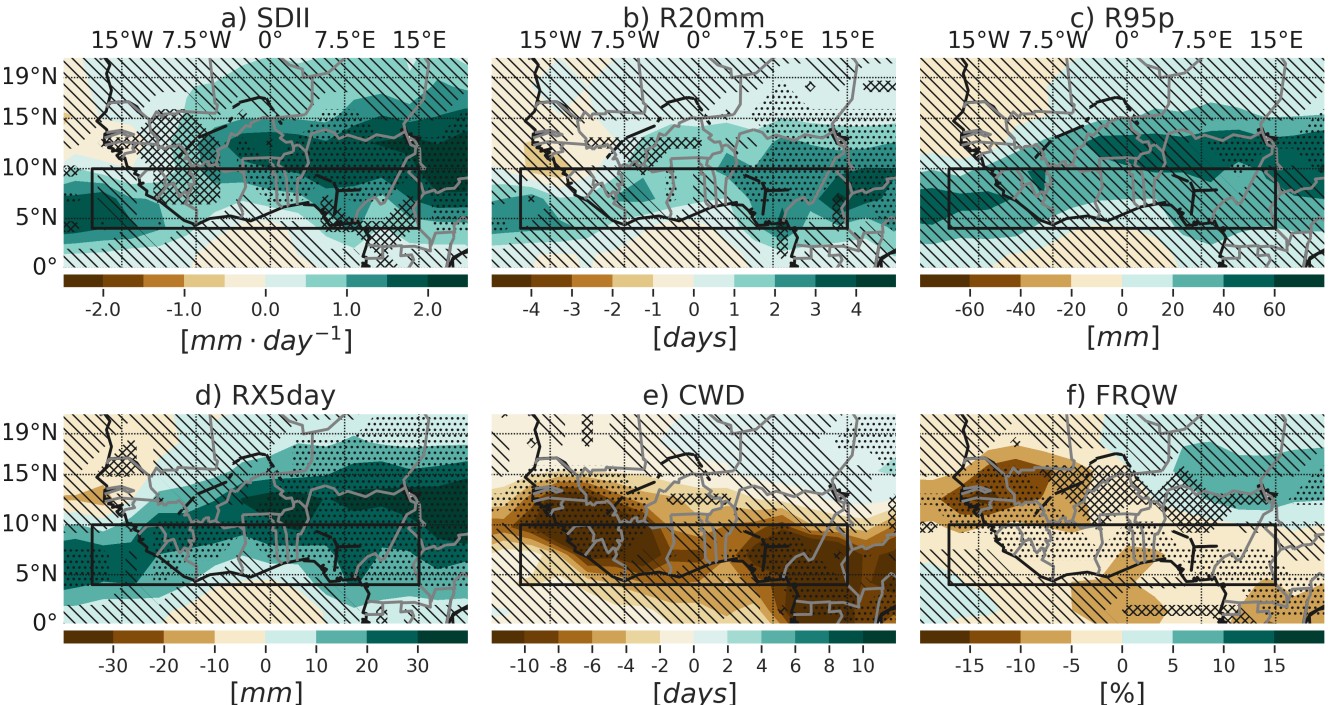

**Figure 5.** Projected multi-model ensemble long-term median change in the JAS rainfall extreme indices over West Africa, relative to the present-day period (2080-2099 minus 1995-2014). The stippling indicates regions where the change robustly emerges from internal variability (at least 66% of the models show a change greater than the IAV and at least 80% of the models agree on the sign of change); hatching (\) indicates regions where the change is non-robust (fewer than 66% of the models show change greater than the IAV); crossed lines (X) indicate conflicting signals where at least 66% of the models show change greater than the IAV, with less than 80% agreement on the sign of the change. Blank areas without stippling/hatching and crossed lines are due to a non-interpolation between the three robustness categories.

In summary, under future global warming, rainfall events over the Guinea Coast are projected to intensify (increase in the daily rainfall intensity over wet days), to be less frequent, to happen over shorter duration, and to be more extreme (increase in the amount of rainfall during very wet days). This would increase the exposure of the West-African population to flooding events in a warmer future. According to the climate scenarios, for which future global warming is limited to $1.5\,°C$, $2.4\,°C$ and $3.5\,°C$ by 2100, people born in 2021 in Sub-Saharan Africa will experience, during their lifetime, $4.6$ , $8$ , and $8.6$ times more river flooding events than without climate change, respectively (Thiery et al., 2021, https://myclimatefuture.info/, last access: 11-08-2022). This increase is two to fourfold higher than the flooding events experienced by people in the same area, born in 1960, and highlights the climate urgency in reducing our greenhouse gas emissions for future generations. Besides, we found that the overall signs of the extreme indices long-term changes over Guinea Coast are in accordance with Akinsanola and Zhou (2019). However, the results of these authors are obtained from RCM-CMIP5 projections and show more robust changes over West Africa, without taking into account the IAV influence. Moreover, there is a projected increase in the variability of most

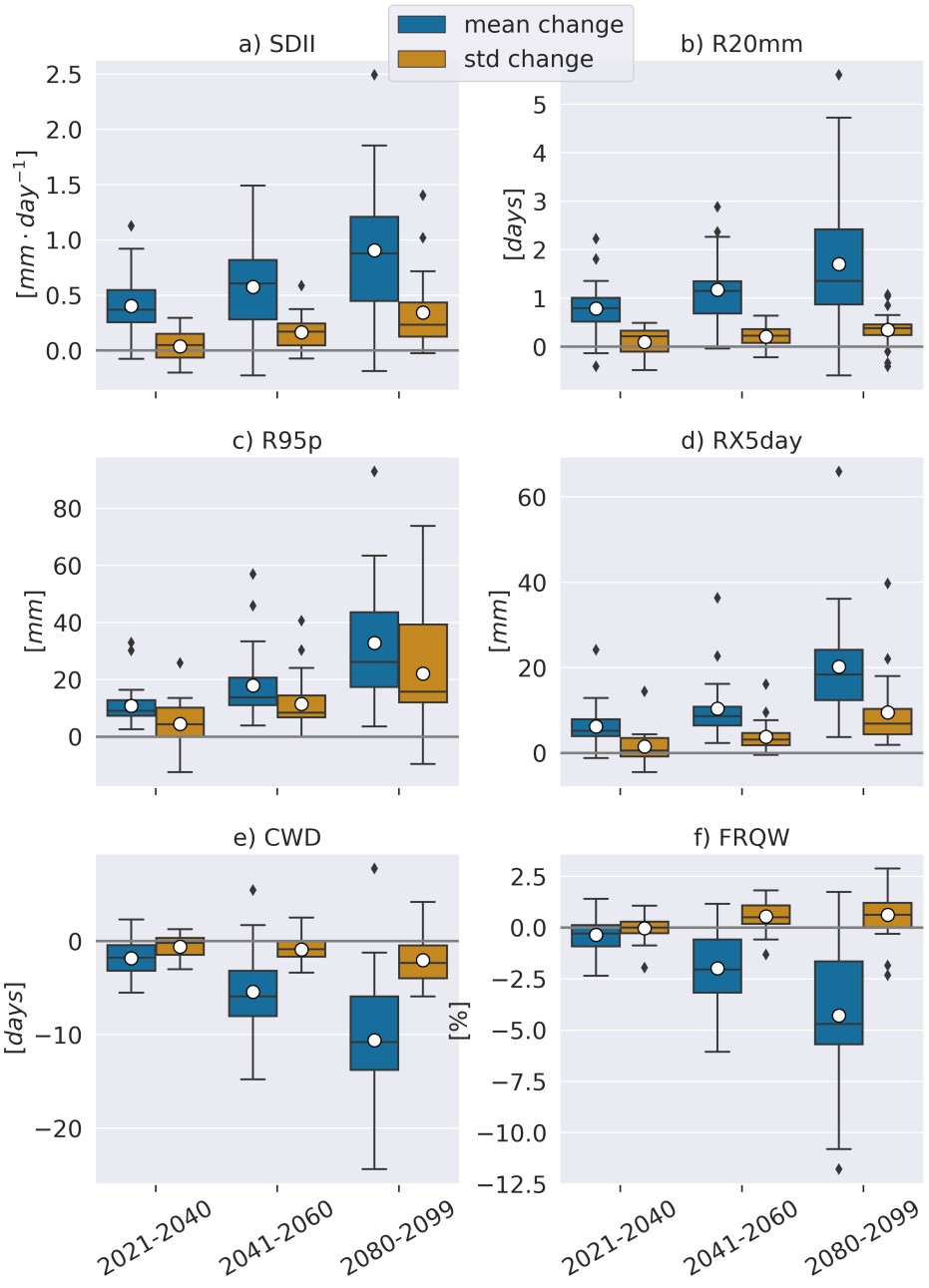

**Figure 6.** Boxplots of the average of the near-term, mid-term and long-term changes (relative to the present-day period) in the mean and standard deviation of the rainfall extreme indices over the Guinea Coast. The averages were computed from 24 different GCMs. Outliers are indicated by the black marks (losanges). The median (mean) value of each distribution is indicated with a black horizontal bar (a white circle).

of the rainfall-based extreme indices, which is consistent with Akinsanola et al. (2020). The multi-model median long-term percentage of changes in mean (variability) averaged Guinea Coast, for SDII, R20mm, R95p, RX5day, CWD and FRQW correspond to $11 (29)$, $35 (22)$, $20 (29)$, $21 (48)$, $-23 (-12)$ and $-5 (37)\%$, respectively (Fig. S12-S13).

Wet and dry extreme events over Guinea Coast could also happen under positive or negative phases of the Atlantic equatorial mode, respectively. As forced changes induced by future global warming can be exacerbated or damped by internal climate variability, study of the impacts of known internal climate modes of variability becomes crucial. It is the objective of this study, to understand the future changes in the rainfall extreme indices over the Guinea Coast that are connected with the AEM on the interannual timescales. The impact of the AEM on the extreme rainfall variability during the boreal summer is assessed here by regressing the JAS extreme indices onto the standardized JAS AEM index.

## 4   Impact of the AEM on the rainfall extreme events over the Guinea Coast

### 4.1   AEM impact on the rainfall extreme events under present-day climate conditions

Over the majority of the Guinea Coast region, positive anomalies of the extreme rainfall indices are associated with positive phases of the AEM (Figs. 7, S14). In GCMs, the extreme responses to this oceanic mode of variability are robust, according to the two-third sign agreement metric, except for the CWD index. In all cases, there is an anomalous increase (decrease) in the wet indices, in association with warm (cold) phases of the AEM over the Atlantic oceanic region (between the equator and the southern limit of Guinea Coast). A noteworthy feature of the anomalous patterns is the lack of robust signals over the majority of the Sahelian region. Particularly, the westernmost area of Sahel exhibits an opposite phase relationship with the AEM, for CWD and FRQW indices (Fig. 7 (e)-(f)), compared to Guinea Coast.

The GCMs EnsMedian regression patterns associated with one standard deviation of the AEM index show an increased intensity of rainfall over wet days (SDII, $0.2$ to $0.6\,\mathrm{mm}\cdot\mathrm{day}^{-1}$) over the Guinea Coast (Fig. 7(a)). There is about one-day increase in the number of days with very heavy precipitation (R20mm), during positive AEM events (Fig. 7(b)). Warmer than average sea surface conditions in the eastern equatorial Atlantic favour an increased amount of rainfall during wet days (R95p) (Fig. 7(c)). Over the Guinea Coast, the anomalies related to one standard deviation of the AEM index range between $10$ and $30\,\mathrm{mm}$ for R95p. The maximum precipitation over five consecutive days (RX5day) is also intensified over Guinea Coast (Fig. 7(d)), with positive anomalies ranging between $2$ to $6\,\mathrm{mm}$. The anomalous EnsMedian patterns of the maximum consecutive wet days (CWD) related to a warm phase of the AEM shows a non-robust increase of one to three days in wet spells duration (Fig. 7(e)). Similarly, the frequency of wet days over the Guinea Coast is higher up to $4\%$ under warm sea surface conditions in the eastern equatorial Atlantic (Fig. 7(f)).

Fig. 8 displays the biases in the regression patterns associated with the AEM, for the different extreme indices. Over Guinea Coast, SDII and R20mm show no systematic biases relative to the different observations (Fig. 8 (a)-(b)). In the eastern Guinea Coast, there is an overestimation (underestimation) of the R95p and RX5day (CWD) anomalies by the GCMs (Fig. 8 (c)-(e)). Regarding the FRQW, there is a robust underestimation of its anomalies related to one standard deviation of the AEM index over West Africa (Fig. 8 (f)).

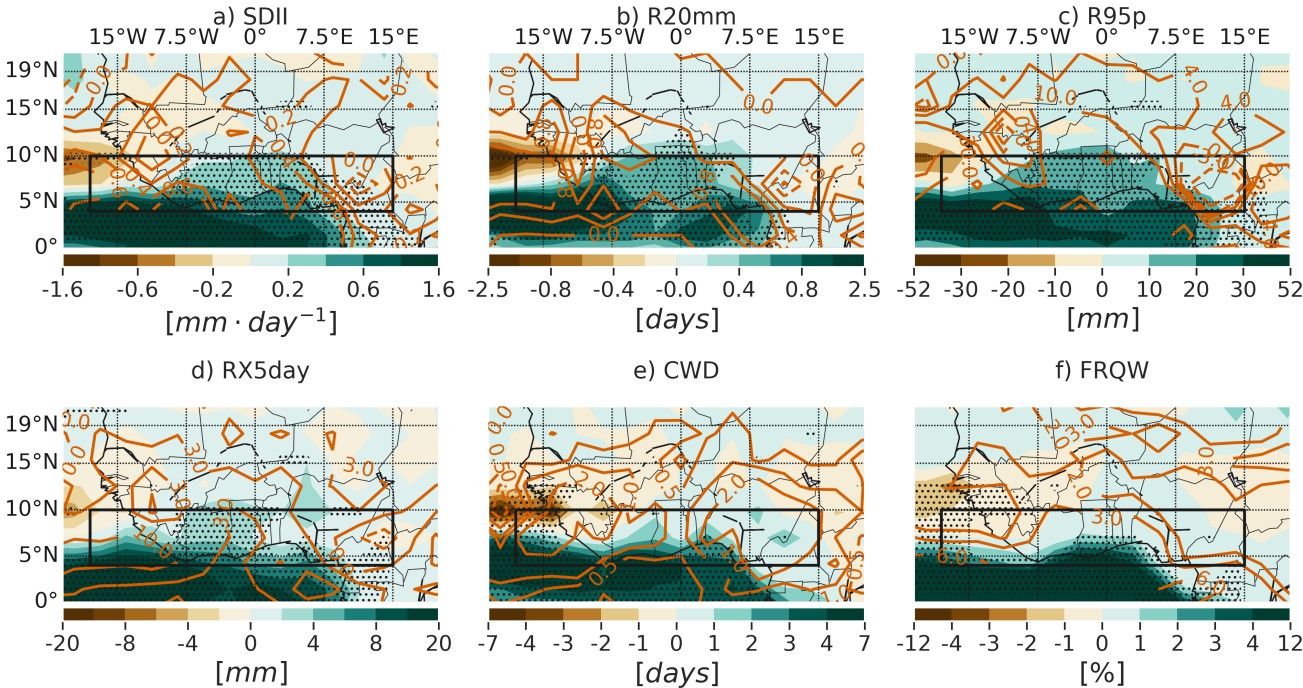

**Figure 7.** Maps of the JAS rainfall extreme indices regressed onto the standardized JAS AEM SST index over the 1995-2014 period. The stippling represents grid points where two thirds of the models agree on the sign of the EnsMedian regression coefficient across 24 GCMs. Contours indicate the multi-observation median values of the regression coefficients from six different rainfall data. The black rectangles indicate the Guinea Coast region.

Figure 9 displays the performance metrics for the GCMs extreme rainfall indices related to the AEM. The EnsMedian values of the %BIAS correspond to $-17$, $-30$, $34.5$, $6$, $-31$ and $-75\%$ for SDII, R20mm, R95p, RX5day, CWD and FRQW, respectively (Fig. 9 (a)). The biases for FRQW are the largest. Moreover, the NRMSE of R95p has the highest values, compared to the other variables (Fig. 9 (b)) with an EnsMedian value reaching $4$. In more than $75\%$ of the cases, the pattern correlation (PCC) between the individual models and the different observations is lower than $0.5$ (Fig. 9(c)). Particularly, the CWD anomalies are negatively correlated with the observation patterns in much of the models (the correlations are weak, however). According to the TSS, the response of the SDII index to AEM is better represented by the models, and this is indicated by a median TSS value of $0.4$ among the set of $24$ GCMs $\times$ $6$ observations = $144$ values (Fig. 9 (d)). This can also be seen from Fig. S18, where much of the good performances of the different models relate to SDII. The individual %BIAS, NRMSE and PCC statistics for the individual models are available in Figs. S15-S17.

In summary, the anomalous warming of the eastern equatorial Atlantic during positive phases of the AEM leads to positive anomalies in the wet extreme indices over the Guinea Coast. This result is consistent with Atiah et al. (2020); Diatta et al. (2020). However, according to the Taylor skill score, the performance of the models in representing the anomalous responses

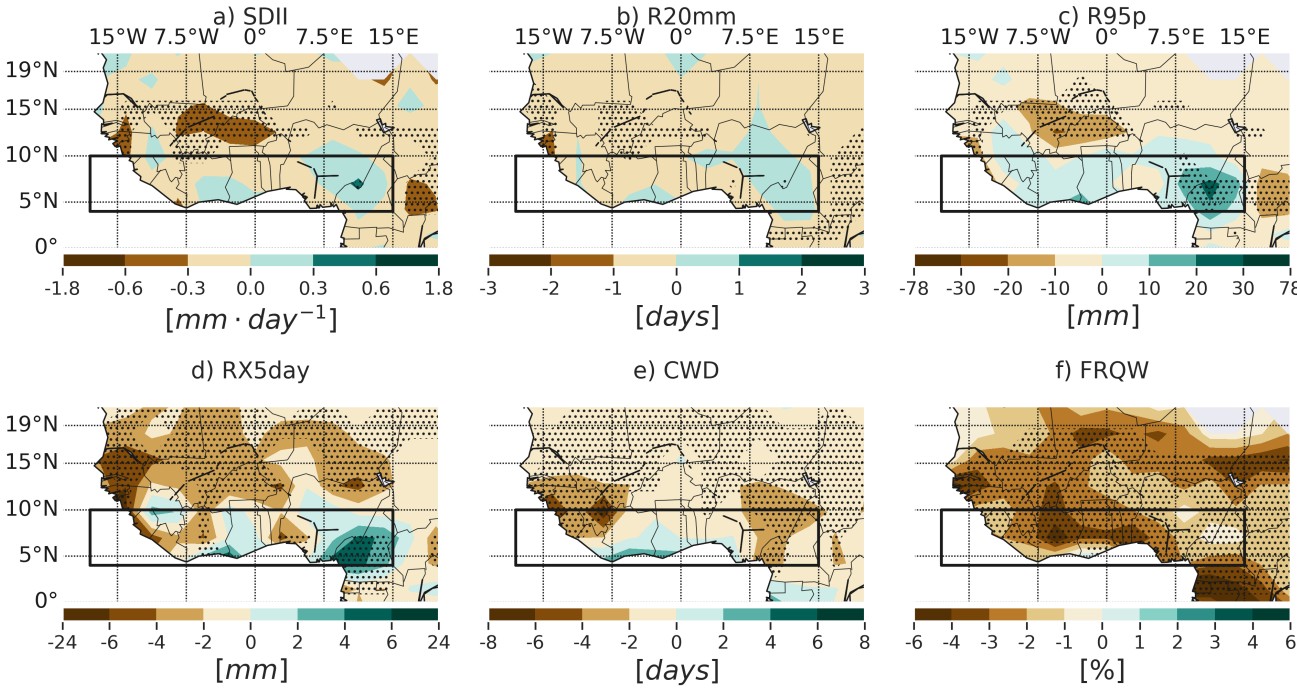

**Figure 8.** Spatial distribution of the EnsMedian bias (model minus observation) relative to six observations for the JAS regression pattern of the extreme rainfall indices related to one standard deviation of the Atlantic equatorial mode SST index. In total, there are six observations × 24 models = 144 different biases. The stippling indicates regions where two-third of the biases agree on the sign of the EnsMedian of the 144 biases. The black rectangles indicate the Guinea Coast region.

in the different extreme indices is poor to modest, although some indices exhibit a good pattern correlation coefficient. Over Guinea Coast, the biases in the indices (except FRQW) are relatively small compared with the spread in the regression patterns of the observations (Fig. S19). Furthermore Worou et al. (2022) showed that the current GCMs can replicate the SST variability associated with the AEM over the tropical Atlantic Ocean, but they struggle to replicate the spatial distribution of the observed
rainfall response to its phases over the tropical Atlantic and the Guinea Coast. In the next section, the future changes in the Guinea Coast extreme rainfall indices related to the AEM will be assessed.

## 4.2  Future changes in rainfall extreme events associated with the AEM

The changes of the equatorial Atlantic mean state under the highest greenhouse gas emission scenario (SSP5-8.5) lead to a weakened variability of the AEM in the future (Worou et al., 2022; Crespo et al., 2022; Yang et al., 2022). Worou et al.
(2022) showed a projected weakening of the trade winds climatology over the equatorial Atlantic and a deeper thermocline in the eastern equatorial Atlantic which would lead to a reduction of the coupling between the surface and the thermocline depth. This implies a future weakening of the Bjerknes feedback (Bjerknes, 1969) in the equatorial Atlantic that explains the

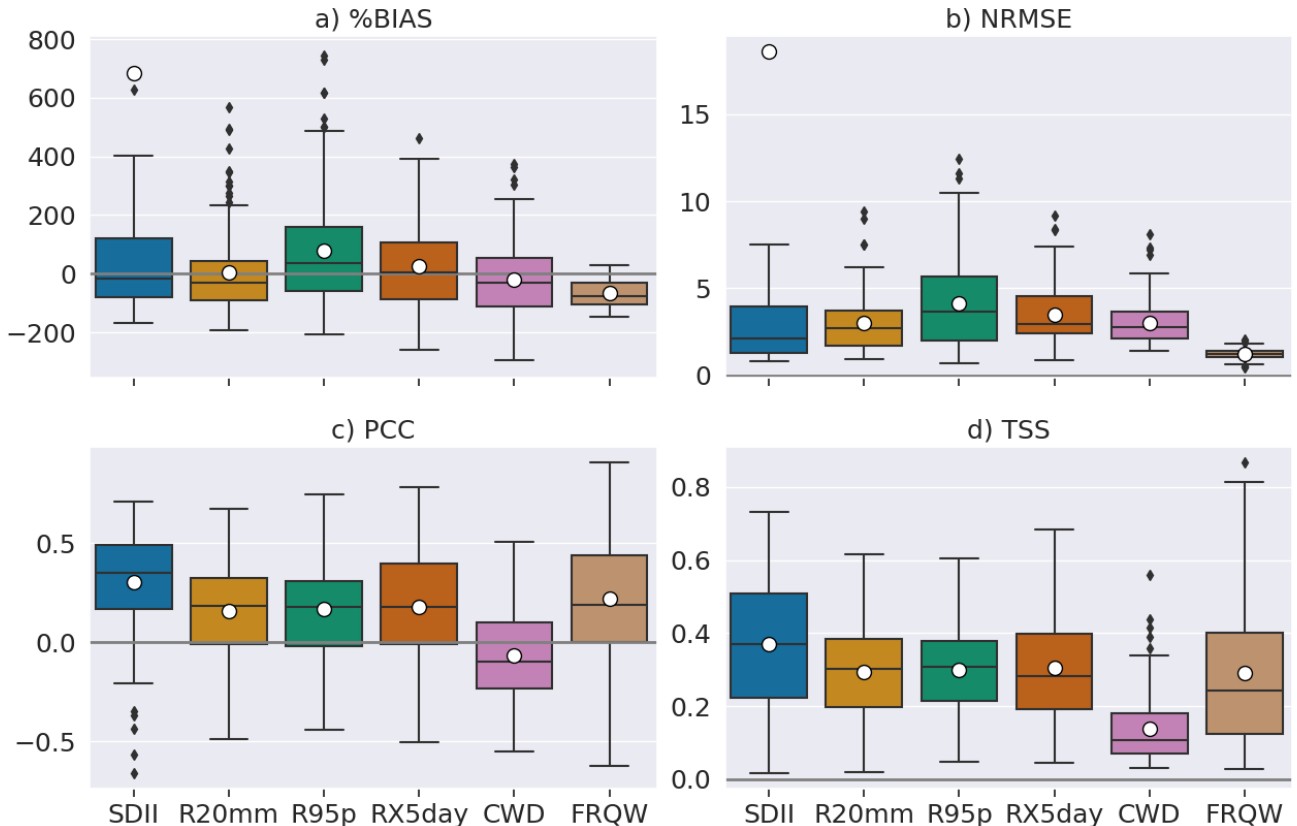

**Figure 9.** Performance of the 24 GCMs relative to the six rainfall observations, in representing the spatial distribution of extreme rainfall responses to AEM over the Guinea Coast. Each boxplot indicates the distribution of $24 \times 6 = 144$ different values. The median (mean) values of the statistics are represented by the horizontal bar (white circle). Outliers are plotted individually with the black markers. Note that all the outliers for SDII are not plotted, to improve the readability of the figure.

reduced variability of the AEM under global warming, and the weaker future impact of the AEM on the rainfall over the equatorial Atlantic and the Guinea Coast. These results are confirmed by Crespo et al. (2022), who found a future reduction of the AEM variability in a warmer climate, mainly due to a weakening of the third component of the Bjerknes feedback (the SST response to the variations of the thermocline depth). Yang et al. (2022) also found a reduced AEM variability. They underlined a greater role of a more stable tropical Atlantic atmosphere background in a future warmer climate (Jia et al., 2019), in reducing the AEM variability, compared to the weakening associated with the deepening of the eastern equatorial Atlantic mean thermocline. Subsequently, the variability of the JAS rainfall over the equatorial Atlantic and Guinea Coast that is related to the different phases of AEM is reduced (Worou et al., 2022).

The magnitude of the change in the Guinea Coast SDII-AEM EnsMedian is about $-0.1\,\mathrm{mm} \cdot \mathrm{day}^{-1}$ in the three future periods (Fig. 10(a)). The EnsMedian values of the percentage of changes in the average of SDII response to AEM over Guinea Coast correspond to $-37\%$, $-32\%$ and $-87\%$ in the near-term, mid-term and long-term periods, respectively (Fig.

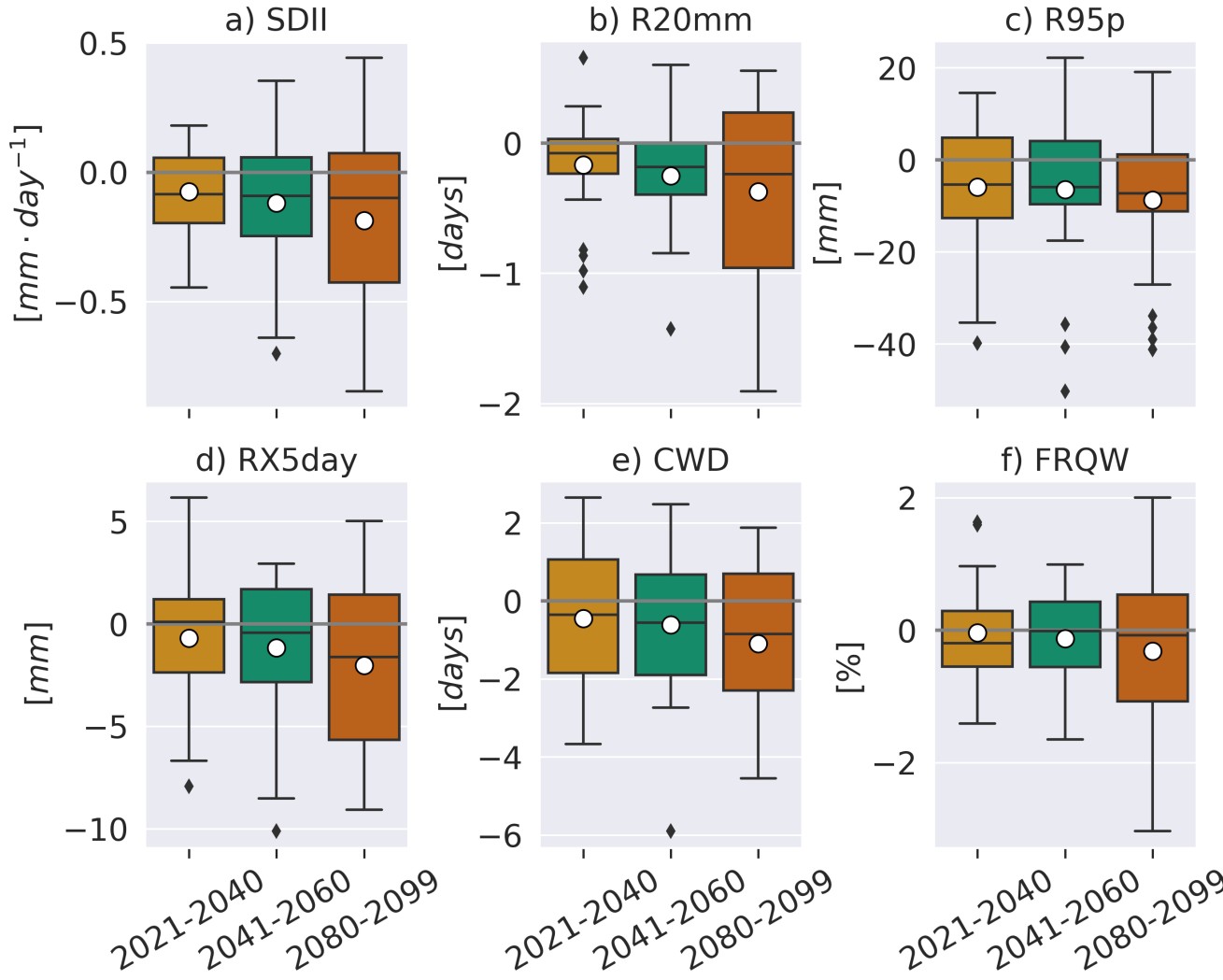

**Figure 10.** Near-term, mid-term and long-term changes in the Guinea Coast area average of the JAS extreme rainfall responses to one standard deviation of JAS AEM index. Each boxplot represents the distribution of 24 GCMs. The median (mean) value of each distribution is indicated by a black horizontal bar (a white circle).

S20 (a)). However, less than two-third of the models agree on the sign of these changes, whatever the future considered. The spatial distribution of the long-term change in the SDII anomalies over the Guinea Coast indicates values that reach $-0.4\,\mathrm{mm}\cdot\mathrm{day}^{-1}$, which is robust only over the central Guinea Coast (Fig. 11(a)). The maps corresponding to the near-term and mid-term changes in the regression patterns are available in Figs. S21-S22. These maps show weak changes in the regression patterns over Guinea Coast, compared to the long-term future changes. The regression maps corresponding to the four different periods are also displayed in Fig. S23.

The number of days with very heavy precipitation associated with one standard deviation of the AEM also shows a small decrease less than one day in the long-term future period compared to the present-day period (Fig. 11(b)). The average of the R20mm changes relative to 1995-2014 over the Guinea Coast equals $-0.1$, $-0.2$ and $-0.2\,\mathrm{days}$ in the 2021-2040, 2041-2060, 2080-2099 periods, respectively (Fig. 10 (b)). The multi-model median values corresponding to the percentage of changes in the area averaged R20mm regression coefficients over the Guinea Coast lie between $-29\%$ and $-64\%$ over the three future periods (Fig. S20 (b)).

The spatial patterns of the future changes in the variability of rainfall during the very wet days (R95p) that is related to the warm and cold AEM events indicate a reduction in R95p magnitudes over most of the Guinea Coast areas west of 5°E (Fig. 11(c)). Elsewhere over Guinea Coast, there is no evidence of a robust change in the R95p pattern. The EnsMedian change in the average of R95p over the Guinea Coast ranges from $-5.3\,\mathrm{mm}$ to $-7.3\,\mathrm{mm}$ over the three future periods (Fig. 10(c)). The multi-model median values of the Guinea Coast R95p change percentage (and the agreement among the models) corresponds to $-41\%$ (62%), $-38\%$ (67%) and $-50\%$ (67%) in the near-term, mid-term and long-term future periods, respectively, relative to the present-day period (Fig. S20 (c)).

Lower maximum five days precipitation related to one standard deviation of the AEM index averaged over the Guinea Coast are projected in the 2021-2040, 2041-2060 and 2080-2099 periods, compared to their magnitude in the 1995-2014 period (Fig. 10(d)). Over the mid-term and long-term future periods, the RX5day EnsMedian reduction is between $-0.4$ and $-1.6\,\mathrm{mm}$. The near-term EnsMedian change is positive but very close to zero (0.1 mm). In the three cases, there is a low agreement among the models in the sign of the changes averaged over Guinea Coast (less than 55%). The multimodel EnsMedian values of the percentage of change in the RX5day averaged over the Guinea Coast range between $-24\%$ and $-58\%$ over the three future periods (Fig. S20 (d)). Furthermore, Fig. 11(d) shows no robust change signal in the spatial distribution of the RX5day anomalies over Guinea Coast.

The projected decrease in the AEM variability in the future leads to a decrease in the magnitude of the wet spells corresponding to one standard deviation of the AEM index. However, the EnsMedian change in the anomalous responses of these indices averaged over the Guinea Coast is less than one day in the different three future periods (Fig. 10(e)). The EnsMedian value of the CWD change percentages averaged over the Guinea Coast (as well as the percentage of agreement) approximate $-56\%(54\%)$, $46\%(67\%)$ and $-71\%(71\%)$ for the near-term, mid-term and long-term future periods, respectively, relative to the present-day period (Fig. S20 (e)). These area-average statistics are robust for the mid-term and long-term future changes,

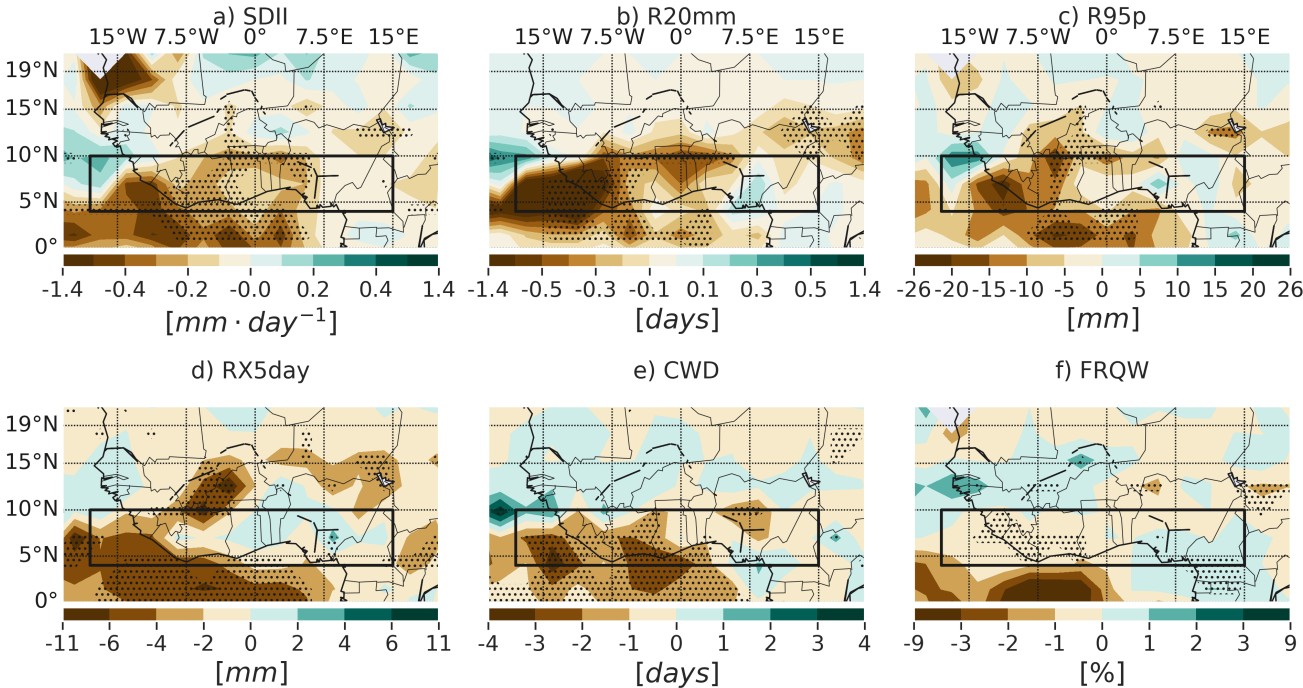

**Figure 11.** Long-term changes in the regression patterns of the JAS extreme rainfall indices associated with the standardized JAS AEM SST index (2080-2099 minus 1995-2014). The stippling indicates grid points where two thirds of the models agree on the sign of the change.

according to the two-third sign agreement metric. Figure 11(e) displays the spatial distribution of the long-term changes in the CWD response to AEM. It also shows a robust decrease in CWD magnitudes over Guinea Coast, which is between one and three days over the central and western Guinea Coast.

Finally, a reduction in the positive anomalies of the frequency of wet days over the Guinea Coast that are linked with one standard deviation of the AEM is projected in the future, under the SSP5-8.5 scenario. Robust long-term reduction up to $1\%$ are detected in the western Guinea Coast. In the eastern part of the region, non-robust weak long-term changes are projected (Fig. 11(f)). The averages of the change patterns over the Guinea Coast are equal to $-0.2\%$, $-0.0\%$ and $-0.1\%$, for the three consecutive future periods, respectively, relative to the present-day period (Fig. 10(f)). The multi-model median value of the percentage of the change in the FRQW-AEM index averaged over the Guinea Coast ranges between $-20\%$ and $-38\%$ over the three future periods (Fig. S20 (f)), but these values are non-robust. About half of the 24 GCMs agree on the sign of the EnsMedian change, which makes the result uncertain.

To quantify the contribution of AEM in the total variability of the rainfall extremes over Guinea Coast, we correlate each extreme index averaged over Guinea Coast with the standardized AEM index. The square of the correlation coefficient gives the fraction of explained variance (FEV). Figure 12 shows the distribution of the FEV across 24 GCMs, for each extreme index and for each period of the current study. There is a clear decrease in the FEV by the AEM between the present-day period

and the long-term future. However, change in the FEV is not linear with time, and this could be due to the small set of models
used in our study. Under present-day conditions, the EnsMedian FEV ranges between 13 % and 28 %, for the six indices. In
the long-term future, the FEV EnsMedian does not exceed 8 %. This indicates a weaker role of AEM in the last decades of 21$^{st}$
century, under SSP5-8.5 scenario.

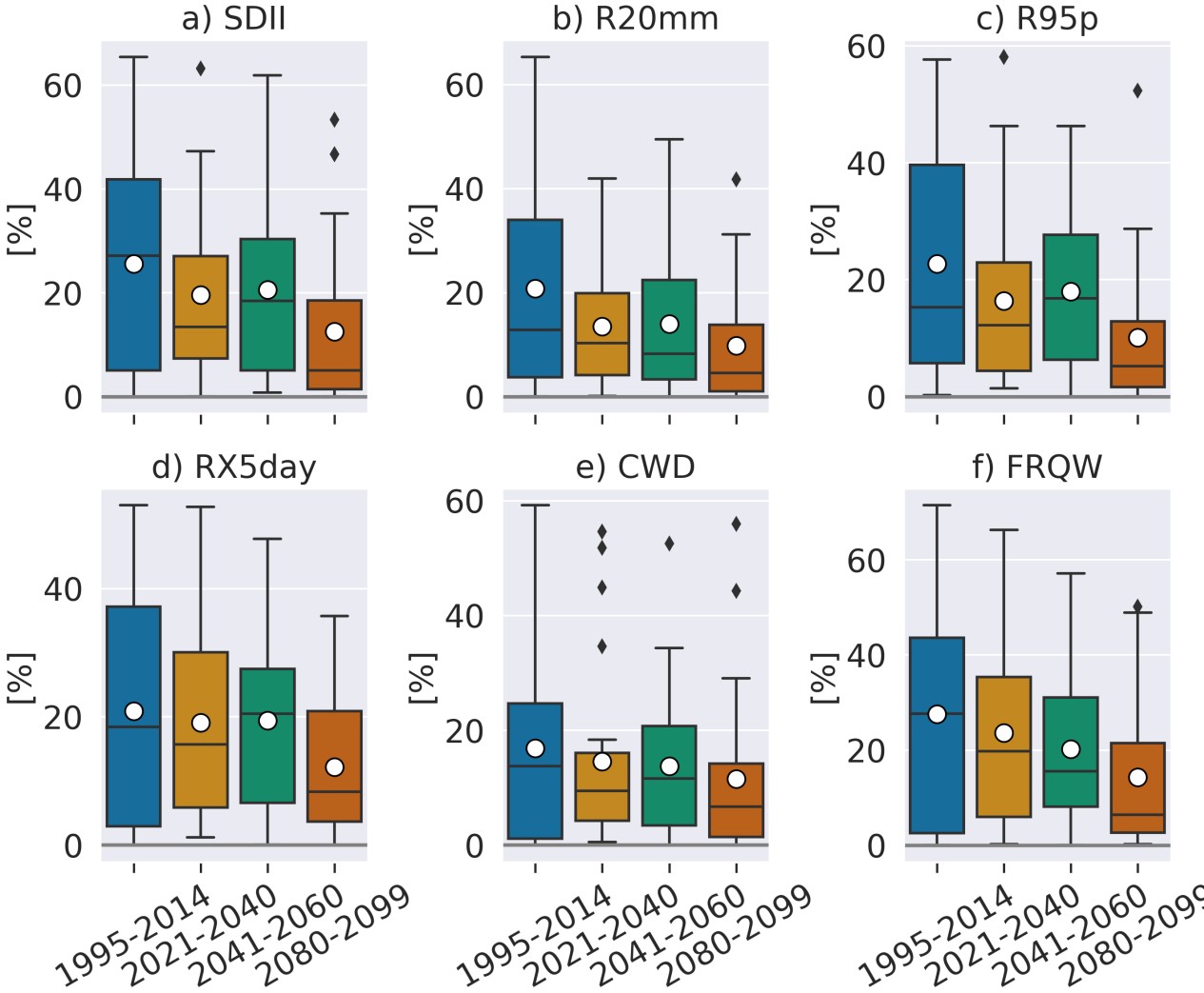

**Figure 12.** Regression of the Guinea Coast rainfall extreme indices onto the standardized AEM index: proportion of the variance explained
by the AEM over four different periods (in %). Each boxplot represents the distribution of 24 GCMs. The median (mean) value of each
distribution is indicated by a black horizontal bar (a white circle).

In summary, subsequent to the decrease in the AEM variability in a future warmer climate relative to the present-day
conditions, there is a decrease in the variability of the extreme rainfall indices associated with the AEM. For the majority

of the indices, the average of the changes in the teleconnection amplitude between the Guinea Coast and the eastern equatorial Atlantic indicates an overall enhancement in magnitude with time. Crespo et al. (2022) demonstrated that the models with the strongest warm sea surface biases exhibit the weakest change in both the AEM variability and the third component of the Bjerknes feedback. Given the positive SST biases in most of the CMIP6 models in the eastern Atlantic Basin (Richter and Tokinaga, 2020; Worou et al., 2022; Crespo et al., 2022), improving these biases would likely increase the change in the

AEM variability, suggesting that the projected weakened AEM variability is robust despite these biases. Moreover, despite the increase in the total variability of the majority of the extreme indices under global warming in the future (Section 3.3), there is a reduction in the proportion of variance explained by the AEM for these extreme indices. This result highlights a weaker role played by the AEM in explaining the future rainfall extreme variability over the Guinea Coast. Furthermore, there is more confidence in the results averaged over the Guinea Coast than in the spatial distribution of the changes in the teleconnection

patterns.

## 5   Summary and conclusions

The Atlantic equatorial mode of variability has mainly influenced rainfall changes over Guinea Coast during the last century. The current study aimed to understand the links between this oceanic mode of variability and the extreme rainfall events over the Guinea Coast under different climate conditions. The performance of 24 GCMs participating in CMIP6 in simulating six

JAS rainfall-based climate extremes indices over the Guinea Coast was assessed as well as the projected mean changes in these extremes under global warming. We used six different observation database to evaluate the models. Historical and SSP5-8.5 simulations were considered to study changes in the near-term (2021-2040), mid-term (2041-2060) and long-term (2080-2099) future periods, compared to the present-day climate period (1995-2014).

    Under present-day conditions, we found that the GCMs EnsMedian simulates less intense rainfall events. All the models

exhibit too frequent rainfall and longer wet spells over the Guinea Coast, compared to the observations. These models show a wide range of FRQW spatial correlation with the different observations, which makes this variable uncertain in the models, as indicated by its poorest skill score (0.26 for the EnsMedian), compared to the other variables. On the other hand, the R20mm is better represented in the models, according to the distribution of the PCC and TSS values in the GCMs. The EnsMedian values for these metrics are 0.7 and 0.6, respectively. For the other variables, the EnsMedian PCC (TSS) values are between 0.5

and 0.6 (0.4 and 0.5). Those results indicate that despite the good performance of the models for some aspects of the rainfall characteristics, there is still a need to continue the effort toward an improvement of the GCMs to better represent the mean state of the West African hydroclimate. This would lead to a more reliable use of climate models for climate services over West Africa, in order to implement better mitigation and adaptation strategies to climate change.

    Under the SSP5-8.5 scenario, changes in the near-term, mid-term and long-term (2080-2099) periods are evaluated, relative

to the present-day climate conditions. Consistent with previous studies (Dosio et al., 2021; Wainwright et al., 2021), results of the average of the mean changes in the extreme indices over the Guinea Coast indicate an intensification of the daily rainfall, together with a decrease in the frequency of wet days and the duration of wet spells. The number of days with precipitation

exceeding $20\,\mathrm{mm}$ is increased, as well as the rainfall amount during the very wet days and the maximum rainfall amount over five consecutive days.

The area average of the extremes indices over Guinea Coast indicates an intensification of extreme conditions in the projections, with a gradual increase in magnitude over the three future periods. However, it is important to note that the spatial distribution of the changes is not uniform over the region, and the robustness of the changes at a grid point varies among different indices and the future period considered. In general, internal variability is large enough to mask the forced signal during the first decades of the projection over Guinea Coast. In the mid-term changes, only a few areas in the western Guinea Coast

exhibit a robust change in SDII, R20mm and RX5day. In the long-term projections, the R95p projections remain uncertain over Guinea Coast, while robust change signal with a uniform sign can be found for the other extreme indices. These conclusions are in overall agreement with Monerie et al. (2017), who estimated the different numbers of models ensemble members needed to resolve the forced signal in the projections over Sahel. This kind of study needs to be performed over Guinea Coast. It is, however, beyond the aim of our current study, which is rather focused on teleconnection processes.

Anomalous warming (cooling) of the eastern equatorial Atlantic in positive (negative) phases of the AEM is associated with above (below) normal values of wet extreme rainfall indices over Guinea Coast in the current climate. Accordingly, our results indicate an in-phase relationship between the different extreme indices of the Guinea Coast and the AEM SST index. However, over Guinea Coast, the multi-model EnsMedian spatial distributions of the extreme rainfall anomalies related to one standard deviation of the AEM under the present-day conditions are not significant for CWD, and are limited to few areas

very close to the coastal line for FRQW. These two indices present the poorest TSS values, in term of performance of the 24 GCMs relative to the six different observations, and compared to the other extreme indices. The index that has the closest pattern to the observations is SDII. The distributions of the TSS values for R20mm, R95p and RX5day indices are similar, with an EnsMedian value close to $0.3$. The six observations are consistent with the sign of the regression patterns of the different extreme indices over Guinea Coast. However, the biases in the GCMs's patterns vary a lot through the 24 models

and the six observations, leading to no consensus on the sign of the bias, except for the FRQW for which there is a consistent underestimation of the responses to AEM.

     Additionally, there is a projected weakening of the AEM under global warming. Subsequently, there is overall a projected decrease in magnitude of the Guinea Coast rainfall extreme responses to this mode of climate variability in the three future periods, relative to the present-day situation. However, the spatial distribution of the changes in the regression patterns is

different from one future period to another period and generally non-robust among the models. These results are also limited by the poor skill of the models in representing the extremes responses to AEM under present-day conditions. Moreover, for a given future period, the sign and the significance of the change are highly heterogeneous in space. As an example, there is a mid-term robust increase in the RX5day response to AEM over the center of Guinea Coast, whereas no significant changes are found over the entire region for the near-term and long-term periods. In general, there is no robust changes in the majority

of the indices for the near-term and mid-term periods. The long-term changes are, however, stronger in magnitude with more robustness over Guinea Coast, according to the sign agreement among the models. Thus, in general, it is questionable whether it makes sense to perform an average of non-robust changes over the Guinea Coast region, for a given variable. Moreover,

differences among the models' responses could be highlighted by grouping them in different categories based on different criteria. According to the poorest representation (in general) of the extreme responses to AEM under present-day conditions, one could choose to separate models that project an increase in the magnitude of the responses, from the models that project a decrease in the extreme responses to AEM. This is not done in our current study. Another solution would be to use few GCMs with 10 or more ensemble members at high spatial resolution, with a good representation of the AEM impact on West Africa. The increase in the spatial resolution will allow the study of regional differences within the Guinea Coast region.

Despite the biases and uncertainties, the area average of the changes over Guinea Coast is more robust, and gives an overall tendency. However, although there is an overall increase in the total variance of most of the extreme indices over the Guinea Coast, the contribution of the AEM to explain the variance of these extreme indices is reduced. This reduction is clear if the long-term fraction of explained variance is compared to is values in the present-day situation. This result suggests a weaker role of the AEM in driving the extreme rainfall variability over the Guinea Coast in a future scenario of high emission of greenhouse gases. Therefore, there is a need to identify other oceanic and atmospheric modes of variability or other processes that explain the future changes in the extremes' variability over the Guinea Coast. These processes should compensate for the decreasing role of the AEM. For instance, Akinsanola et al. (2020) argued that the changes in the circulation (the dynamics) should contribute less to the increase in the rainfall variability over West Africa, while the local thermodynamics should be the dominant factor of these changes.

Our conclusions are based on 24 CMIP6 GCMs that have clear biases and whose resolution is too coarse to represent well some important processes controlling extremes and their changes. It would thus be useful to reevaluate our results with RCMS or even convection permitted models. The processes that drive the biases in the mean state and teleconnection patterns also need to be better understood and better represented in order to gain more reliability on the projected changes in the rainfall extremes over the Guinea Coast. Additional insights on the impact of the internal variability on the projected changes over the Guinea Coast could also be gained by using more GCMs or more ensemble members within one GCM, to assess the impact of the internal climate variability on the projected changes over Guinea Coast.

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

*Author contributions.*   KW, TF and HG conceptualized the paper. KW performed the analyses, prepared the figures and wrote the manuscript based on the comments and suggestions from TF and HG.

*Competing interests.*   The contact author has declared that neither they nor their co-authors have any competing interests

*Acknowledgements.*   We acknowledge the World Climate Research Programme, which, through its Working Group on Coupled Modelling, coordinated and promoted CMIP6. We thank the climate modelling groups for producing and making available their model output, the Earth

System Grid Federation (ESGF) for archiving the data and providing access, and the multiple funding agencies who support CMIP6 and ESGF. Computational resources have been provided by the supercomputing facilities of the Université catholique de Louvain (CISM/U-CLouvain) and the Consortium des Equipements de Calcul Intensif en Fédération Wallonie Bruxelles (CECI) funded by the F.R.S.-FNRS under convention 2.5020.11. HG is research director with the Fonds de la Recherche Scientifique – FNRS (F.R.S.-FNRS). We also thank the two anonymous reviewers and Paul-Arthur Monerie for their constructive comments. Many thanks to the Editor, Peter Knippertz, for his advice and encouragement, and to the Editorial Support Team. A last thanks to Steve Delhaye and Sidiki Sanogo for the exciting discussions we had.