# Peer review of "Future changes in the mean and variability of extreme rainfall indices over the Guinea Coast and role of the Atlantic equatorial mode"

_Weather and Climate Dynamics, 2022_

## Referee Comment (RC1)

Review for "Future changes in the mean and variability of extreme rainfall indices over the Guinea Coast and role of the Atlantic equatorial mode" by Worou et al.

**Overview**

The present manuscript investigates the current and future characteristics of West African rainfall according to an ensemble of 24 CMIP6 GCM models and further relates them to the development of the Atlantic equatorial mode (AEM), which has been a driver of rainfall over the Guinea Coast region. The authors use 12 ETCCDI rainfall indices to compare the nature of present-day rainfall with gridded observation data, i.e., CHIRPS, as well as with the rainfall behaviour in three future periods (near-term, mid-term, long-term) under the strongest SSP5 scenario (SSP5-8.5). By regressing CHIRPS rainfall indices to the AEM index, a reference is created in order to assess and quantify changes in the relationship between future extreme rainfall and the variability of AEM over the Guinea coast. The main results indicate increasingly less frequent but more intense rainfall over a gradually shorter period in the future. Furthermore, the AEM is suggested to have a diminished influence on future extreme rainfall over the Guinea coast region due to a decrease of AEM variability.

An increased knowledge about climate extremes over West Africa, especially that of rainfall, is of high importance for the assessment of current and future risk and subsequent development of risk mitigation strategies, action plans and decision making. In that regard, this study has the potential to be of high relevance for this part of the scientific community. However, as will be pointed out in more detail in the comments below, my biggest concern is the usage of CHIRPS for a study revolving around rainfall extremes, which potentially require a major revision. Furthermore, the study becomes diluted by the high number of rainfall-related indices, some of which are not particularly necessary to include in my opinion. Therefore, the presentation of the results somewhat suffers from an overload of numbers and a lack of structure. Overall, the topic of the manuscript is within the scope of WCD. Being non-native in English, language appears fine to me with only minor corrections to perform.

**General comments/questions**

- As mentioned above, by biggest concern is related to CHIRPS being the reference in a study about extreme rainfall. While it excels at interannual timescales over Africa (e.g., Camberlin et al., 2019), it struggles with the representation of rainfall in the extreme spectrum by showing substantial underestimation tendencies at a daily timescale (e.g., Sanogo et al., 2021; Ageet et al., 2022). Therefore, I do not think that relying on CHIRPS alone is sufficient to address the research question of this manuscript. Based on the outcomes of recent validation studies, I suggest to include the exercise with either GPCC-FDD (Becker et al., 2013) or one of the GPM products GsMAP or IMERG (Kubota et al., 2007; Huffman et al., 2015). This will allow to make at least basic statements about the uncertainty that stems from the observational data themselves.

- How did the authors choose the study periods? I was wondering why the long-term future was selected such that it exhibits a 20-year gap to the mid-term future, while the latter directly follows up to the near-term future. Can the authors elaborate on that?

- The study somewhat suffers from the plethora of indices and/or the lack of a structured presentation of their results. For instance, is it necessary to include both the 95th and 99th percentile (same for R10mm and R20mm)? Is it worth to include PRCPTOT when SDII is basically the same but defined as a rate? Regarding the structure, with this many indices, it can become difficult for readers not familiar with these to keep track of the arguments being made and being confronted with too big panels. I understand that the current structure follows a logical build-up going from the current situation to the perspective in the future followed by the link to AEM. However, the authors may consider to form groups, e.g., extreme precipitation indices (R99p, R20mm, RX1day, …) and frequency (R99pf, CDD, CWD, …), and analyze their current and future characteristics in their own subsection. I believe that this will

particularly improve the readability of the tables, which currently mixes indices with different units together.

- The introduction is too long, which inhibits a sharper formulation of the research question. Looking at each paragraph, the following topics are addressed:

  1. Socio-economic impacts of extreme rainfall
  2. IPPC AR6
  3. Recent trends in AMJ daily rainfall in Guinea coast region
  4. Recent trends in JAS daily rainfall in Guinea coast region
  5. 3.) and 4.) for specific coastal areas
  6. CMIP6 models behaviour of current rainfall characteristics over Africa
  7. GCM behaviour of future rainfall characteristics over Africa
  8. RCM behaviour of future rainfall variability on daily and seasonal timescale
  9. MCSs, AEM, SST
  10. Historical and current relationship between AEM and Guinea coast rainfall
  11. Goal of study

  Some of the paragraphs, e.g., 3-5, can be easily merged and stripped-down to the fundamental information. I advise the authors to revise and shorten the introduction. Also, I believe that AEM should be introduced earlier to facilitate a better build-up to the research topic.

- The beginning of summary and conclusion section lacks a systematic and brief recapitulation of the motivation and research question of the study, the data and methods used and a point-by-point summary of the key results. Also, some discussion was integrated in the result section, which should rather be shifted to the this section (see specific comments). Overall, I am somewhat missing the link of the presented results with existing literature, i.e., a proper discussion, and how they integrate, complement, or disagree with them. As mentioned, a part of it can be just shifted from the result section.

**Specific comments/questions**

L28: "…the exposure to river flooding events is expected to increase 4.6, 8, and 8.6 times more than without climate change". Are these numbers related to the 1.5°C, 2.4°C and 3.5°C scenarios further down the sentence? It didn't get quite clear from the structure of the sentence.

L32: "…for the safety of young people". Maybe better "future generations"?

L49: "… which represents up to 4% of the seasonal daily mean rainfall". Do the authors mean that the JAS mean daily rainfall has decreased by 4%?

L67: "Regional Climate Models (RCMs) forced with CMIP5 GCMs outputs…". Are they related to CORDEX Africa?

L107: "a mode characterized by anomalous warming and cooling in the eastern equatorial Atlantic basin". It should be additionally mentioned that the phenomenon refers to an interannual variability.

L141-143: See general comment on the choice of the periods.

L150-151: "Only one realization of each simulation is considered for each GCM". Based on what criterion did the authors decide on the ensemble member per GCM?

Table 1: The meaning of the metadata IDs (r, i, p, f) should be explained in the caption or in the text. Why do the historical members differ from SSP5-8.5 members for some models?

L153: See general comment on CHIRPS.

L161-162: "This study is focused on the July-September season (JAS), when the Guinea Coast rainfall covariability with the AEM is at its maximum in the current climate models". I believe that this covariability in JAS was not mentioned in the introduction!? For the Guinea coast region, JAS largely falls within the little dry season over the Guinea coast region, thus outside of the two rainy seasons. Have the authors also investigated April-June (AMJ) and/or September-November (SON) when MCSs are more prevalent in the coastal region?

If so, I think it would be worth a few sentences (e.g., in the conclusion) on how future extreme rainfall develops in these seasons.

L162-165: This text snippet looks like it could also be shifted to the introduction.

L167: Have the authors tested the sensitivity of R10mm and R20mm on the choice of the resolution? I would think that different resolutions lead to different outcomes with indices where the threshold is an absolute value (e.g., 1mm for a wet day).

L175: What remapping procedure (bilinear, bicubic, …) is used?

Table 2: Mention again in the caption that the indices are based on ETCCDI. Is PR95/PR99 calculated from wet days only or from all timesteps?

L222: I think the TSS deserves a bit more explanation here about which combination of measures it attempts to create a skill score with (i.e., correlation coefficient and standard deviation). It is worth noting that these components can be weighted differently. For instance, deviations in the standard deviation can be penalized harsher than the correlation coefficient. I believe however that the standard formulation of the Taylor skill score is used here.

L250: "… and 250 the percentage of bias reaches 0.98%, against −1.28% for RX1day". This is not much, is it?

L251-252: "Alongside, the wet days and extremely wet days (R95p and R99p) statistics over the entire Guinea Coast show positive biases that represent 14.66% and 24.10% of the observations". This potentially falls back again to the issues of CHIRPS and the question, whether this could rather be due to the deficiencies of CHIRPS to resolve extremes.

Table 3: How robust are these numbers? How much variance do they exhibit across the members? The authors may consider expressing these numbers rather in a box-whisker plot to account for the uncertainty of the ensemble.

L281-282: "The spatial distribution of the mean changes in R95p, R10mm and R20mm in our study are different from the patterns obtained in the RCM-CMIP5 projections (Akinsanola and Zhou, 2019)". In what way? Please elaborate on that in more detail.

Figure 3: "… diagonal bars…". Better use "hatching".

L290: "The changes in the near-term period relative to the present-day conditions are not clear for the majority of the indices". How is this seen from Fig. 4?

L291: "The mid-term and long-term changes indicate a clear increase in mean and standard deviation…". Again the question from the general comments about how much of an impact on the increase the 20-year jump has from the mid-term to long-term future period!?

Figure 4: Which significance test was used?

L320: "… there is overall a good spatial distribution of the SDII and PRCPTOT anomalies over West Africa". Do the authors mean a "good agreement"?

Figure 5: While the structures of the indices pretty much look alike for EnsMean, the spatial variability for CHIRPS is visibly higher in Fig. A2. Can the authors elaborate on the sources of these differences?

Table 4: This is again a comparison with CHIRPS, correct? Then it should be mentioned in the caption as such.

L363-375: This belongs in the conclusion section in my view.

Figure 7: Please explain in more detail in the text on what you computed here with respect to one standard deviation of the AEM. You varied AEM by one standard deviation and quantified the difference?

L439: One "a" too many.

Table 5: "… of the proportion of the variance explained by the AEM". What exactly does that mean? Are the median values of the indices potentially calculated from different sample sizes?

Summary: See also general comment on the summary and conclusion section. It does not get clear what the source of the rainfall indices are, which SSP scenario was taken, that the link of extreme rainfall to AEM in current and future period is investigated, etc. This should be more carefully and meticulously summarized right in the beginning.

**References**

Ageet, S., Fink, A. H., Maranan, M., Diem, J. E., Hartter, J., Ssali, A. L., & Ayabagabo, P. (2022). Validation of Satellite Rainfall Estimates over Equatorial East Africa. Journal of Hydrometeorology, 23(2), 129-151.

Becker, A., Finger, P., Meyer-Christoffer, A., Rudolf, B., Schamm, K., Schneider, U., & Ziese, M. (2013). A description of the global land-surface precipitation data products of the Global Precipitation Climatology Centre with sample applications including centennial (trend) analysis from 1901–present. Earth System Science Data, 5(1), 71-99.

Huffman, G. J., Bolvin, D. T., Braithwaite, D., Hsu, K., Joyce, R., Xie, P., & Yoo, S. H. (2015). NASA global precipitation measurement (GPM) integrated multi-satellite retrievals for GPM (IMERG). Algorithm Theoretical Basis Document (ATBD) Version, 4(26).

Kubota, T., Shige, S., Hashizume, H., Aonashi, K., Takahashi, N., Seto, S., ... & Okamoto, K. I. (2007). Global precipitation map using satellite-borne microwave radiometers by the GSMaP project: Production and validation. IEEE Transactions on Geoscience and Remote Sensing, 45(7), 2259-2275.

Sanogo, S., Peyrillé, P., Roehrig, R., Guichard, F., & Ouedraogo, O. (2022). Extreme Precipitating Events in Satellite and Rain Gauge Products over the Sahel. Journal of Climate, 35(6), 1915-1938.

---

## Referee Comment (RC2)

**Review: "Future changes in the mean and variability of extreme rainfall indices over the Guinea Coast and role of the Atlantic equatorial mode"**

Authors: Koffi Worou, Thierry Fichefet, and Hugues Goosse

This study utilizes CMIP6 future climate projections to understand how Guinean Coast extreme rainfall events are likely to change in the future under the SSP5-8.5 scenario. More specifically, this study focuses on the relationship between the Atlantic Equatorial Mode (AEM) and extreme Guinean Coast rainfall. Results indicate that the extreme rainfall responses to the AEM appears to be captured by the ensemble mean of 24 different CMIP6 models for the present day, though there are biases especially in terms of magnitudes of the response. In the future, they find that the decreased variability of the AEM leads to a reduced magnitude of the rainfall extreme responses over the Guinean Coast. Thus, while there is an overall projected increase in the variability of most extreme rainfall indices, the contribution via the AEM to this variability weakens in the future warmer climate.

Overall, the paper is well written and will be a welcome addition to the scientific community. Before acceptance, I have a few issues detailed below that the authors need to think about and address.

**Major Comments:**
- Missing from this manuscript is analysis and/or explicit reference to past studies that have evaluated the ability of the CMIP6 models to realistically represent the AEM in terms of the SSTAs. I think the authors may have done this analysis in their prior work (Worou et al. 2022), but the outcome is not explicitly mentioned here in this manuscript. This dawned on me by the time I got to lines 359 – 360 as I started to ponder whether the poor skill scores were just an artifact that the coupled GCMs cannot replicate this mode of variability very well, so of course the relationship between the AEM and the extreme rainfall would not be well captured.
- The authors select CHIRPS daily rainfall data to evaluate and validate the present day CMIP6 results against. That being said there is no evaluation regarding the uncertainty in the observations that you are trying to match up the CMIP6 models against, so it is unclear what biases CHIRPS has in respects to extreme rainfall. I would suggest drawing in additional daily rainfall datasets for this comparison, maybe ARC2 and/or TAMSAT. Both of these datasets extend back to 1983 which would give the authors close to similar temporal coverage to CHIRPS. Doing so would allow the authors to comment on the extent to which the differences between the observations and CMIP6 models is due to the model compared to the uncertainty in the observations. Similarly, you could evaluate multiple SST datasets, though this is less of an issue compared to the rainfall field in my opinion.
- Around line 159: I didn't pick up on this subtlety at first, but realized later on in the manuscript that the authors are evaluating the SSTs for the AEM at monthly timescales, yet they are evaluating rainfall presumably at daily timescales for extreme rainfall events. I guess this may be appropriate, though I wonder if it would be more appropriate to evaluate SSTs at sub-monthly timescales instead (weekly?), as presumably there are sub-monthly variations in equatorial Atlantic SSTs that could be important. Anyhow, to avoid this confusion I suggest the authors explicitly motivate in the text why they choose to evaluate SSTs at the monthly scale to help clarify this decision for the reader.
- I'm unclear why the authors chose to detrend the SST data. I would think they would want to keep it in raw form because they calculate their extreme indices for each year and then average the yearly totals up over the 20 years evaluated over for each time slice if I understand correctly. This decision needs to be explained better in the manuscript. Also – with all of this data manipulation (detrending, normalizing, standardizing), it would be beneficial if the authors would compare these

among the models and observations, so the reader can understand what is being removed/changed, and how alike/dislike are the various models exactly are.

- I know there is a lot of information to convey in a confined article, but I really think it would be useful here in this study to not only analyze results from the ensemble means, but also evaluate individual models to identify which produce more realistic distributions/frequency/intensities of extreme rainfall events. This information could be used to eliminate inclusion of specific models that are judged to perform "poorly" for a given index (or overall), and thus could potentially increase the accuracy/realism of your ensemble mean by eliminating them from consideration before formulating your ensemble mean.

  An example of what I mean involves section 3.2 (Fig. 1 & Table 2). You could evaluate the individual models here and report on whether most models are outliers, or whether most models are close to the ensemble mean with a few outliers on each side. Knowing this could really strengthen the results and the reader's confidence in the ensemble mean. While I understand it is unfeasible to show Fig. 1 for all 24 models, but maybe you could consider expanding/reimagining Table 3 to include info for all of the models as well as the ensemble mean and organize the individual models from those that perform the best to the worst for your selected indices.

  Likewise, you could do the same for Tables 4 and 5, expanding them to include individual models ranked from best to worst to comment on their relationship to the ensemble mean results already shown.

**Minor Comments:**
- Line 145: I presume you are using daily SST and rainfall data, correct? Or is it 3-hourly? Suggest you explicitly mention this in the text here for better clarity.
- Lines 165 – 168: Would be helpful to include a figure to orient the reader here, showing the countries and a box of the analysis region you defined in the text.
- Table 1: Include information on the spatial resolution for each model evaluated so that information is conveyed to the reader.
- Line 168: I am confused why you are selecting 2.8° resolution for use here. Maybe it is due to the coarsest resolution of the GCMs? Suggest motivating this choice better here at first mention (adding the spatial resolutions for each model to Table 1 would help here too). Actually – I see you have done this later on line 175. I'd suggest moving it up to this line to avoid confusion.
- Section 2.3.1: Motivate why you chose 1° resolution here. I suspect the CMIP6 models are coarser, so how exactly did you interpolate to a higher resolution?
- Line 175-176: Do you mean "….averaged over the JAS season **for each year**"? It would also be informative if the authors could calculate and report on the standard deviation over each time slice interval you evaluate over to see how its variability is changing.
- Line 188 – 190: "performs better" in terms of what exactly? Can you clarify what you mean here better? Also – do you still intend to evaluate individual models to identify which produce more realistic distributions/frequency/intensities of extreme rainfall events? This would be important to include I would think. Furthermore, this information could be used to eliminate inclusion of specific models that perform "poorly" for a given index, and thus could potentially increase the accuracy/realism of your ensemble mean by eliminating them from consideration.
- Figure 2 – again how does this change when a different target other than CHIRPS is used? If it is the same, you could just comment on it in the text without adding additional figures. If it is

different, it may be useful to include/expand a figure showing the changes if a different target is used.

- Line 460: Suggest expanding this to include discussion in terms of the uncertainty that exists in the observations by using more than 1 rainfall dataset to evaluate the CMIP6 models against (see prior comment earlier).
- Lines 464 – 465: Comment on how the ensemble mean relates to the spread of the individual members that are used to determine the ensemble mean. Are most models close to the ensemble mean, or are most outliers and they average out to the mean?
- Lines 483 – 485: So how well can we expect the CMIP6 coupled models to replicate the AEM, and what are the implications for this on your findings here?
- On all spatial map figures in the manuscript it would be helpful if the country outlines were included in each panel.

---

## Author Comment (AC2)

**Reply to reviewer 1**

We would like to thank reviewer 1 for his constructive comments and suggestions that will help to improve our manuscript. Our answers to the reviewer's comments are written in purple, and the comments of the reviewer are in black.

Review for "Future changes in the mean and variability of extreme rainfall indices over the Guinea Coast and role of the Atlantic equatorial mode" by Worou et al.

**Overview**

The present manuscript investigates the current and future characteristics of West African rainfall according to an ensemble of 24 CMIP6 GCM models and further relates them to the development of the Atlantic equatorial mode (AEM), which has been a driver of rainfall over the Guinea Coast region. The authors use 12 ETCCDI rainfall indices to compare the nature of present-day rainfall with gridded observation data, i.e., CHIRPS, as well as with the rainfall behaviour in three future periods (near- term, mid-term, long-term) under the strongest SSP5 scenario (SSP5-8.5). By regressing CHIRPS rainfall indices to the AEM index, a reference is created in order to assess and quantify changes in the relationship between future extreme rainfall and the variability of AEM over the Guinea coast. The main results indicate increasingly less frequent but more intense rainfall over a gradually shorter period in the future. Furthermore, the AEM is suggested to have a diminished influence on future extreme rainfall over the Guinea coast region due to a decrease of AEM variability.

An increased knowledge about climate extremes over West Africa, especially that of rainfall, is of high importance for the assessment of current and future risk and subsequent development of risk mitigation strategies, action plans and decision making. In that regard, this study has the potential to be of high relevance for this part of the scientific community. However, as will be pointed out in more detail in the comments below, my biggest concern is the usage of CHIRPS for a study revolving around rainfall extremes, which potentially require a major revision. Furthermore, the study becomes diluted by the high number of rainfall-related indices, some of which are not particularly necessary to include in my opinion. Therefore, the presentation of the results somewhat suffers from an overload of numbers and a lack of structure. Overall, the topic of the manuscript is within the scope of WCD. Being non-native in English, language appears fine to me with only minor corrections to perform.

**General comments/questions**

- As mentioned above, by biggest concern is related to CHIRPS being the reference in a study about extreme rainfall. While it excels at interannual timescales over Africa (e.g., Camberlin et al., 2019), it struggles with the representation of rainfall in the extreme spectrum by showing substantial underestimation tendencies at a daily timescale (e.g., Sanogo et al., 2021; Ageet et al., 2022). Therefore, I do not think that relying on CHIRPS alone is sufficient to address the research question of this manuscript. Based on the outcomes of recent validation studies, I suggest to include the exercise with either GPCC-FDD (Becker et al., 2013) or one of the GPM products GsMAP or IMERG (Kubota et al., 2007; Huffman et al., 2015). This will allow to make at least basic statements about the uncertainty that stems from the observational data themselves.

  Thank you for this constructive comment. We will include five additional observed rainfall datasets in our analyses: ARCv2, PERSIANN,  REGEN, TAMSAT and GPCC_FDD_v2022. We do not consider GsMAP and IMERG datasets in the revised manuscript, as they do not cover our period of study (1995-2014). However, we consider them in Fig. R1.1, to compare their annual cycle characteristics to the other

observations. This figure will be added in the supplementary material, and it shows that the annual cycles of the extreme indices in GsMAP and IMERG are consistent with the ones of the other datasets.

[Figure]

*Figure R1. 1 Annual cycle of the (a) total wet day precipitation (PRCPTOT), (b) very wet days precipitation (sum of the daily rainfall over days when the rainfall exceeds the 95th percentile), and (c) contribution of the total monthly rainfall to the total annual rainfall, for nine different observational datasets. The period considered for each dataset is displayed in the legend. The ensemble median of the observations is indicated by the black curve. The grey shading shows the 10th to 90th percentile range of the observations.*

- How did the authors choose the study periods? I was wondering why the long-term future wasselected such that it exhibits a 20-year gap to the mid-term future, while the latter directly follows up to the near-term future. Can the authors elaborate on that?

We choose the different periods of the future according to some defined periods in the IPCC AR6, more specifically, Table SPM.1 (page SPM-17 of the summary for Policymakers, in the IPCC AR6 WGI). We will add this information in the revised version of the article.

These periods can be easily found in the IPCC Working Group I Interactive Atlas ( https://interactive-atlas.ipcc.ch/regional-information#eyJ0eXBlIjoiQVRMQVMiLCJjb21tb25zIjp7ImxhdCI6OTc3MiwibG5nIjo0MDA2OTIsInpvb20iOjQsInByb2oiOiJFUFNHOjU0MDMwIiwibW9kZSI6ImNvbXBsZXRX2F0bGFzIn0sInByaW1hcnkiOnsic2NlbmFyaW8iOiJzc3A1ODUiLCJwZXJpb2QiOiJmYXIiLCJzZWFzb24iOiJKdW5BdWciLCJkYXRhc2V0IjoiQ01JUDYiLCJ2YXJpYWJsZSI6IJ4NWRheSIsInZhbHVlVHlwZSI6IlJFTEFUSVZFX0FOT01BTFkiLCJoYXRjaGluZyI6INJTVBMRSIsInJlZ2lvblNldCI6ImFyNiIsImJhc2VsaW5lIjoiQVI2IiwicmVnaW9uc1NlbGVjdGVkIjpbXX0sInBsb3QiOnsiYWN0aXZlVGFiIjoicGx1bWUiLCJtYXNrIjoibm9uZSIsInNjYXR0ZXJTWMFnIjpudWxsLCJjYXRlZ29yaWVzYWVzaGcii6bnVsbCwic2hvd2luZyI6ZmFsc2V9fQ== ) :

[Figure]

*Figure R1. 2 An example of the defined periods in the IPCC AR6. The near-term, medium term and long-term futures correspond to 2021-2040, 2041-2060 and 2081-2100 periods, respectively.*

- The study somewhat suffers from the plethora of indices and/or the lack of a structured presentation of their results. For instance, is it necessary to include both the 95[th] and 99[th] percentile (same for R10mm and R20mm)? Is it worth to include PRCPTOT when SDII is basically the same but defined as a rate? Regarding the structure, with these many indices, it can become difficult for readers not familiar with these to keep track of the arguments being made and being confronted with too big panels. I understand that the current structure followsa logical build-up going from the current situation to the perspective in the future followed by the link to AEM. However, the authors may consider to form groups, e.g., extreme precipitation indices (R99p, R20mm, RX1day, …) and frequency (R99pf, CDD, CWD, …), and analyze their current and future characteristics in their own subsection. I believe that this will particularly improve the readability of the tables, which currently mixes indices with differentunits together.

Thank you for this comment. We will reduce the number of extreme precipitation indices in the revised version of the article. We will consider six indices, instead of twelve:

1. the SDII (simple daily intensity index), which describes the intensity of wet rainfall events;
2. the R95p (very wet day), which describes the intensity of rainfall events exceeding the 95[th] percentile;
3. the CWD (consecutive wet days), which describes the maximum duration of a wet event;
4. the RX5day (maximum 5 days precipitation), which is the intensity of an event over a duration of five days;
5. the FRQW (frequency of wet days), which describes the frequency of wet events;
6. R20mm (very heavy precipitation days), which is a measure of the frequency of rainfall events exceeding 20 mm, and which could have a high socio-economic impact.

- The introduction is too long, which inhibits a sharper formulation of the research question.Looking at each paragraph, the following topics are addressed:

1. Socio-economic impacts of extreme rainfall
2. IPPC AR6
3. Recent trends in AMJ daily rainfall in Guinea coast region
4. Recent trends in JAS daily rainfall in Guinea coast region
5. 3.) and 4.) for specific coastal areas
6. CMIP6 models behaviour of current rainfall characteristics over Africa
7. GCM behaviour of future rainfall characteristics over Africa

8. RCM behaviour of future rainfall variability on daily and seasonal timescale
9. MCSs, AEM, SST
10. Historical and current relationship between AEM and Guinea coast rainfall
11. Goal of study

Some of the paragraphs, e.g., 3-5, can be easily merged and stripped-down to the fundamental information. I advise the authors to revise and shorten the introduction. Also, Ibelieve that AEM should be introduced earlier to facilitate a better build-up to the research topic.

*We will rewrite the introduction according to the suggestions. The AEM will be introduced earlier, the behavior of the observed extreme rainfall events over the Guinea Coast will be summarized, as well as their representations in the climate models. Overall, the introduction will be shortened.*

- **The beginning of summary and conclusion** section lacks a systematic and brief recapitulation of the motivation and research question of the study, the data and methods used and a point-by-point summary of the key results. Also, some discussion was integrated in the result section, which should rather be shifted to the this section (see specific comments). Overall, I am somewhat missing the link of the presented results with existing literature, i.e., a proper discussion, and how they integrate, complement, or disagree with them. As mentioned, a part of it can be just shifted from the result section.

*We will provide a systematic and brief recapitulation of the motivation and research question, the data and methods used as well as a summary of the main results at the beginning of the summary and conclusion. We will also shift some discussions from the results section to the summary and conclusion section as suggested.*

**Specific comments/questions**

L28: "…the exposure to river flooding events is expected to increase 4.6, 8, and 8.6 times morethan without climate change". Are these numbers related to the 1.5°C, 2.4°C and 3.5°C scenarios further down the sentence? It didn't get quite clear from the structure of the sentence.

*Yes, these numbers are related to the 1.5°C, 2.4°C and 3.5°C scenarios. In the revised version of the article, this sentence will be rewritten as follows:*

*"According to the climate scenarios, for which future global warming is limited to 1.5°C, 2.4°C and 3.5°C respectively, people born in 2021 and leaving in the Sub-Saharan Africa will experience, during their lifetime, 4.6, 8, and 8.6 times more river flooding events than without climate change, respectively (https://myclimatefuture.info/, last access: 11-08-2022). This increase is two to fourfold higher than the flooding events experienced by people in the same area, born in 1960, and highlights the climate urgency in reducing our greenhouse gas emissions for future generations (Thiery et al., 2021)."*

L32: "…for the safety of young people". Maybe better "future generations"?
*Thank you for the suggestion, we will take it into account as discussed in the previous comment.*

L49: "… which represents up to 4% of the seasonal daily mean rainfall". Do the authors meanthat the JAS mean daily rainfall has decreased by 4%?
*Yes, the authors mean a negative trend which represents 4% of the JAS climatology, so a decrease of 4% of the JAS mean daily rainfall. We will rewrite the sentence accordingly.*

L67: "Regional Climate Models (RCMs) forced with CMIP5 GCMs outputs…". Are they related toCORDEX Africa?

The cited article (Akinsanola et al., 2020) stated that the integrations were performed over the West African domain. These integrations are part of the CORDEX project, but it is not clear whether they were performed over the "CORDEX Africa" domain or another domain such as the "Middle East North Africa" domain, or a more reduced domain, West Africa as stated. In the revised version of our manuscript, we will just add that they are part of the CORDEX project, as stated in their article.

L107: "a mode characterized by anomalous warming and cooling in the eastern equatorial Atlantic basin". It should be additionally mentioned that the phenomenon refers to aninterannual variability.

We will mention that the phenomenon occurs on an interannual timescale.

L141-143: See general comment on the choice of the periods.

We will refer to the IPCC AR6 WGI definitions to motivate our choices of the different periods.

L150-151: "Only one realization of each simulation is considered for each GCM". Based on whatcriterion did the authors decide on the ensemble member per GCM?

Many studies have used one ensemble member and have shown that current GCMs can simulate relatively well some aspects of oceanic internal modes of variability in the tropical Atlantic (e.g. Kucharski et al.,2017; Richter et al, 2020; Worou et al.,2022; Crespo et al.,2022; Yang et al.,2022), tropical Pacific (e.g. Cai et al.,2015) and Indian Ocean (e.g. Cai et al.,2013,2018). We rely here on the same approach and use one realization in our study. We simply choose the first one for each model as we assume that all realizations are equivalent for the analyses we performed. However, a further perspective would be to increase the number of realizations for each model, if available, or to restrict the study to a few GCMs having 10 or more members for both historical and future simulations. This would help to understand how internal variability can impact the simulation of specific internal modes of variability such as the Atlantic equatorial mode of variability and its associated patterns. Such an analysis is out of the scope of our current study. We will add this perspective in our revised manuscript.

Table 1: The meaning of the metadata IDs (r, i, p, f) should be explained in the caption or in the text.

Why do the historical members differ from SSP5-8.5 members for some models?
The meaning of the variant-id in CMIP6 metadata will be provided:
- "r" represents the realization index,
- "i" represents the initialization method,
- "p" represents the physics and
- "f" represents the forcing.

In the SSP5-8.5 outputs, the parent_variant_label does not necessarily correspond to the variant_label. We then read thoroughly the metadata in future simulation outputs and associate them to their corresponding parents, from which they were branched. Please, find below a print screen of the metadata in the daily precipitation file from the SSP5-8.5 simulation performed with CESM2. Both the variant_label (r2i1p1f1) and its associated parent (parent_variant_label) are highlighted with the red rectangle. We will add this information in the revised manuscript.

```
            :external_variables = "areacella" ;
            :forcing_index = 1LL ;
            :frequency = "day" ;
            :further_info_url = "https://furtherinfo.es-doc.org/CMIP6.NCAR.CESM2.ssp585.none.r2i1p1f1" ;
            :grid = "native 0.9x1.25 finite volume grid (192x288 latxlon)" ;
            :grid_label = "gn" ;
            :initialization_index = 1LL ;
            :institution = "National Center for Atmospheric Research, Climate and Global Dynamics Laboratory, 1850 Table Mesa Drive, Boulder, CO 80305, USA" ;
            :institution_id = "NCAR" ;
            :license = "CMIP6 model data produced by <The National Center for Atmospheric Research> is licensed under a Creative Commons Attribution-[]ShareAlike 4.0 In
ternational License (https://creativecommons.org/licenses/). Consult https://pcmdi.llnl.gov/CMIP6/TermsOfUse for terms of use governing CMIP6 output, including citation req
uirements and proper acknowledgment. Further information about this data, including some limitations, can be found via the further_info_url (recorded as a global attribute
in this file)[]. The data producers and data providers make no warranty, either express or implied, including, but not limited to, warranties of merchantability and fitness
 for a particular purpose. All liabilities arising from the supply of the information (including any liability arising in negligence) are excluded to the fullest extent per
mitted by law." ;
            :mip_era = "CMIP6" ;
            :model_doi_url = "https://doi.org/10.5065/D67H1H0V" ;
            :nominal_resolution = "100 km" ;
            :parent_activity_id = "CMIP" ;
            :parent_experiment_id = "historical" ;
            :parent_mip_era = "CMIP6" ;
            :parent_source_id = "CESM2" ;
            :parent_time_units = "days since 0001-01-01 00:00:00" ;
            :physics_index = 1LL ;
            :product = "model-output" ;
            :realization_index = 2LL ;
            :realm = "atmos" ;
            :source = "CESM2 (2017): atmosphere: CAM6 (0.9x1.25 finite volume grid; 288 x 192 longitude/latitude; 32 levels; top level 2.25 mb); ocean: POP2 (320x384 lo
ngitude/latitude; 60 levels; top grid cell 0-10 m); sea_ice: CICE5.1 (same grid as ocean); land: CLM5 0.9x1.25 finite volume grid; 288 x 192 longitude/latitude; 32 levels;
top level 2.25 mb); aerosol: MAM4 (0.9x1.25 finite volume grid; 288 x 192 longitude/latitude; 32 levels; top level 2.25 mb); atmoschem: MAM4 (0.9x1.25 finite volume grid; 2
88 x 192 longitude/latitude; 32 levels; top level 2.25 mb); landIce: CISM2.1; ocnBgchem: MARBL (320x384 longitude/latitude; 60 levels; top grid cell 0-10 m)" ;
            :source_id = "CESM2" ;
            :source_type = "AOGCM BGC AER" ;
            :sub_experiment = "none" ;
            :sub_experiment_id = "none" ;
            :table_id = "day" ;
            :tracking_id = "hdl:21.14100/72dc82db-57be-4507-99c5-688ecb69458e" ;
            :variable_id = "pr" ;
            :variant_info = "CMIP6 SSP5-8.5 experiments (2015-2100) with CAM6, interactive land (CLM5), coupled ocean (POP2) with biogeochemistry (MARBL), interactive s
ea ice (CICE5.1), and non-evolving land ice (CISM2.1)" ;
            :variant_label = "r2i1p1f1" ;
            :parent_variant_label = "r1i1p1f1" ;
}
```

L153:    See general comment on CHIRPS.
         We will add five additional datasets for the observed daily rainfall and we will
modify the section accordingly.

L161-162: "This study is focused on the July-September season (JAS), when the Guinea Coast rainfall covariability with the AEM is at its maximum in the current climate models". I believe that this covariability in JAS was not mentioned in the introduction!? For the Guinea coast region, JAS largely falls within the little dry season over the Guinea coast region, thus outside of the two rainy seasons. Have the authors also investigated April-June (AMJ) and/or September-November (SON) when MCSs are more prevalent in the coastal region?

If so, I think it would be worth a few sentences (e.g., in the conclusion) on how futureextreme rainfall develops in these seasons.

The maximum covariability between the AEM and the GCR over JAS was previously mentioned in the Methods and will be shifted to the introduction section.

We don't focus our analysis on the specific contribution of MCSs to the rainfall over the Guinea Coast, as there are different types of rainfall over the region (Maranan et al 2018). Although the MCSs have an important contribution to the GCR during AMJ and SON, the annual cycle of the R95p index (very wet days index) across different observations indicates the highest values during July, August, September and October (Fig. R1.1 (b)). The total wet day precipitation (PRCPTOT) and the contribution of the monthly rainfall to the annual rainfall (ANNPCT) also show their highest values over these months (Fig. R1.1 (a),(c)). We will therefore keep the JAS season as the focus of our study but we will add a short discussion about the prevalence of MCSs in the AMJ and SON seasons over the Guinea Coast, without additional analyses. Besides, the "little dry season" is not pronounced in the annual cycle of the average of the total rainfall (PRCPTOT) over the region that we defined as the Guinea Coast in the present analysis (20°W - 15°E,4°N -10°N) and over the 1995-2014 period, as can be seen in the figure R1.1 (a), and also mentioned by Worou et al. 2020. We will add this information to the revised manuscript.

L162-165: This text snippet looks like it could also be shifted to the introduction.

We will shift this text to the introduction. Thank you for the suggestion.

L167: Have the authors tested the sensitivity of R10mm and R20mm on the choice of the resolution? I would think that different resolutions lead to different outcomes with indiceswhere the threshold is an absolute value (e.g., 1mm for a wet day).

We will do not analyze the R10mm in our revision. Figure R1.3 shows the annual cycle of the 6 selected indices during the 1995-2015 period. Two GCMs (among 24) provide low (CNRM-CM6-1, MPI-ESM1-2-LR) and high resolution (CNRM-CM6-1-HR, MPI-ESM1-2-HR) outputs. The analysis of Fig. R1.3 reveals that the low-resolution version of CNRM-CM6 model simulates more frequent rainfall events exceeding 20mm (R20mm) compared to the high-resolution version, mainly between June and November. The five other indices also have higher magnitudes in the low-resolution version of this model, compared to the high-resolution simulations. This is not the case for the model MPI-ESM-2 which simulates an annual cycle with a similar magnitude in its two versions, for the different indices except the maximum consecutive wet days (CWD) and frequency of wet events (FRQW). In these latter cases, the low-resolution simulation shows higher values compared to the high-resolution values. Therefore, we can not make a general statement about the impact of resolution on the simulation of the R20mm. We can also say that both the low-resolution versions of CNRM-CM6-1 and MPI-ESM1-2 models simulate higher CWD and FRQW, compare to their high-resolution versions. We will add this information in the revised version of the article. We will also test the sensitivity of our results to the changes in the 1mm threshold used to define a wet day

[Figure]

*Figure R1. 3 Annual cycle of the extreme rainfall indices over the Guinea Coast, during the 1995-2014 period. Low and high-resolution outputs are analyzed for CNRM-CM6-1 and MPI-ESM1-2 models. The ensemble median for the 24 GCMs and the 6 observations are also shown.*

L175:  What remapping procedure (bilinear, bicubic, …) is used?

    We used a bilinear remapping procedure with a routine from the climate data operator (CDO). We will add this information to our revised text.

Table 2: Mention again in the caption that the indices are based on ETCCDI. Is PR95/PR99calculated from wet days only or from all timesteps?

    We will add in the table caption that the indices' definitions are provided by the ETCCDI. We will also mention that the PR95 is computed from only wet days timeseries over a considered period, for each month. Moreover, in this new version, we compute each index over each month before considering an average over a season. These modifications will be applied to Table 2. This thus considers the annual cycle of each index, which was not the case in the first version of the article.

L222:   I think the TSS deserves a bit more explanation here about which combination of measures it attempts to create a skill score with (i.e., correlation coefficient and standard deviation). It is worth noting that these components can be weighted differently. For instance, deviations in the standard deviation can be penalized harsher than the correlation coefficient. I believe however that the standard formulation of the Taylor skill score is used here.

    Thank you for having drawn our attention to the formulation of this performance skill. We used a modified version of the standard Taylor skill score, where the terms containing a correlation are to power 2 (instead of 1 in the standard formulation). The aim is to penalize models with low spatial correlation. However, there is an error in our text, where we missed the power 2 in the term $(1 + PCC_0)^2$ in the denominator. This will be corrected in our revised text.

$$TSS = \frac{4(1 + PCC)^2}{\left(\frac{\sigma_{cmip6}}{\sigma_{observation}} + \frac{\sigma_{observation}}{\sigma_{cmip6}}\right)^2 (1 + PCC_0)^2}$$

    Moreover, we will mention in the new text that one can choose to penalize models with low correlations or low spatial variability by weighting differently the terms in the TSS equation.

L250:  "… and 250 the percentage of bias reaches 0.98%, against −1.28% for RX1day". This isnot much, is it?

It is true that these bias percentages are very low. We will change the way we present these results, by simply stating that the biases are low instead of presenting the values.

L251-252: "Alongside, the wet days and extremely wet days (R95p and R99p) statistics over the entire Guinea Coast show positive biases that represent 14.66% and 24.10% of the observations". This potentially falls back again to the issues of CHIRPS and the question, whether this could rather be due to the deficiencies of CHIRPS to resolve extremes.

In the revised version, we will compute the biases of each model relative to each of the six observation datasets. The results of the median values of the biases across all the models and observations will be presented.

Table 3: How robust are these numbers? How much variance do they exhibit across the members?

The authors may consider expressing these numbers rather in a box-whisker plot to account for the uncertainty of the ensemble.

Thank you for this comment. In the revised manuscript, we will present these performance metrics in a box-whisker plot, considering six different observations.

L281-282: "The spatial distribution of the mean changes in R95p, R10mm and R20mm in our study are different from the patterns obtained in the RCM-CMIP5 projections (Akinsanola and Zhou, 2019)". In what way? Please elaborate on that in more detail.

Akinsanola and Zhou (2019) found a robust increase in the mean June-September R10mm and R20mm over large areas in Guinea Coast in the long-term future projections. Our study shows insignificant mean July-September R10mm and R20mm over this region in the long-term future, compared to the present day. Moreover, in the western (eastern) Sahel, there is a projected robust long-term decrease (increase) in the R10mm in our analysis, whereas Akinsanola and Zhou found a robust and uniform decrease in the R10mm over the Sahel. Finally, we found an insignificant increase in the long-term mean of R95p, while Akinsanola and Zhou found a significant increase in R95p over almost all West Africa. We will add these differences in the revised manuscript, for the R20mm and R95p indices.

Figure 3: "… diagonal bars…". Better use "hatching".

We will use "hatching" instead of "diagonal bar". Thank you for the suggestion.

L290: "The changes in the near-term period relative to the present-day conditions are not clear for the majority of the indices". How is this seen from Fig. 4?

We see this from the error bars associated with each change. If for a variable and a specific period in the future, the error bar around the mean change covers negative to positive values, this means that we cannot see a clear increase or a clear decrease in the considered variable. This is the case for the majority of the near-term changes in mean and standard deviation for different indices. In the revised manuscript, this figure will be replaced by a box-whisker plot (Figs. R1.4-R1.5).

[Figure]

*Figure R1. 4 Change (relative to 1995-2014) in the mean JAS extreme rainfall indices averaged over the Guinea Coast.*

[Figure]

*Figure R1. 5 Change (relative to 1995-2014) in the standard deviation of the extreme rainfall indices averages of the Guinea Coast.*

L291:   "The mid-term and long-term changes indicate a clear increase in mean and standard deviation…". Again the question from the general comments about how much of an impacton the increase the 20-year jump has from the mid-term to long-term future period!?

  We could have used two 30 years periods in the last decades of the 21st century, as in Worou et al (2022). This would cancel the gap between the mid-term and long-term future periods. However, the main message of our study would not have changed, which is the reduction in the AEM contribution to the variability of the extreme rainfall events over the Guinea Coast. In the revision, we will add in supplementary material the mean changes in 2061-2079 compared to 1995-2014, and we will plot these changes together with the changes in the main three future periods defined in our manuscript. We can also quantify the changes from the 2061-2079 period to the long-term (2070-2099) period. We will add a comment about the outcomes in the main revised text.

Figure 4:   Which significance test was used?

  We did not make a test of significance in the difference between of the multi-model mean of an index when comparing a future period to the historical period. Rather, we computed the mean difference and we associated an error bound in the estimation of the mean difference (across the different models), at a confidence level of 90%.
In the revised version of the manuscript, we will modify this figure by showing the box plot of the changes in mean and standard deviation, as indicated in Figs. R1.4 and R1.5. We will also apply a Kolmogorov-Smirnov test to the distribution of an index in two different periods, which will allow us to confirm if there is a significant change.

L320:   "… there is overall a good spatial distribution of the SDII and PRCPTOT anomalies overWest Africa". Do the authors mean a "good agreement"?

  Yes, we mean a "good agreement". We will modify the text accordingly. Moreover, we no longer consider PRCPTOT in the newly selected indices for the revised manuscript.

Figure 5:   While the structures of the indices pretty much look alike for EnsMean, the spatial variability for CHIRPS is visibly higher in Fig. A2. Can the authors elaborate on the sourcesof these differences?

  In the revised manuscript, we will compare the spatial distribution of the extreme indices responses to the AEM phases for the 6 different rainfall datasets. This will help to see if the responses are similar in the different data, or if there are differences among them. This will help in the analysis of the differences with the multi-model ensemble mean response. Furthermore, in the revised manuscript, we will consider the multi-model ensemble median, instead of the multi-model ensemble mean, to avoid any impact of outliers.

Table 4:   This is again a comparison with CHIRPS, correct? Then it should be mentioned in thecaption as such.

  Yes, this is a comparison with CHIRPS. In the revised manuscript, this table will be replaced by a figure with different boxplots of the models performances relative to the six rainfall observations. We will adapt the caption accordingly.

L363-375:   This belongs in the conclusion section in my view.

  We will move this point to the conclusion as suggested. Thank you.

Figure 7:   Please explain in more detail in the text on what you computed here with respect to one standard deviation of the AEM. You varied AEM by one standard deviation and quantifiedthe difference?

  Let's consider the AEM index computed from monthly SST anomalies (detrended linearly) over the ATL3 box and averaged over JAS. For the 1995-2014 period, for instance, we obtain a timeseries of 20 timesteps. Then, this index is divided by its standard deviation. When we regress a grid point value of a variable onto the above-described AEM index, we obtain a pattern of anomalies related to one standard deviation of the AEM index (or the standardized AEM index).

  In Figure 7 of the submitted manuscript, we computed the changes in the

regression coefficients of the different indices related to the standardized AEM index. Then, the changes in the regression coefficients are averaged over the Guinea Coast for each model. Figure 7 shows the multi-model ensemble mean changes in the area-averaged regression coefficients and the associated 90% confidence interval. In the revised manuscript, this figure will be replaced by a box-whisker plot, using six indices instead of twelve.

L439:   One "a" too many.
          Thank you for the correction. We will suppress one "a".

Table 5:  "… of the proportion of the variance explained by the AEM". What exactly does that mean?

Are the median values of the indices potentially calculated from different sample sizes?

For each period and for each index (anomaly) averaged over Guinea Coast, we computed the proportion of variance explained by the AEM for each model. We obtain 24 different values from 24 GCMs. Values in Table 5 represent the multi-model median from each of the 24 values, each period and each index. In the revised manuscript, Table 5 will be replaced by a box-whisker plot.

Summary: See also general comment on the summary and conclusion section. It does not get clearwhat the source of the rainfall indices are, which SSP scenario was taken, that the link ofextreme rainfall to AEM in current and future period is investigated, etc. This should be more carefully and meticulously summarized right in the beginning.
          We will improve the summary and conclusion section. We will clearly describe at the beginning of this section the different rainfall datasets used for the observations, the CMIP6 historical simulations and the SSP5-8.5 scenario. Thank you for the comment.

**References**

Ageet, S., Fink, A. H., Maranan, M., Diem, J. E., Hartter, J., Ssali, A. L., & Ayabagabo, P. (2022). Validation of Satellite Rainfall Estimates over Equatorial East Africa. Journal of Hydrometeorology,23(2), 129-151.

Becker, A., Finger, P., Meyer-Christoffer, A., Rudolf, B., Schamm, K., Schneider, U., & Ziese, M. (2013). A description of the global land-surface precipitation data products of the Global Precipitation Climatology Centre with sample applications including centennial (trend) analysis from 1901–present.Earth System Science Data, 5(1), 71-99.

Huffman, G. J., Bolvin, D. T., Braithwaite, D., Hsu, K., Joyce, R., Xie, P., & Yoo, S. H. (2015). NASAglobal precipitation measurement (GPM) integrated multi-satellite retrievals for GPM (IMERG).
Algorithm Theoretical Basis Document (ATBD) Version, 4(26).

Kubota, T., Shige, S., Hashizume, H., Aonashi, K., Takahashi, N., Seto, S., ... & Okamoto, K. I. (2007).Global precipitation map using satellite-borne microwave radiometers by the GSMaP project: Production and validation. IEEE Transactions on Geoscience and Remote Sensing, 45(7), 2259-2275.

Sanogo, S., Peyrillé, P., Roehrig, R., Guichard, F., & Ouedraogo, O. (2022). Extreme Precipitating Events in Satellite and Rain Gauge Products over the Sahel. Journal of Climate, 35(6), 1915-1938.

References:

Maranan, M, Fink, AH, Knippertz, P. Rainfall types over southern West Africa: Objective identification, climatology and synoptic environment. Q J R Meteorol. Soc. 2018; 144: 1628-1648. https://doi.org/10.1002/qj.3345

Worou, K., Goosse, H., Fichefet, T., Guichard, F., Diakhaté, M. : Interannual variability of rainfall in the Guinean Coast region and its links with sea surface temperature changes over the twentieth century for the different seasons. Clim Dyn 55, 449–470 (2020).

https://doi.org/10.1007/s00382-020-05276-5

Kucharski, F. and Joshi, M.K. (2017), Influence of tropical South Atlantic sea-surface temperatures on the Indian summer monsoon in CMIP5 models. Q.J.R. Meteorol. Soc, 143: 1351-1363. https://doi.org/10.1002/qj.3009

Richter, I., Tokinaga, H. An overview of the performance of CMIP6 models in the tropical Atlantic: mean state, variability, and remote impacts. Clim Dyn 55, 2579–2601 (2020). https://doi.org/10.1007/s00382-020-05409-w

Worou, K., Goosse, H., Fichefet, T., and Kucharski, F.: Weakened impact of the Atlantic Niño on the future equatorial Atlantic and Guinea Coast rainfall, Earth Syst. Dynam., 13, 231–249, https://doi.org/10.5194/esd-13-231-2022, 2022.

Crespo, L.R., Prigent, A., Keenlyside, N. et al. Weakening of the Atlantic Niño variability under global warming. Nat. Clim. Chang. 12, 822–827 (2022). https://doi.org/10.1038/s41558-022-01453-y

Yang, Y., Wu, L., Cai, W. et al. Suppressed Atlantic Niño/Niña variability under greenhouse warming. Nat. Clim. Chang. 12, 814–821 (2022). https://doi.org/10.1038/s41558-022-01444-z

Cai, W., Santoso, A., Wang, G. et al. ENSO and greenhouse warming. Nature Clim Change 5, 849–859 (2015). https://doi.org/10.1038/nclimate2743

Cai, W., Zheng, XT., Weller, E. et al. Projected response of the Indian Ocean Dipole to greenhouse warming. Nature Geosci 6, 999–1007 (2013). https://doi.org/10.1038/ngeo2009

Cai, W., Wang, G., Gan, B. et al. Stabilised frequency of extreme positive Indian Ocean Dipole under 1.5 °C warming. Nat Commun 9, 1419 (2018). https://doi.org/10.1038/s41467-018-03789-6

---

## Author Comment (AC3)

**Responses to reviewer 2**

We would like to thank reviewer 2 for his constructive comments and remarks that help to improve our current study. The questions of the reviewer are recalled in black, and our responses are written in purple.

Review: "Future changes in the mean and variability of extreme rainfall indices over the Guinea Coast and role of the Atlantic equatorial mode"

Authors: Koffi Worou, Thierry Fichefet, and Hugues Goosse

This study utilizes CMIP6 future climate projections to understand how Guinean Coast extreme rainfall events are likely to change in the future under the SSP5-8.5 scenario. More specifically, this study focuses on the relationship between the Atlantic Equatorial Mode (AEM) and extreme Guinean Coast rainfall. Results indicate that the extreme rainfall responses to the AEM appears to be captured by the ensemble mean of 24 different CMIP6 models for the present day, though there are biases especially in terms of magnitudes of the response. In the future, they find that the decreased variability of the AEM leads to a reduced magnitude of the rainfall extreme responses over the Guinean Coast. Thus, while there is an overall projected increase in the variability of most extreme rainfall indices, the contribution via the AEM to this variability weakens in the future warmer climate.

Overall, the paper is well written and will be a welcome addition to the scientific community. Before acceptance, I have a few issues detailed below that the authors need to think about and address.

**Major Comments:**

- Missing from this manuscript is analysis and/or explicit reference to past studies that have evaluated the ability of the CMIP6 models to realistically represent the AEM in terms of the SSTAs. I think the authors may have done this analysis in their prior work (Worou et al. 2022), but the outcome is not explicitly mentioned here in this manuscript. This dawned on me by the time I got to lines 359 – 360 as I started to ponder whether the poor skill scores were just an artifact that the coupled GCMs cannot replicate this mode of variability very well, so of course the relationship between the AEM and the extreme rainfall would not be well captured.

  Thank you for this excellent comment. Current GCMs can replicate the SST variability associated with the AEM over the tropical Atlantic Ocean, but they struggle to reproduce the spatial distribution of the observed rainfall response over the tropical Atlantic and the Guinea Coast. This has been highlighted in our prior work (Worou et al. 2022), and we will recall this point in the revised manuscript.

- The authors select CHIRPS daily rainfall data to evaluate and validate the present day CMIP6 results against. That being said there is no evaluation regarding the uncertainty in the observations that you are trying to match up the CMIP6 models against, so it is unclear what biases CHIRPS has in respects to extreme rainfall. I would suggest drawing in additional daily rainfall datasets for this comparison, maybe ARC2 and/or TAMSAT. Both of these datasets extend back to 1983 which would give the authors close to similar temporal coverage to CHIRPS. Doing so would allow the authors to comment on the extent to which the differences between the observations and CMIP6 models is due to the model compared to the uncertainty in the observations. Similarly, you could evaluate multiple SST datasets, though this is less of an issue compared to the rainfall field in my opinion.

  Thank you for the suggested rainfall data. We will add a total of five additional observed rainfall data in the revision of this work. These data are: ARCv2, PERSIANN, REGEN, TAMSAT and GPCC_FDD_v2022. As an example, Fig. R2.1 shows that the model multi-median performs well in the simulation of some characteristics of the rainfall over the Guinea Coast. However, it overestimates the frequency of wet days (FRQW) and the maximum consecutive wet days (CWD). In our revised manuscript, we will assess the performance of the models against the six different observed rainfall datasets. When needed, the multimodel median values across the different observations and the different models will be indicated (for the mean biases map for instance). On the other hand, we won't include additional observed

SST data as we consider that this is not a critical point for out analyses.

[Figure]

*Figure R2. 1 Annual cycle of the extreme rainfall indices over the Guinea Coast (1995 – 2014): median (black and brown curves), 10th - 90th percentile range (gray and brown shades) are indicated in black for 6 observations and in brown for 24 CMIP6 GCMs. Vertical lines indicate the season considered in our study.*

- Around line 159: I didn't pick up on this subtlety at first, but realized later on in the manuscript that the authors are evaluating the SSTs for the AEM at monthly timescales, yet they are evaluating rainfall presumably at daily timescales for extreme rainfall events. I guess this may be appropriate, though I wonder if it would be more appropriate to evaluate SSTs at sub-monthly timescales instead (weekly?), as presumably there are sub-monthly variations in equatorial Atlantic SSTs that could be important. Anyhow, to avoid this confusion I suggest the authors explicitly motivate in the text why they choose to evaluate SSTs at the monthly scale to help clarify this decision for the reader. To clarify the point, on the one hand, we are computing monthly extreme rainfall indices from daily rainfall values. Then, we consider an average over the boreal summer season, from July to September. This means that, at the end, we have seasonal values for the rainfall indices for each year of the considered period. On the other hand, we are interested in the variability of the Atlantic equatorial mode (also called Atlantic Niño), which peaks from the end of spring to the boreal summer and occurs mainly on the interannual timescale. Our study does not address the variability on weekly timescales, though they are also important for sub-seasonal to seasonal coupled atmosphere-ocean processes such as the Madden-Julian Oscillation, which have also a large impact on the West African hydroclimate. We will clarify our choice in the revised manuscript accordingly.
- I'm unclear why the authors chose to detrend the SST data. I would think they would want to keep it in raw form because they calculate their extreme indices for each year and then average the yearly totals up over the 20 years evaluated over for each time slice if I understand correctly. This decision needs to be explained better in the manuscript. Also – with all of this data manipulation (detrending, normalizing, standardizing), it would be beneficial if the authors would compare these

among the models and observations, so the reader can understand what is being removed/changed, and how alike/dislike are the various models exactly are.

There are two main points in our study. First, we look at the mean changes in the extreme variables as well as the changes in their variability. The extreme rainfall indices are kept in their raw seasonal values for each year. Second, we address changes in the variability of the extreme indices around their mean values. We remove any linear drift in the 20-year period timeseries, from each monthly series before averaging over the July-September season. Our aim is not to study the trend in the data, but rather the interannual variability in the extreme indices that is partly explained by the Atlantic equatorial mode. Similarly, the Atlantic equatorial mode positive or negative phases are defined by SST anomalies exceeding at least one standard deviation around the SST mean values in the eastern equatorial Atlantic. Any linear trend (due to global warming for example) needs to be removed to focus on the AEM phenomenon during the 20-year period, since climate change could have an impact on the AEM variability. We propose to include in the online supplementary material, for the present-day period, for each extreme rainfall index, for the AEM SST index, for each model, and for each observation, a figure illustrating the JAS average of the raw indices, as well as the trend of these indices. This will help the reader to have an idea of the behavior of the trend in each index. We will add this clarification in the revised manuscript.

- I know there is a lot of information to convey in a confined article, but I really think it would be useful here in this study to not only analyze results from the ensemble means, but also evaluate individual models to identify which produce more realistic distributions/frequency/intensities of extreme rainfall events. This information could be used to eliminate inclusion of specific models that are judged to perform "poorly" for a given index (or overall), and thus could potentially increase the accuracy/realism of your ensemble mean by eliminating them from consideration before formulating your ensemble mean.

An example of what I mean involves section 3.2 (Fig. 1 & Table 2). You could evaluate the individual models here and report on whether most models are outliers, or whether most models are close to the ensemble mean with a few outliers on each side. Knowing this could really strengthen the results and the reader's confidence in the ensemble mean. While I understand it is unfeasible to show Fig. 1 for all 24 models, but maybe you could consider expanding/reimagining Table 3 to include info for all of the models as well as the ensemble mean and organize the individual models from those that perform the best to the worst for your selected indices.

Likewise, you could do the same for Tables 4 and 5, expanding them to include individual models ranked from best to worst to comment on their relationship to the ensemble mean results already shown.

Thank you for this interesting suggestion. First, we will replace Tables 3, 4 and 5 by a box-whisker plot to highlight the variability among the models in their representation of the different extreme rainfall characteristics. When needed, we will use the multi-model ensemble median instead of the multi-model ensemble mean, which is less impacted by outliers.

We will also show some diagrams of the models' performance relative to the 6 observations: for a given metric, the different models will be on the horizontal axis and the different extreme indices will be on the vertical axis. This was termed the "portrait diagram" in Faye et al. (2021). This will help the reader to see, for each model, the skill in representing the different variables.

Furthermore, a model which poorly represents a particular index does not necessarily struggles in representing another index. This choice of models' stratification in different categories could be easily done for one variable but it will make difficult the presentation of the models' performances for all the variables. We will therefore do this stratification only when studying the teleconnection patterns to the Guinea Coast. In that case, we will simply split the models into two groups: the first group will have a good sign of the extreme rainfall response to the Atlantic equatorial mode phases,

and the second group will have a sign opposite to the observations.

**Minor Comments:**

- Line 145: I presume you are using daily SST and rainfall data, correct? Or is it 3-hourly? Suggest you explicitly mention this in the text here for better clarity.
  We are using monthly SST data and daily rainfall data. We will mention explicitly the temporal resolution of the datasets as suggested. Thank you.

- Lines 165 – 168: Would be helpful to include a figure to orient the reader here, showing the countries and a box of the analysis region you defined in the text.
  We will add countries on a map and a box showing the different regions mentioned.

- Table 1: Include information on the spatial resolution for each model evaluated so that information is conveyed to the reader.
  The spatial resolution of each model will be added in Table 1 of the revised manuscript.

- Line 168: I am confused why you are selecting 2.8° resolution for use here. Maybe it is due to the coarsest resolution of the GCMs? Suggest motivating this choice better here at first mention (adding the spatial resolutions for each model to Table 1 would help here too). Actually – I see you have done this later on line 175.  I'd suggest moving it up to this line to avoid confusion.
  The 2.8° resolution corresponds to the coarser resolution of the 24 models. We will move the information to the right place, as suggested. Thank you.

- Section 2.3.1: Motivate why you chose 1° resolution here.  I suspect the CMIP6 models are coarser, so how exactly did you interpolate to a higher resolution?
  There are models with a low resolution that only have a few grid points within the ATL3 region (20°W-10°E,3°S-3°N). If the choice was made on each model grid, this would imply to use different regions for different models. We therefore use a bilinear interpolation method with the climate data operator routine (CDO) to interpolate all the sea surface temperature datasets to a common grid of 1° of resolution. We will add this information to the revised manuscript. A similar procedure has been applied in Worou et al., (2022) and Kucharski et al., (2017) for instance.

- Line 175-176: Do you mean "….averaged over the JAS season **for each year**"? It would also be informative if the authors could calculate and report on the standard deviation over each time slice interval you evaluate over to see how its variability is changing.
  Thank you for this remark. In the revised manuscript, we will improve the description of the method. In the submitted version, we computed the rainfall indices over each season, for each year. In the revised manuscript, we will compute the rainfall indices for each month (to show the annual cycle). For uniformity, we will keep the monthly values of the extreme indices, and when we need a seasonal index, we will average over the months in the season (e.g. July-August-September) **for each year**. This will also be the case for seasonal SST and rainfall anomalies. Table 2 will also be modified accordingly by reducing the number of variables and by replacing the JAS season by monthly values.
  Moreover, as suggested, we will add a box-whisker plot of the standard deviation of the different seasonal extreme rainfall indices, probably in the supplementary material. We show below the distribution of the changes in the standard deviation (Fig. R2.2):

[Figure]

*Figure R2. 2 Near-term (2021-2040), mid-term (2041-2060) and long-term (2080-2099) mean changes (relative to 1995-2014) in the standard deviation of the timeseries of JAS extreme rainfall indices averaged over the Guinea Coast. Each box-whisker plot represents the changes from 24 different GCMs, in the SSP5-8.5 scenario relative to the historical simulations.*

- Line 188 – 190: "performs better" in terms of what exactly? Can you clarify what you mean here better? Also – do you still intend to evaluate individual models to identify which produce more realistic distributions/frequency/intensities of extreme rainfall events? This would be important to include I would think. Furthermore, this information could be used to eliminate inclusion of specific models that perform "poorly" for a given index, and thus could potentially increase the accuracy/realism of your ensemble mean by eliminating them from consideration.

  By "performs better", we mean that the multimodel ensemble mean values are closer the observations than each individual model. This can be seen in Fig. R2.1, where we show the multimodel ensemble median instead of the multimodel mean. We will modify the text accordingly. We will also add portrait diagrams showing the skills of each model, compared to the different observations.

  Furthermore, we have decided to include a stratification of the models into "poor" and "good" categories in the analysis of the AEM impact of the extreme indices over the Guinea Coast, to have a more specific analysis.

  Finally, we will consider the multimodel ensemble median when needed, instead of the multimodel ensemble mean, as it is more resistant to outliers.

- Figure 2 – again how does this change when a different target other than CHIRPS is used? If it is the same, you could just comment on it in the text without adding additional figures.
  If it is different, it may be useful to include/expand a figure showing the changes if a different target is used.
  Figure 2 of the submitted manuscript will be modified. For each model, the biases will be computed relative to each of the six observed datasets. Then, the new figure that will resume the models biases relative to the different observations  will be the multi-model ensemble median along all the different models' biases computed. The portrait diagram that we will provide will show the biases for each model, averaged over the Guinea Coast, which will give another complement information at an individual model level.
- Line 460: Suggest expanding this to include discussion in terms of the uncertainty that exists in the observations by using more than 1 rainfall dataset to evaluate the CMIP6 models against (see prior comment earlier).
  We will update in the revised manuscript the discussion of uncertainties related to the observations.
- Lines 464 – 465: Comment on how the ensemble mean relates to the spread of the individual members that are used to determine the ensemble mean. Are most models close to the ensemble mean, or are most outliers and they average out to the mean?
  We will provide in the supplement material six different Taylor diagrams relative to the six different indices, to illustrate the spread of the models' spatial distributions compared to the multimodel ensemble median. This information will be taken into account in the revised manuscript.

- Lines 483 – 485: So how well can we expect the CMIP6 coupled models to replicate the AEM, and what are the implications for this on your findings here?
  The CMIP6 models can simulate reasonably  well the SST pattern associated with the AEM over the tropical Atlantic. However, they show some difficulties in simulating the rainfall responses over the tropical Atlantic and the Guinea Coast. To get more insight into the models' performance, we will focus on two groups of models as discussed above, i.e. models with a correct response in the rainfall and models with a wrong response. We think that this is the most adequate way to discuss the models' ability to simulate the AEM-related patterns in the present framework. This will be considered in the revised article.
- On all spatial map figures in the manuscript it would be helpful if the country outlines were included in each panel.
  Thank you for this comment. We will add country contours in the different maps.

References
Worou, K., Goosse, H., Fichefet, T., and Kucharski, F.: Weakened impact of the Atlantic Niño on the future equatorial Atlantic and Guinea Coast rainfall, Earth Syst. Dynam., 13, 231–249, https://doi.org/10.5194/esd-13-231-2022, 2022.
Kucharski, F. and Joshi, M.K. (2017), Influence of tropical South Atlantic sea-surface temperatures on the Indian summer monsoon in CMIP5 models. Q.J.R. Meteorol. Soc, 143: 1351-1363. https://doi.org/10.1002/qj.3009
Faye, A., Akinsanola, A.A. Evaluation of extreme precipitation indices over West Africa in CMIP6 models. Clim Dyn 58, 925–939 (2022). https://doi.org/10.1007/s00382-021-05942-2

---

## Author Comment (AC4)

**Responses to the community comment CC1**

We would like to thank Paul-Arthur Monerie for his valuable comments and suggestions that really help to improve our manuscript. The comments are recalled in black. Our responses to each comment are written in purple.

**Comment on wcd-2022-53**

Paul-Arthur Monerie

Community comment on "Future changes in the mean and variability of extreme rainfall indices over the Guinea Coast and role of the Atlantic equatorial mode" by Koffi Worou et al., Weather Clim. Dynam. Discuss., https://doi.org/10.5194/wcd-2022-53-CC1, 2022

**Review of "Future changes in the mean and variability of the extreme rainfall indices over the Guinea coast and role of the Atlantic equatorial mode" by Koffi Worou, Thierry Fichefet, and Hugues Goosse.**

The authors assess the future evolution of precipitation characteristics (*e.g.*, extreme rainfall events) and the effects of changes in the Atlantic equatorial mode on Guinea coast precipitation variability and on rainfall indices. The topic is relevant for the scientific community and could help understanding better future changes in precipitation over West Africa.

I have several concerns that should be assessed before considering the study for publication.

- The novelty of the work is unclear. All results on changes in precipitation extremes have already been shown in the literature, and the section on the effects of the Atlantic Equatorial mode is not convincing. Please explain better what is the novelty of the study.

  Thank you for this comment. The motivation of this study is based on the fact that, over the 20th century, the Atlantic equatorial mode of variability has been the main oceanic driver of the interannual rainfall variability over the Guinea Coast, as shown in many studies (Giannini et al.,2003; Rodríguez-Fonseca et al.,2011,2015; Losada et al.,2012, Lübbecke et al.,2018; Worou et al.,2020). Other studies pointed out that this oceanic internal mode of variability could also impact extreme rainfall events over West Africa (Diatta et al,2020). An extraordinary high impact rainfall event occurs in Burkina Faso, on the 1st of September 2009, coinciding with below normal sea surface temperature (SST) conditions in the eastern equatorial Atlantic. This motivates us to study future changes in the SST variability in the eastern equatorial Atlantic and their impacts on the occurrence of extreme rainfall events over the Guinea Coast. This approach is new, because it studies a well-known couple ocean-atmosphere phenomenon, the Atlantic equatorial mode, and how future changes in this mode potentially influence changes in extremes. To our knowledge, such a specific analysis of the links between the AEM and extremes for both current and future climates has not been performed before. We will improve our motivation in the revised manuscript, accordingly.

- The introduction is too long, and its structure could be revised, with the information streamlined to provide a summary of the literature. For instance, information from Bichet and Diedhiou (2018), Odoulami and Akinsanola (2017), and Kpanou et al. (2018) is for different seasons (*g.*, AMJ, JAS), for different periods, and different metrics. Information seems also contradictory, with Odoulami and Akinsanola (2017) stating that there is a negative trend in extreme events (relative to the 95th percentile)

over the Guinea coast while Kpanou et al. (2018) state that the number of such events has increased over some of the West African countries (Ivory coast, Togo, and Benin). The authors have used monthly SSTs to compute the AEM index. Please explain how monthly variabilities in SSTs could lead to changes in the variability of rainfall extremes.

We would like first to insist that we computed seasonal SST indices of the Atlantic equatorial mode (AEM) from monthly SST anomalies (linearly detrended) averaged over July, August and September (JAS). The impact of the AEM is evaluated on the seasonal rainfall characteristics.

Moreover, we did not describe the dynamics explaining the link between monthly SST variabilities and rainfall extremes in the submitted version. This will be included in the new version of the article, a few statements about the mechanism. Warm phases of the AEM during the boreal summer lead to strong low-level convergence and rising motion of humid air over the eastern equatorial Atlantic. The land-sea surface pressure gradient weakens, and limits the northward penetration of the monsoon flow toward the Sahel. Humid air is advected by the low-level circulation from the equatorial Atlantic to the coastal areas of West Africa. This provides favorable conditions for an increase in the occurrence of rainfall as well as of extreme rainfall events over the Guinea Coast. The physics behind the connection of the AEM to West Africa and the tropical regions is discussed in detail in Losada et al. (2010); Lübbecke et al.(2018) and Worou et al.(2020,2022), among others. We will modify the manuscript accordingly.

We will also reduce the length of the introduction, and better structure the review of the works on the extreme rainfall events over the Guinea Coast, with focus on the boreal summer season. It is hard to compare Odoulami and Akinsanola (2017) with Kpanou et al (2018), because the former study is focused on June-September (1998-2013), while the latter is on an annual basis (1981-2015).

■ Method: The authors defined an anomaly as robust when 50% of the models present a significant regression coefficient. However, this low value could be eventually obtained by chance and would not show robustness in the results. The authors should revise this threshold, using the two-thirds threshold used for the sign of the EnsMean. Data have been linearly detrended. However, it is shown that anthropogenic aerosols have strong effects on West African precipitation and have driven a part of the 1970s-1980s drought and of the precipitation recovery (*e.g.*, Herman et al., 2020, Hirasawa et al. 2020; Monerie et al. 2022). Removing a linear trend will thus not allow considering the full effect of the anthropogenic activity on West African precipitation. The authors should check the robustness of the results using other methods, such as estimating theforced response using the ensemble mean (see for instance Ting et al. 2009).

Thank you for the suggestions.

We will revise the robustness metric for the change in the regression patterns related to the AEM. We will consider the two-third threshold for the sign-agreement among the 24 models as suggested.

Our aim is not to assess the specific impact of the aerosols on the extreme rainfall trend over the Guinea Coast, which is why we remove any drift from the anomalies computed for each index. We will better motivate the detrending of the data in the method section of the revised manuscript. However, anthropogenic aerosol forcing can have a long-term impact on the climate in many regions of the world. Over the past decades, changes in the anthropogenic aerosol forcing have led to differences in the changes in sea surface temperature between the North Atlantic and the global tropics, which yielded modifications of the hydrological conditions over the Sahel (A Giannini et al., 2013). You showed (Monerie et al.,2022) that the Anthropogenic Aerosol emissions induce a negative trend in the rainfall over West Africa during 1950-1980, and a positive trend afterward. These aerosol emissions have also induced a north-to-south SST gradient trend in the tropical Atlantic, which led to a weakening of the AEM and its impact on the

rainfall over the Guinea Coast. Consequently, we will mention the impact of different forcings on the West African climate, and highlight the role of aerosols in our introduction.

For the last point of the comment, we will compute a signal-to-noise ratio to compare the forced response to the internal variability. This additional diagnostic will be added to the maps of future mean changes in the extreme indices compared to the present day, following Monerie et al. (2017):

$$SNR = mean(\Delta X)/\sigma(\Delta X)$$

Where for a variable $X$,
- $\Delta X$ is the change for a given model,
- $mean(\Delta X)$ is the forced change signal, which is the average over the 24 GCMs, and
- $\sigma(\Delta X)$ is the intermodel spread of the changes.

■ The authors argue that future changes in the variability of the Atlantic equatorial mode would have significant effects on future variability in precipitation extremes. However, the authors should that there is no relationship between the Atlantic equatorial mode and the precipitation indices over land (Figure 5). How could it then be possible that changes in the Atlantic equatorial mode could impact precipitation extremes over land? Please explain. Here I strongly disagree with the comments on lines 326-327, and 336-337, which are not supported by the results. Would averaging precipitation over the box be useful to extract a significant signal in the relationship with the Atlantic equatorial mode?

We agree that, according to the metric that we used to check if a signal is robust or not, it seems that there is no significant point over Guinea Coast. One should bear in mind, however, that the robustness could be assessed in different ways, such as the two-third metric for instance. In the revised manuscript, we will separate models into two groups, based on the sign of their responses to the Atlantic equatorial mode. We will, however, show the average over the Guinea Coast box and we will apply the two-third sign agreement diagnostic to see if it gives clearer results.

It is highlighted in the conclusion that "extreme rainfall anomalies related to one standard deviation of the AEM under the present-day conditions are barely significant over the Guinea coast". Please note that those anomalies are not significant, not barely significant.
We will modify our statement according to the different modifications that are planned in the revised manuscript.

In the abstract, it is stated that "the decreased variability of the AEM in a warmer climate leads to a reduced magnitude of the rainfall extreme response associated with AEM". (i) This will be more about a weaker effect of the AEM than because of a reduced variability of the AEM, because the AEM index is standardized, and Figure 8 shows reduced effects of the AEM for one standard deviation.

Thank you for the comment, we will modify out text in the revised version, accordingly.

(ii) It is argued in the introduction that the EAM effect on rainfall is stationary. This is here contradictory to the comments of the authors, please comment. The authors show the change in the regression patterns of the JAS extreme rainfall indices associated with the standardizedJAS AEM SST index (Figure 8). It would be best to also know if the regression coefficient is significant over the 2080-2099 period (*i.e.*, as for figure 5 but for the period 2080-2099). This would help understand the results of the authors.

We will improve our text to clarify the apparent contradiction that you mention. We are discussing the stationary relationship between the AEM index and the Guinea Coast rainfall (GCR) over the last century. By stationarity, we mean no change in the significant positive correlation between the AEM index and the GCR index over the observation period. We propose to add a figure in the revised manuscript, showing a moving correlation between the AEM and the GCR indices (with a window length of 30

years), from 1995 to 2099 in the GCMs. This will help to see how the overall relationship between the AEM and Guinea Coast rainfall may change with time.

We will also show the long-term future (2080-2099) regression maps of the extreme indices onto the standardized AEM index.

Ting, M., Kushnir, Y., Seager, R., & Li, C. (2009). Forced and Internal Twentieth-Century SST Trends in the North Atlantic, Journal of Climate, 22(6), 1469-1481. Retrieved Nov 4, 2022, from https://journals.ametsoc.org/view/journals/clim/22/6/2008jcli2561.1.xml

Herman, R.J., Giannini, A., Biasutti, M. *et al.* The effects of anthropogenic and volcanic aerosols and greenhouse gases on twentieth century Sahel precipitation. *Sci Rep* **10**, 12203 (2020). https://doi.org/10.1038/s41598-020-68356-w

Monerie, P., Wilcox, L. J., & Turner, A. G. (2022). Effects of Anthropogenic Aerosol and Greenhouse Gas Emissions on Northern Hemisphere Monsoon Precipitation: Mechanisms and Uncertainty, Journal of Climate, 35(8), 2305-2326. Retrieved Nov 4, 2022, from https://journals.ametsoc.org/view/journals/clim/35/8/JCLI-D-21-0412.1.xml

Hirasawa, H., Kushner, P. J., Sigmond, M., Fyfe, J., & Deser, C. (2020). Anthropogenic

Aerosols Dominate Forced Multidecadal Sahel Precipitation Change through Distinct Atmospheric and Oceanic Drivers, Journal of Climate, 33(23), 10187-10204. Retrieved Nov 4, 2022, from https://journals.ametsoc.org/view/journals/clim/33/23/jcliD190829.xml

**Additional comments**

Lines 33-35 are about future changes in heavy precipitation trends, but the following part of the paragraph (lines 35-39) is about the total wet day rainfall and rx5day. There is therefore no rationale for the "for instance" of line 36. Lines 38-39: What "could be" mean here in terms of confidence?

Lines 33-35 are about the changes in the heavy precipitation over West Africa during the last decades (and not future changes). The next sentence follows thus logically but we will modify the text in the revised version to make this clearer. Moreover, we believe that a proper detection and attribution study would be needed to allow a strong statement that attributes a flooding event as a consequence of an extreme rainfall event. It could also result from compounding events. As this has not been done in the cited article, we believe that Dike et al. (2020) prefer to use "could be". We will add a few words in our revised version to highlight this point.

Lines 40-47: What the authors are trying to demonstrate is not clear. Is there a spatial inhomogeneity in changes in rainfall indices, or is it about the complexity of changes in precipitation characteristics, that will be rainfall indices-dependent? Please rephrase the text to show the main point of the paragraph more clearly.

In the revised manuscript, this section will be rewritten, with a focus on a few extreme indices common in the literature. The complexity of this paragraph is in part due to different indices computed from different data sources, different periods, and different seasons. That did not help to synthetize easily the main information. In the revised manuscript, the text will be restructured to clarify our demonstration. It is clear that there is a spatial inhomogeneity in the trend of observed extreme indices. The different indices however help to understand the character of the rainfall, but studies over different periods and seasons can lead to different conclusions.

Line 48: Is this information obtained from observations?

This information is from observations (CHIRPS data and rain gauges). We will add this comment to the revised manuscript.

Lines 61-69: Do the authors note a relationship between bias in the different rainfall indices and bias in seasonal mean precipitation?

We did not look at the links between bias in the indices and the mean precipitation. We will provide this information in our revised manuscript. Thank you for this suggestion.

Line 71 and Line 72: "anthropogenic emission of greenhouse gases", and "the shared socioeconomic pathway scenarios". Please name the scenarios

Line 71: This is a general context statement, and we believe that there is no need to go too much into details as the different studies use different experimental designs. For instance, Rind et al. (1989); Mearns et al. (1995) used a doubling CO2 simulation. Hegerl et al. (2015) and Diedhiou et al. (2018) discussed RCP2.6,RCP4.5,RCP6.0,RCP8.5 projections. Akinsanola et al. (2020) evaluated the

RCP4.5 and RCP4.8 scenarios. Van der Wiel et al. (2021) perform specific simulations (present-day, 2°C, 3°C, and outputs from CMIP5 RCP8.5 GCMs simulations). However, for specific results that we mentioned, we provide information about the scenarios.

For line 72, Li et al, 2021, we will add the SSP scenarios used (1-2.6, 2-4.5, 3-7.0, 5-8.5) in the revised manuscript.

Line 74: The sentence is about RX1day and RX5day while the previous sentences are about extreme events. Please be clearer.

The previous sentence that you mention talks about "extreme rainfall events" in general. These events can be characterized in terms of frequency, duration and intensity. The index RX1day describes the maximum rainfall quantity in one day (so an intensity). The RX5day describes the maximum of 5 consecutive days of rainfall, which is a measure of duration and intensity. We will modify the sentence in the revised manuscript to lake this point clearer.

Line 77: "RCM-CMIP5" Please define and explain.

We will modify this expression: "Regional climate models (RCM)" forced with outputs from CMIP5 models". Thank you for the comment.

Lines 75-79: Are the results also model-dependent?

There are noticeable differences between GCMs projections and those from RCMs, in terms of rainfall characteristics. Moreover, within the RCMs simulations, there is still a spread in the magnitude of the changes, and even in the sign of the changes depending on the variable of interest. This can be seen in Figure 8 of Akinsanola and Zhou (2019). In our revised manuscript, we will show box-whisker plots for different diagnostics, which will give an idea of the model dependency of the changes.

Line 80: Does "These simulations" refer to Akinsanola and Zhou (2019)? Please be more specific.

Yes, we will clarify this point by referring to Akinsanola and Zhou (2019) at the beginning of the paragraph. Thank you for the remark.

Line 89: both enhanced.
Thank you for the remark, we will take it into account.

Lines 94-96: The increase in air moisture following Clausius Clapeyron explains a part of the seasonal mean increase in water vapor. I am puzzled about how the increase in water vapor, following Clausius Clapeyron could lead to a change in precipitation variability. Do the authors mean that it would be due to a change in variability of the temperature (SSTs) that would lead to different changes in air moisture?

This is an interesting point. Unfortunately, the authors did not provide more information. They computed a change in the rainfall variability due to the enhanced water vapor content of the atmosphere in the future . As the obtained change in variability exceeds the simulated change in the rainfall variability, and as they expect a decrease in the contribution of the dynamics, they conclude that the increase in the variability is due to thermodynamic changes. Because of this lack of information , we propose to keep  in Lines 94-96 .

Line 103: Please replace "More" with ", where"

The corrections will be applied, and the dot will be removed. Thank you

Line 107: "warming and cooling", is that following a north/south dipole? Please be more specific.

No, we were not talking about a dipole. Rather, we were referring to the different

positive and negative phases of the AEM, which are characterized by positive and negative SST anomalies in the eastern equatorial Atlantic, respectively. We will modify the statement accordingly.

Lines 113-114 could be shortened, removing "The first mode...indicates a strong", and removing ", and", in line 115.

We will shorten the statement as suggested. Thank you

Line 115: Is it the "total variability" in Guinea coast rainfall? What is the time scale considered for the variability? (*e.g.*, daily, interannual?).

This is not the variability over the Guinea Coast, but the covariability between West African rainfall anomalies and the tropical Atlantic SST anomalies. This covariability between the Guinea Coast rainfall and the eastern equatorial Atlantic SST is dominant on the interannual timescale. We will add the information to the new text.

Line 116: Is it about the wind convergence?

Yes, this is about a low-level wind convergence. We will add "wind" in the revised text.

Line 124: "variability of the AEM". Is it the daily variability? Please be more specific throughout the text.

All our analysis is focused on the interannual timescale. We will put this information at the beginning of this paragraph. Thank you for the remark.

Lines 205-215: The authors could add sentences to explain briefly why the authors are using these metrics.

We will give more motivation to the choices of the metrics.

Lines 225-230: Are the changes in rainfall indices following the changes in seasonal mean precipitation (sign and pattern)?

We will provide in the supplementary material some figures showing a scatter plot of the changes in the mean precipitation and the changes in the rainfall indices. We will say a few words in the main revised manuscript about the outcomes.

Line 225: "simulated by climate models". The authors are not showing the results for each model individually in Figure 1. Please change it to "shown by the CMIP6 ensemble mean".

We will correct the sentence. Thank you.

Line 230: "observations". Shown where?

This was shown on Figure A1.g of the submitted manuscript. This will be specified in the revised manuscript.

Lines 236-246: Are these results model-dependent?

We will provide in the revised manuscript the results at a model level. We will also comment on the model dependency of the results.

Figure 1: It would be helpful to have the observation with contours.

In the revised manuscript, we will show in color the multimodel ensemble median and in contours the ensemble median of the 6 rainfall observation datasets. Thank you for the suggestion.

Lines 261: There are plenty of references on the change in seasonal mean precipitation over West Africa and over the Sahel. Please acknowledge the literature.

We will acknowledge the literature as mentioned. Thank you.

Lines 272-273: Is the change in RX1day consistent with the shortening of the rainy season over the western Sahel?

The change in the frequency of wet days is consistent with the shortening of the rainy season over the western Sahel (Fig. 3I of the submitted manuscript). The RX1day just tells us about the maximum intensity of a daily rainfall event, which is increasing over all the regions of West Africa (Fig. 3h of the submitted manuscript). We did not aim to discuss the Sahel in our study, but we will say a few words about these changes in the revised manuscript and also acknowledge the literature.

Figure 3: The pattern of R10mm is very similar to the pattern of PRCPTOT (Figure 3). The authors could comment on the possible strong role of R10mm in the total change in precipitation. Does R10mm provide a similar result for the number of rainy days?

The best way to make this link is to evaluate the contribution of the R10mm events (defined as days when more than 10mm of rainfall occurs) to the total rainfall. We do not compute this index, and we won't do it, as we have to reduce the number of indices in our article and ??? will not be included. This choice has been motivated by the suggestions of the two anonymous reviewers. Still, this would be a nice approach.

Line 285: What is the timescale used for computing the standard deviation here?

The standard deviation is simply computed over the 20 years period (with one value per year), without removing any trend or any frequency from the raw indices. We will add this information to the new version of the manuscript.

Lines 311-312: As for the precipitation indices, result sensitive to how the forced response was removed? (*e.g.*, a linear trend here)

We didn't test different ways to remove the trend, but in our previous work (Worou et al., 2022), we tested the impact of a linear trend compared to a quadratic trend on the AEM variability. Similar results were obtained in both cases. This has not been done here to avoid to simplify the discussion.

Line 314: "total wet-day precipitation index" does not show significant differences over land in Figure 5.

Much of the robust responses to the AEM are over the equatorial Atlantic, and we have few areas over land which show a robust response in the "total wet day precipitation index". In the revised manuscript, we will apply only the two-third sign agreement metric, and we will see it the results will be clearer.

Line 364: "weakened variability". What is the considered time scale?

We are considering the interannual timescale. We will add this information to the revised manuscript.

Line 375-377: Are differences between periods significant? How would this be consistent with Figure 5 which shows no robust effects of the tropical Equatorial mode on precipitation extreme over land?

The significance between different periods depends on the metrics used. In Figure 8 of the submitted manuscript, we do not penalize models which show "no robust signal aver land" in the historical simulation. The results show some robust differences for some indices. Moreover, In the revised manuscript, we will consider only the sign of the changes and a two-third agreement, as you suggested (corresponding to Figure 8, submitted manuscript).

**References**

Giannini, A. (2003). Oceanic forcing of sahel rainfall on interannual to interdecadal time scales. Science,

302(5647):1027–1030. doi: 10.1126/science.1089357

Lübbecke, JF, Rodríguez-Fonseca, B, Richter, I, et al. Equatorial Atlantic variability— Modes, mechanisms, and global teleconnections. WIREs Clim Change. 2018; 9:e527. https://doi.org/10.1002/wcc.527

Rodríguez-Fonseca, B., Janicot, S., Mohino, E., Losada, T., Bader, J., Caminade, C., Chauvin, F., Fontaine, B., García-Serrano, J., Gervois, S., Joly, M., Polo, I., Ruti, P., Roucou, P. and Voldoire, A. (2011), Interannual and decadal SST-forced responses of the West African monsoon. Atmosph. Sci. Lett., 12: 67-74. https://doi.org/10.1002/asl.308

Rodríguez-Fonseca, B., Mohino, E., Mechoso, C. R., Caminade, C., Biasutti, M., Gaetani, M., Garcia-Serrano, J., Vizy, E. K., Cook, K., Xue, Y., Polo, I., Losada, T., Druyan, L., Fontaine, B., Bader, J., Doblas-Reyes, F. J., Goddard, L., Janicot, S., Arribas, A., Lau, W., Colman, A., Vellinga, M., Rowell, D. P., Kucharski, F., & Voldoire, A. (2015). Variability and Predictability of West African Droughts: A Review on the Role of Sea Surface Temperature Anomalies, Journal of Climate, 28(10), 4034-4060. Retrieved Jan 26, 2023, from https://journals.ametsoc.org/view/journals/clim/28/10/jcli-d-14-00130.1.xml

Losada, T., Rodriguez-Fonseca, B., Mohino, E., Bader, J., Janicot, S., and Mechoso, C. R. (2012), Tropical SST and Sahel rainfall: A non-stationary relationship, Geophys. Res. Lett., 39, L12705, doi:10.1029/2012GL052423.

Worou, K., Goosse, H., Fichefet, T. et al. Interannual variability of rainfall in the Guinean Coast region and its links with sea surface temperature changes over the twentieth century for the different seasons. Clim Dyn 55, 449–470 (2020). https://doi.org/10.1007/s00382-020-05276-5

Diatta, S.; Diedhiou, C.W.; Dione, D.M.; Sambou, S. Spatial Variation and Trend of Extreme Precipitation in West Africa and Teleconnections with Remote Indices. Atmosphere 2020, 11, 999. https://doi.org/10.3390/atmos11090999

Losada, T., Rodríguez-Fonseca, B., Polo, I. et al. Tropical response to the Atlantic Equatorial mode: AGCM multimodel approach. Clim Dyn 35, 45–52 (2010). https://doi.org/10.1007/s00382-009-0624-6

A Giannini1, S. Salack, T. Lodoun, A. Ali, A. T. Gaye and O. Ndiaye: unifying view of climate change in the Sahel linking intra-seasonal, interannual and longer time scales. Environ. Res. Lett. 8 024010 - https://doi.org/10.1088/1748-9326/8/2/024010

Monerie, P., Wilcox, L. J., & Turner, A. G. (2022). Effects of Anthropogenic Aerosol and Greenhouse Gas Emissions on Northern Hemisphere Monsoon Precipitation: Mechanisms and Uncertainty, Journal of Climate, 35(8), 2305-2326. Retrieved Jan 27, 2023, from https://journals.ametsoc.org/view/journals/clim/35/8/JCLI-D-21-0412.1.xml

Paul-Arthur Monerie et al 2017, Impact of internal variability on projections of Sahel precipitation change. Environ. Res. Lett. 12 114003 DOI 10.1088/1748-9326/aa8cda

Akinsanola, A. A. and Zhou, W. (2019). Projections of west african summer monsoon rainfall extremes from two cordex models. Climate Dynamics, 52(3):2017–2028.

---

## Author Response (AR1)

Dear Editor,

Please, find below our point-by-point responses to the different comments of the reviewers and the community comment. Our responses are highlighted in blue color.

**Responses to RC1**

Review for "Future changes in the mean and variability of extreme rainfall indices over the Guinea Coast and role of the Atlantic equatorial mode" by Worou et al.

**Overview**

The present manuscript investigates the current and future characteristics of West African rainfall according to an ensemble of 24 CMIP6 GCM models and further relates them to the development of the Atlantic equatorial mode (AEM), which has been a driver of rainfall over the Guinea Coast region. The authors use 12 ETCCDI rainfall indices to compare the nature of present-day rainfall with gridded observation data, i.e., CHIRPS, as well as with the rainfall behaviour in three future periods (near- term, mid-term, long-term) under the strongest SSP5 scenario (SSP5-8.5). By regressing CHIRPS rainfall indices to the AEM index, a reference is created in order to assess and quantify changes in the relationship between future extreme rainfall and the variability of AEM over the Guinea coast. The main results indicate increasingly less frequent but more intense rainfall over a gradually shorter period in the future. Furthermore, the AEM is suggested to have a diminished influence on future extreme rainfall over the Guinea coast region due to a decrease of AEM variability.

An increased knowledge about climate extremes over West Africa, especially that of rainfall, is of high importance for the assessment of current and future risk and subsequent development of risk mitigation strategies, action plans and decision making. In that regard, this study has the potential to be of high relevance for this part of the scientific community. However, as will be pointed out in more detail in the comments below, my biggest concern is the usage of CHIRPS for a study revolving around rainfall extremes, which potentially require a major revision. Furthermore, the study becomes diluted by the high number of rainfall-related indices, some of which are not particularly necessary to include in my opinion. Therefore, the presentation of the results somewhat suffers from an overload of numbers and a lack of structure. Overall, the topic of the manuscript is within the scope of WCD. Being non-native in English, language appears fine to me with only minor corrections to perform.

**General comments/questions**

- As mentioned above, by biggest concern is related to CHIRPS being the reference in a study about extreme rainfall. While it excels at interannual timescales over Africa (e.g., Camberlin etal., 2019), it struggles with the representation of rainfall in the extreme spectrum by showing substantial underestimation tendencies at a daily timescale (e.g., Sanogo et al., 2021; Ageet et al., 2022). Therefore, I do not think that relying on CHIRPS alone is sufficient to address the research question of this manuscript. Based on the outcomes of recent validation studies, I suggest to include the exercise with either GPCC-FDD (Becker et al., 2013) or one of the GPM products GsMAP or IMERG (Kubota et al., 2007; Huffman et al., 2015). This will allow to make at least basic statements about the uncertainty that stems from the observational data themselves.

  Thank you for this constructive comment. We include five additional observed rainfall datasets in our analyses: ARCv2, PERSIANN,  REGEN, TAMSAT, GPCC_FDD_v2022. We do not consider GsMAP and IMERG datasets, as they do not cover our period of study (1995-2014). However, we consider them in Fig. R1.1

(Fig. S1 of the revised manuscript), to compare their annual cycle characteristics compared to the other observations.

[Figure]

*Figure R1. 1 Annual cycle of the (a) total wet day precipitation (PRCPTOT), (b) very wet days precipitation (sum of the daily rainfall over days when the rainfall exceeds the 95th percentile), and (c) the contribution of the total monthly rainfall to the total annual rainfall, for nine different observational datasets. The periods considered for each dataset are displayed in the legend. The ensemble median of the observations is indicated with the black curve. The grey shading shows the 10th to 90th percentile range of the observations.*

- How did the authors choose the study periods? I was wondering why the long-term future wasselected such that it exhibits a 20-year gap to the mid-term future, while the latter directly follows up to the near-term future. Can the authors elaborate on that?

We choose the different periods of the future according to some defined periods in the IPCC AR6. More specifically, Table SPM.1 (page SPM-17 of the summary for Policymakers, in the IPCC AR6 WGI). We added this information in the revised version of the article.
These periods can be easily found in the IPCC Working Group I Interactive Atlas ( https://interactive-atlas.ipcc.ch/regional-information#eyJ0eXBlIjoiQVRMQVMiLCJjb21tb25zIjp7ImxhdCI6OTc3MiwibG5nIjo0MDA2OTIsInpvb20iOjQsInByb2oiOiJFUFNHOjU0MDMwIiwibW9kZSI6ImNvbXBsZXRIX2F0bGFzIn0sInByaW1hcnkiOnsic2NlbmFyaW8iOiJzc3A1ODUiLCJwZXJpb2QiOiJmYXIiLCJzZWFzb24iOiJKdW5BdWciLCJkYXRhc2V0IjoiQ01JUDYiLCJ2YXJpYWJsZSI6IJ4NWRheSIsInZhbHVlVHlwZSI6IJFTEFUSVZFX0FOT01BTFkiLCJoYXRjaGluZ2I6IINTVBMRSIsInJlZ2lvbINldCI6ImFyNiIsImJhc2VsaW5lIjoiQVI2IiwicmVnaW9uc1NlbGVjdGVkIjpbXX0sInBsb3QiOnsiYWN0aXZlVGFiIjoicGx1bWUiLCJtYXNraW9pbm9uZSIsInNjYXR0ZXJZTWFuIjpudWxsLCJzY2F0dGVyWWVhciI6bnVsbCwic2hvd2luZyI6ZmFsc2V9fQ== ) :

[Figure]

CMIP6 - Maximum 5-day precipitation (RX5day) Change % - Long Term (2081-2100) SSP5-8.5 (rel. to 1995-2014) - June to August (33 models)

*Figure R1. 2 An example of the defined periods in the IPCC AR6. The near-term, medium term and long-term futures correspond to 2021-2040, 2041-2060 and 2081-2100 periods, respectively.*

- The study somewhat suffers from the plethora of indices and/or the lack of a structured presentation of their results. For instance, is it necessary to include both the 95[th] and 99[th] percentile (same for R10mm and R20mm)? Is it worth to include PRCPTOT when SDII is basically the same but defined as a rate? Regarding the structure, with these many indices, it can become difficult for readers not familiar with these to keep track of the arguments being made and being confronted with too big panels. I understand that the current structure followsa logical build-up going from the current situation to the perspective in the future followed by the link to AEM. However, the authors may consider to form groups, e.g., extreme precipitation indices (R99p, R20mm, RX1day, …) and frequency (R99pf, CDD, CWD, …), and analyze their current and future characteristics in their own subsection. I believe that this will particularly improve the readability of the tables, which currently mixes indices with differentunits together.

  Thank you for this comment. We reduced the number of extreme precipitation indices in the revised version of the article. We considered six indices, instead of twelve:

  1. the SDII (simple daily intensity index), which describes the intensity of wet rainfall events
  2. the R95p (very wet day), which describes the intensity of rainfall events exceeding the 95[th] percentile
  3. the CWD (consecutive wet days), which describes the maximum duration of a wet event
  4. the RX5day (Maximum 5 days precipitation), which is the intensity of an event over a duration of five days
  5. the FRQW (Frequency of wet days), which describes the frequency of wet events
  6. R20mm (very heavy precipitation days), which is a measure of the frequency of rainfall events exceeding 20 mm, and which could have a high socio-economic impact.

- The introduction is too long, which inhibits a sharper formulation of the research question.Looking at each paragraph, the following topics are addressed:

  1. Socio-economic impacts of extreme rainfall
  2. IPPC AR6
  3. Recent trends in AMJ daily rainfall in Guinea coast region
  4. Recent trends in JAS daily rainfall in Guinea coast region
  5. 3.) and 4.) for specific coastal areas
  6. CMIP6 models behaviour of current rainfall characteristics over Africa
  7. GCM behaviour of future rainfall characteristics over Africa

8. RCM behaviour of future rainfall variability on daily and seasonal timescale
9. MCSs, AEM, SST
10. Historical and current relationship between AEM and Guinea coast rainfall
11. Goal of study

Some of the paragraphs, e.g., 3-5, can be easily merged and stripped-down to the fundamental information. I advise the authors to revise and shorten the introduction. Also, I believe that AEM should be introduced earlier to facilitate a better build-up to the research topic.

We rewrote the introduction according to the suggestions.

- **The beginning of summary and conclusion** section lacks a systematic and brief recapitulation of the motivation and research question of the study, the data and methods used and a point-by-point summary of the key results. Also, some discussion was integrated in the result section, which should rather be shifted to the this section (see specific comments). Overall, I am somewhat missing the link of the presented results with existing literature, i.e., a proper discussion, and how they integrate, complement, or disagree with them. As mentioned, a part of it can be just shifted from the result section.

We provided a systematic and brief recapitulation of the motivation and research question, the data used as well as a summary of the main results at the beginning of the summary and conclusions.

**Specific comments/questions**

L28:   "…the exposure to river flooding events is expected to increase 4.6, 8, and 8.6 times morethan without climate change". Are these numbers related to the 1.5°C, 2.4°C and 3.5°C scenarios further down the sentence? It didn't get quite clear from the structure of the sentence.

Yes, these numbers are related to the 1.5°C, 2.4°C and 3.5°C scenarios. In the revised version of the article, this sentence has been rewritten and moved to discussions in section 3.3.

L32:   "…for the safety of young people". Maybe better "future generations"?
Thank you for the suggestion, we take it into account as shown in the previous comment.

L49:   "… which represents up to 4% of the seasonal daily mean rainfall". Do the authors meanthat the JAS mean daily rainfall has decreased by 4%?
Yes, the authors mean a negative trend which represents 4% of the JAS climatology, so a decrease of 4% of the JAS mean daily rainfall. This sentence is removed from the revised manuscript.

L67:   "Regional Climate Models (RCMs) forced with CMIP5 GCMs outputs…". Are they related toCORDEX Africa?
The cited article (Akinsanola et al., 2020) stated that the integrations were performed over the West African domain. These integrations are part of the CORDEX project, but its not clear whether they were performed over the "CORDEX Africa" domain or another domain such as the "Middle East North Africa" domain, or a more reduced domain, West Africa as stated. In the revised version of our manuscript, we added that they are part of the CORDEX project, as stated in their article.

L107:  "a mode characterized by anomalous warming and cooling in the eastern equatorial Atlantic basin". It should be additionally mentioned that the phenomenon refers to aninterannual variability.

This sentence has been removed, but we take it into account.

L141-143: See general comment on the choice of the periods.
We refered to the IPCC AR6 WGI definitions to motivate our choices of the different periods.

L150-151: "Only one realization of each simulation is considered for each GCM". Based on whatcriterion did the authors decide on the ensemble member per GCM?
Since many studies using one ensemble member have shown that the current GCMs can simulate relatively well some aspects of some known oceanic internal modes of variability in the tropical Atlantic (e.g. Kucharski et al.,2017; Richter et al, 2020; Worou et al.,2022; Crespo et al.,2022; Yang et al.,2022), tropical Pacific (e.g. Cai et al.,2015) and Indian Ocean (e.g. Cai et al.,2013,2018), we rely on their conclusions and use one realization in our study. However, a further perspective would be to increase the number of realizations for each model, if available, or to restrict the study to a few GCMs having 10 or more members for both historical and future simulations, and having a common surface and atmospheric variables. This will help to understand how internal variability can impact the stimulation of specific internal modes of variability such as the Atlantic equatorial mode of variability and its associated patterns. Such an analysis is out of the scoop of our current study. We added this perspective in our revised manuscript (in the conclusion).

Table 1: The meaning of the metadata IDs (r, i, p, f) should be explained in the caption or in the text.

Why do the historical members differ from SSP5-8.5 members for some models?
The meaning of the variant-id in CMIP6 metadata are provided in the revised manuscript:
   • "r" represents the realization index
   • "i" represents the initialization method,
   • "p" represents the physics, and
   • "f" represents the forcing
In the SSP5-8.5 outputs, the parent_variant_label does not necessarily corresponds to the variant_label. We then read thoroughly the metadata in future simulation outputs and associate them to their corresponding parents, from which they were branched. Please, find below a print screen of the metadata in the daily precipitation file from the SSP5-8.5 simulation performed with CESM2. Both the variant_label (r2i1p1f1) and its associated parent (parent_variant_label) are highlighted with the red rectangle.

L153: See general comment on CHIRPS.
We added five additional datasets for the observed daily rainfall and we modified

the section accordingly.

L161-162: "This study is focused on the July-September season (JAS), when the Guinea Coast rainfall covariability with the AEM is at its maximum in the current climate models". I believe that this covariability in JAS was not mentioned in the introduction!? For the Guinea coast region, JAS largely falls within the little dry season over the Guinea coast region, thus outside of the two rainy seasons. Have the authors also investigated April-June (AMJ) and/or September-November (SON) when MCSs are more prevalent in the coastal region?

If so, I think it would be worth a few sentences (e.g., in the conclusion) on how future extreme rainfall develops in these seasons.

> The maximum covariability between the AEM and the GCR over JAS has been  shifted to the introduction section.
>
> We don't focus our analysis on the specific contribution of MCSs to the rainfall over the Guinea Coast, as there are different types of rainfall over the region (Maranan et al 2018). Although the MCSs have an important contribution to the GCR during AMJ and SON, the annual cycle of the R95p index (very wet days index) across different observations indicates the highest values during July, August, September and October (Fig. R1.1 (b)). The total wet day precipitation (PRCPTOT) and the contribution of the monthly rainfall to the annual rainfall (ANNPCT) also show their highest values over these months (Fig. R1.1 (a),(c)). We will therefore keep the JAS season, and will add the information about the prevalence of MCSs in the AMJ and SON seasons over the Guinea Coast, without any other supplement analyses. Besides, the little dry season is less pronounced in the annual cycle of the average of the total rainfall (PRCPTOT) over the region that we defined as the Guinea Coast in the present analysis (20°W - 15°E,4°N -10°N) and over the 1995-2014 period, as it can be seen in the figure below, and also mentioned by Worou et al. 2020. We added this information to the revised manuscript.

L162-165: This text snippet looks like it could also be shifted to the introduction.
> We wrote a new section about the motivation for the choice of the season and keep the text there.

L167: Have the authors tested the sensitivity of R10mm and R20mm on the choice of the resolution? I would think that different resolutions lead to different outcomes with indiceswhere the threshold is an absolute value (e.g., 1mm for a wet day).

> We do not analyze the R10mm in our revision. Figure R1.3 shows the annual cycle of the 6 indices during the1995-2015 period. Two GCMs (among 24) provide low (CNRM-CM6-1, MPI-ESM1-2-LR) and high resolution (CNRM-CM6-1-HR, MPI-ESM1-2-HR) outputs. The analysis of Fig. R1.3 reveals that the low-resolution version of CNRM-CM6 model simulates more frequent rainfall events exceeding 20mm (R20mm) compared to the high-resolution version, mainly between June and November. The five other indices also have higher magnitudes in the low-resolution version of this model, compared to the high-resolution simulations. This is not the case for the model MPI-ESM-2 which simulates an annual cycle of with a similar magnitude in its two versions, except for the maximum consecutive wet days (CWD) and frequency of wet events (FRQW).  In these latter cases, the low-resolution simulation shows higher values compared to the high-resolution values. Therefore, we can not make a general statement about the impact of resolution on the simulation of the R20mm. We can also say that both the CNRM-CM6-1 and MPI-ESM1-2 models simulate higher CWD and FRQW in their low-resolution simulations, compare to their high-resolution simulations. We did not give any comment about these

features in the new version of the manuscript, to keep the flow of our main objectives.

[Figure]

*Figure R1. 3 Annual cycle of the extreme rainfall indices over the Guinea Coast, during the 1995-2014 period. Low and high-resolution outputs are analyzed for CNRM-CM6-1 and MPI-ESM1-2 models. The ensemble median for the 24 GCMs and the 6 observations are also shown, respectively.*

L175:  What remapping procedure (bilinear, bicubic, …) is used?
We used a bilinear (conservative) remapping procedure with a routine from the climate data operator (CDO), to interpolate the sea surface temperature (rainfall) data to a common grid. We added this information to our revised text.

Table 2:  Mention again in the caption that the indices are based on ETCCDI. Is PR95/PR99calculated from wet days only or from all timesteps?
We added in the table caption that the indices' definitions are provided by the ETCCDI. We also mentioned that the PR95 is computed from only wet days timeseries over a considered period, for each month.

L222:  I think the TSS deserves a bit more explanation here about which combination of measures it attempts to create a skill score with (i.e., correlation coefficient and standard deviation). It is worth noting that these components can be weighted differently. For instance, deviations in the standard deviation can be penalized harsher than the correlation coefficient. I believe however that the standard formulation of the Taylor skill score is used here.

Thank you for having drawn our attention to the formulation of this performance skill. We used a modified version of the standard Taylor skill score, where the terms containing a correlation are to power 2 (instead of 1 in the standard formulation). However, there is an error in our text, where we miss the power 2 in the term $(1 + PCC_0)^2$ in the denominator. This will be corrected in our revised text.

$$TSS = \frac{4(1 + PCC)^2}{\left(\frac{\sigma_{cmip6}}{\sigma_{observation}} + \frac{\sigma_{observation}}{\sigma_{cmip6}}\right)^2 (1 + PCC_0)^2}$$

Moreover, we noted in the new text that one can choose to penalize models with low correlations or low spatial variability by weighting differently the terms in the TSS equation.

L250:  "… and 250 the percentage of bias reaches 0.98%, against −1.28% for RX1day". This isnot much, is it?
This variable is no longer considered in the new version of the article.

L251-252: "Alongside, the wet days and extremely wet days (R95p and R99p) statistics

over the entire Guinea Coast show positive biases that represent 14.66% and 24.10% of the observations". This potentially falls back again to the issues of CHIRPS and the question, whether this could rather be due to the deficiencies of CHIRPS to resolve extremes.

In the revised version, we computed the biases of each model relative to each of the six observations. The results of the median values of the biases across all the models and observations are presented.

Table 3: How robust are these numbers? How much variance do they exhibit across the members?

The authors may consider expressing these numbers rather in a box-whisker plot to account for the uncertainty of the ensemble.

Thank you for this comment. In the revised manuscript, we presented these performance metrics in a box-whisker plot, considering six different observations.

L281-282: "The spatial distribution of the mean changes in R95p, R10mm and R20mm in our study are different from the patterns obtained in the RCM-CMIP5 projections (Akinsanola and Zhou, 2019)". In what way? Please elaborate on that in more detail.

We reformulated the comparison with Akinsanola and Zhou in Section 3.3 of the revised manuscript. The main differences arise from the methods used to identify robust changes in the projections. We mentioned it in the revised manuscript.

Figure 3: "… diagonal bars…". Better use "hatching".

We used "hatching" instead of "diagonal bar". Thank you for the suggestion.

L290: "The changes in the near-term period relative to the present-day conditions are not clear for the majority of the indices". How is this seen from Fig. 4?

We see this from the error bars associated with each change. If for a variable and a specific period in the future, the error bar around the mean change covers negative to positive values, this means we cannot see a clear increase or a clear decrease in the considered variable. This is the case for the majority of the near-term changes in mean and standard deviation for different indices. In the revised manuscript, this figure has been replaced by a box-whisker plot similar to Figs. R1.4-R1.5.

[Figure]

*Figure R1. 4 Change (relative to 1995-2014) in the mean JAS extreme rainfall indices averaged over the Guinea Coast.*

[Figure]

*Figure R1. 5 Change (relative to 1995-2014) in the standard deviation of the extreme rainfall indices averages of the Guinea Coast.*

L291:    "The mid-term and long-term changes indicate a clear increase in mean and standard deviation…". Again the question from the general comments about how much of an impacton the increase the 20-year jump has from the mid-term to long-term future period!?

We could have used two 30 years periods in the last decades of the 21st century, as in Worou et al 2022. This would cancel the gap between the mid-term and long-term future periods. However, the main message of our study would not have changed, which is the reduction in the AEM contribution to the variability of the extreme rainfall events over the Guinea Coast. We discussed in the revised manuscript (section 3.3) the important role of internal variability in the first decades of the projections, which decreases in the end of the 21st century. We left this question without any additional analysis.

Figure 4:  Which significance test was used?

We did not make a test of significance in the difference of the multi-model mean of an index when comparing a future period to the historical period. Rather, we computed the mean difference and we associated an error bound in the estimation of the mean difference (across the different models), at a confidence level of 90%.
In the revised version of the manuscript, we modified this figure by showing the box plot of the changes in mean and standard deviation, as indicated in Figs. R1.4 and R1.5.

L320:    "… there is overall a good spatial distribution of the SDII and PRCPTOT anomalies overWest Africa". Do the authors mean a "good agreement"?

Yes, we mean a "good agreement. We no longer consider PRCPTOT in the newly selected indices for the revised manuscript.

Figure 5:  While the structures of the indices pretty much look alike for EnsMean, the spatial variability for CHIRPS is visibly higher in Fig. A2. Can the authors elaborate on the sourcesof these differences?

In the revised manuscript, we compared the spatial distribution of the extreme indices responses to the AEM phases between GCMs and six different rainfall datasets, to take into account the variability in the observations.

Table 4:  This is again a comparison with CHIRPS, correct? Then it should be mentioned in thecaption as such.

Yes, this is a comparison with CHIRPS. In the revised manuscript, this table has been replaced by a figure with different boxplots of the model's performances relative to the six rainfall observations. The caption has been adapted accordingly.

L363-375:  This belongs in the conclusion section in my view.

We keep this point here to remind the reader that the AEM variability is projected to decrease in the future, and we expect a decrease in its influence on extreme rainfall indices over Guinea Coast.

Figure 7:  Please explain in more detail in the text on what you computed here with respect to one standard deviation of the AEM. You varied AEM by one standard deviation and quantifiedthe difference?

Let's consider the AEM index computed from monthly SST anomalies (detrended linearly) over the ATL3 box and averaged over JAS. For 1995-2014 period, for instance, we obtain a timeseries of 20 time steps. Then, this index is divided by its standard deviation. When we regress a grid point value of a variable onto the above-described AEM index, we obtain a pattern of anomalies related to one standard deviation of the AEM index (or the standardized AEM index).
In Figure 7 computed the changes in the regression coefficients of the different indices related to the standardized AEM index. Then, the changes in the regression coefficients are averaged over the Guinea Coast for each model. Figure 7 shows the multi-model ensemble mean changes in the area-averaged regression coefficients and the associated 90% confidence interval. In the revised manuscript, this figure 7 is replaced by a box-whisker plot, using six indices instead of twelve. We also improve the AEM index description in the methodology.

L439:     One "a" too many.

          Thank you for the correction. We suppressed one "a".

Table 5:   "… of the proportion of the variance explained by the AEM". What exactly does that mean?

          Are the median values of the indices potentially calculated from different sample sizes?

          For each period, and for each index (anomaly) averaged over Guinea Coast, we computed the proportion of variance explained by the AEM for each model. We obtain 24 different values from 24 GCMs. Values in Table 5 represent the multi-model median from each of the 24 values, each period and each index. In the revised manuscript, Table 5 is replaced by a box-whisker plot, and we provided more details on the variance explained fraction in section 4.2.

Summary: See also general comment on the summary and conclusion section. It does not get clearwhat the source of the rainfall indices are, which SSP scenario was taken, that the link ofextreme rainfall to AEM in current and future period is investigated, etc. This should be more carefully and meticulously summarized right in the beginning.

          We improved the summary and conclusion accordingly.

**References**

Ageet, S., Fink, A. H., Maranan, M., Diem, J. E., Hartter, J., Ssali, A. L., & Ayabagabo, P. (2022). Validation of Satellite Rainfall Estimates over Equatorial East Africa. Journal of Hydrometeorology,23(2), 129-151.

Becker, A., Finger, P., Meyer-Christoffer, A., Rudolf, B., Schamm, K., Schneider, U., & Ziese, M. (2013). A description of the global land-surface precipitation data products of the Global Precipitation Climatology Centre with sample applications including centennial (trend) analysis from 1901–present.Earth System Science Data, 5(1), 71-99.

Huffman, G. J., Bolvin, D. T., Braithwaite, D., Hsu, K., Joyce, R., Xie, P., & Yoo, S. H. (2015). NASAglobal precipitation measurement (GPM) integrated multi-satellite retrievals for GPM (IMERG).
Algorithm Theoretical Basis Document (ATBD) Version, 4(26).

Kubota, T., Shige, S., Hashizume, H., Aonashi, K., Takahashi, N., Seto, S., ... & Okamoto, K. I. (2007).Global precipitation map using satellite-borne microwave radiometers by the GSMaP project: Production and validation. IEEE Transactions on Geoscience and Remote Sensing, 45(7), 2259-2275.

Sanogo, S., Peyrillé, P., Roehrig, R., Guichard, F., & Ouedraogo, O. (2022). Extreme Precipitating Events in Satellite and Rain Gauge Products over the Sahel. Journal of Climate, 35(6), 1915-1938.

References:

Maranan, M, Fink, AH, Knippertz, P. Rainfall types over southern West Africa: Objective identification, climatology and synoptic environment. Q J R Meteorol. Soc. 2018; 144: 1628-1648. https://doi.org/10.1002/qj.3345

Worou, K., Goosse, H., Fichefet, T., Guichard, F., Diakhaté, M. : Interannual variability of rainfall in the Guinean Coast region and its links with sea surface temperature changes over the twentieth century for the different seasons. Clim Dyn 55, 449–470 (2020). https://doi.org/10.1007/s00382-020-05276-5

Kucharski, F. and Joshi, M.K. (2017), Influence of tropical South Atlantic sea-surface temperatures on the Indian summer monsoon in CMIP5 models. Q.J.R. Meteorol. Soc, 143: 1351-1363. https://doi.org/10.1002/qj.3009

Richter, I., Tokinaga, H. An overview of the performance of CMIP6 models in the tropical

Atlantic: mean state, variability, and remote impacts. Clim Dyn 55, 2579–2601 (2020). https://doi.org/10.1007/s00382-020-05409-w

Worou, K., Goosse, H., Fichefet, T., and Kucharski, F.: Weakened impact of the Atlantic Niño on the future equatorial Atlantic and Guinea Coast rainfall, Earth Syst. Dynam., 13, 231–249, https://doi.org/10.5194/esd-13-231-2022, 2022.

Crespo, L.R., Prigent, A., Keenlyside, N. et al. Weakening of the Atlantic Niño variability under global warming. Nat. Clim. Chang. 12, 822–827 (2022). https://doi.org/10.1038/s41558-022-01453-y

Yang, Y., Wu, L., Cai, W. et al. Suppressed Atlantic Niño/Niña variability under greenhouse warming. Nat. Clim. Chang. 12, 814–821 (2022). https://doi.org/10.1038/s41558-022-01444-z

Cai, W., Santoso, A., Wang, G. et al. ENSO and greenhouse warming. Nature Clim Change 5, 849–859 (2015). https://doi.org/10.1038/nclimate2743

Cai, W., Zheng, XT., Weller, E. et al. Projected response of the Indian Ocean Dipole to greenhouse warming. Nature Geosci 6, 999–1007 (2013). https://doi.org/10.1038/ngeo2009

Cai, W., Wang, G., Gan, B. et al. Stabilised frequency of extreme positive Indian Ocean Dipole under 1.5 °C warming. Nat Commun 9, 1419 (2018). https://doi.org/10.1038/s41467-018-03789-6

**Review: "Future changes in the mean and variability of extreme rainfall indices over the GuineaCoast and role of the Atlantic equatorial mode"**

Authors: Koffi Worou, Thierry Fichefet, and Hugues Goosse

This study utilizes CMIP6 future climate projections to understand how Guinean Coast extreme rainfall events are likely to change in the future under the SSP5-8.5 scenario. More specifically, this study focuseson the relationship between the Atlantic Equatorial Mode (AEM) and extreme Guinean Coast rainfall. Results indicate that the extreme rainfall responses to the AEM appears to be captured by the ensemble mean of 24 different CMIP6 models for the present day, though there are biases especially in terms of magnitudes of the response. In the future, they find that the decreased variability of the AEM leads to a reduced magnitude of the rainfall extreme responses over the Guinean Coast.  Thus, while there is an overallprojected increase in the variability of most extreme rainfall indices, the contribution via the AEM to this variability weakens in the future warmer climate.

Overall, the paper is well written and will be a welcome addition to the scientific community. Before acceptance, I have a few issues detailed below that the authors need to think about and address.

**Major Comments:**

- Missing from this manuscript is analysis and/or explicit reference to past studies that have evaluatedthe ability of the CMIP6 models to realistically represent the AEM in terms of the SSTAs. I thinkthe authors may have done this analysis in their prior work (Worou et al. 2022), but the outcome isnot explicitly mentioned here in this manuscript.  This dawned on me by the time I got to lines 359

    – 360 as I started to ponder whether the poor skill scores were just an artifact that the coupled GCMs cannot replicate this mode of variability very well, so of course the relationship between theAEM and the extreme rainfall would not be well captured.
    Thank you for this excellent comment. The current GCMs can replicate the SST variability associated with the AEM over the tropical Atlantic Ocean, but they struggle to replicate the spatial distribution of the observed rainfall response to its phases over the tropical Atlantic and the Guinea Coast. This has been highlighted in our prior work (Worou et al. 2022), and we recalled this point in the revised manuscript.

- The authors select CHIRPS daily rainfall data to evaluate and validate the present day CMIP6 results against.  That being said there is no evaluation regarding the uncertainty in the observationsthat you are trying to match up the CMIP6 models against, so it is unclear what biases CHIRPS has in respects to extreme rainfall.  I would suggest drawing in additional daily rainfall datasets for thiscomparison, maybe ARC2 and/or TAMSAT. Both of these datasets extend back to 1983 which would give the authors close to similar temporal coverage to CHIRPS.  Doing so would allow theauthors to comment on the extent to which the differences between the observations and CMIP6 models is due to the model compared to the uncertainty in the observations. Similarly, you could evaluate multiple SST datasets, though this is less of an issue compared to the rainfall field in my opinion.
    Thank you for the suggested rainfall data. We have added a total of five additional observed rainfall data in the revision of this work. These data are: ARCv2, PERSIANN,  REGEN, TAMSAT, GPCC_FDD_v2022. As an example, Fig. R2.1  shows that the model multi-median performs well in the simulation of some characteristics of the rainfall over Guinea Coast. It however overestimates the frequency of wet days (FRQW) and the maximum consecutive wet days (CWD). In our revised manuscript, we considered the performance of the models across the 6 different rainfall datasets. When needed, the multimodel median values across the different observations and the different models will be indicated (for the mean biases map for instance).

Furthermore, we do not add additional SST data, as we also think that this is not a big issue.

[Figure]

*Figure R2. 1 Annual cycle of the extreme rainfall indices over the Guinea Coast (1995 – 2014): median (black and brown curves), 10th - 90th percentile range (gray and brown shades) are indicated in black for 6 observations and in brown for 24 CMIP6 GCMs. Vertical lines indicate the season considered in our study.*

- Around line 159: I didn't pick up on this subtlety at first, but realized later on in the manuscript that the authors are evaluating the SSTs for the AEM at monthly timescales, yet they are evaluating rainfall presumably at daily timescales for extreme rainfall events. I guess this may be appropriate, though I wonder if it would be more appropriate to evaluate SSTs at sub-monthly timescales instead (weekly?), as presumably there are sub-monthly variations in equatorial Atlantic SSTs that could be important. Anyhow, to avoid this confusion I suggest the authors explicitly motivate in the text why they choose to evaluate SSTs at the monthly scale to help clarify this decision for the reader.

Just to clarify the point, on the one hand, we are computing monthly extreme rainfall indices from daily rainfall values. Then we consider an average over the boreal summer season, from July to September. This means that in the end, we have seasonal values for the rainfall indices for each year of the considered period. On the other, we are interested in the variability of the Atlantic equatorial mode (also called Atlantic Niño), which peaks from the end of spring to the boreal summer and occurs mainly on the interannual timescale. There is a second Atlantic Niño phenomenon in the eastern equatorial Atlantic, which variability peaks in the boreal winter (November – December), and is named Atlantic Niño II. This is just to show that we have different oceanic modes of variability on different timescales, and to underline that we are interested in a particular phenomenon, the Atlantic equatorial mode. We are not interested in the variability at weekly timescales, though they are also important for sub-seasonal to seasonal coupled atmosphere-ocean processes such as the Madden-Julian Oscillation, which have also important impacts on the West African hydroclimate. We improved the description of the indices computation in the revised manuscript.

- I'm unclear why the authors chose to detrend the SST data. I would think they would want to keepit in raw form because they calculate their extreme indices for each year and then average the yearlytotals up over the 20 years evaluated over for each time slice if I understand correctly. This decisionneeds to be explained better in the manuscript. Also – with all of this data manipulation(detrending, normalizing, standardizing), it would be beneficial if the authors would compare these among the models and observations, so the reader can understand what is being removed/changed,and how alike/dislike are the various models exactly are.

  There are two main points in our study. First, we look at the mean changes in the extreme variables, as well as the changes in their variability. The extreme rainfall indices are kept in their raw seasonal values for each year. In the second point, we address changes in the variability of the extreme indices around their mean values. We remove any linear drift in the 20-year period timeseries, from each monthly series before averaging over the July-September season. Our aim is not to study the trend in the data, but rather, the interannual variability in the extreme indices that is partly explained by the Atlantic equatorial mode. Similarly, the Atlantic equatorial mode positive or negative phases are defined by SST anomalies exceeding at least one standard deviation around the SST mean values in the eastern equatorial Atlantic. Any linear trend (due to global warming for example) needs to be removed, to be focused on the AEM phenomenon, though climate change could have an impact on the AEM variability. We added a little clarification in the revised manuscript (section 2.4.1).

- I know there is a lot of information to convey in a confined article, but I really think it would be useful here in this study to not only analyze results from the ensemble means, but also evaluate individual models to identify which produce more realistic distributions/frequency/intensities of extreme rainfall events. This information could be used to eliminate inclusion of specific models that are judged to perform "poorly" for a given index (or overall), and thus could potentially increase the accuracy/realism of your ensemble mean by eliminating them from consideration before formulating your ensemble mean.

  An example of what I mean involves section 3.2 (Fig. 1 & Table 2). You could evaluate the individual models here and report on whether most models are outliers, or whether most models are close to the ensemble mean with a few outliers on each side. Knowing this could really strengthen the results and the reader's confidence in the ensemble mean. While I understand it is unfeasible to show Fig. 1 for all 24 models, but maybe you could consider expanding/reimaginingTable 3 to include info for all of the models as well as the ensemble mean and organize the individual models from those that perform the best to the worst for your selected indices.

  Likewise, you could do the same for Tables 4 and 5, expanding them to include individual modelsranked from best to worst to comment on their relationship to the ensemble mean results already shown.

  Thank you for this meaningful comment. First, we replaced tables 3, 4, and 5 by a box-whisker plot, to highlight the variability among the models in their representation of the different extreme rainfall characteristics. When needed, we used the multi-model ensemble median instead of the multi-model ensemble mean, which is resistant to outliers.
  We also showed each model's performance relative to the 6 observations. This will help the readers to see for each model, the skill in representing the different variables.

Furthermore, a model which poorly represents a particular index does not necessarily struggle in representing another index. This choice of models' stratification could be easily done for one variable but it will make difficult the presentation of the models. We do not split the models into different categories, as the robustness of the changes in the different variables as well as the skill of the models varies a lot, according to the extreme index considered.

**Minor Comments:**

- Line 145: I presume you are using daily SST and rainfall data, correct? Or is it 3-hourly? Suggest you explicitly mention this in the text here for better clarity.

  We are using monthly SST data and daily rainfall data. We mentioned explicitly the temporal resolution of the datasets as suggested. Thank you.

- Lines 165 – 168: Would be helpful to include a figure to orient the reader here, showing the countries and a box of the analysis region you defined in the text.

  We added countries on a map, and a box showing the Guinea Coast region.

- Table 1: Include information on the spatial resolution for each model evaluated so that information is conveyed to the reader.

  The spatial resolution of each model has been added to Table 1 in the revised manuscript.

- Line 168: I am confused why you are selecting 2.8° resolution for use here. Maybe it is due to the coarsest resolution of the GCMs? Suggest motivating this choice better here at first mention (adding the spatial resolutions for each model to Table 1 would help here too). Actually – I see you have done this later on line 175. I'd suggest moving it up to this line to avoid confusion.

  The 2.8° resolution corresponds to the coarser resolution of the 24 models. We moved the information to the right place, as suggested, and motivated the choice of this resolution. Thank you.

- Section 2.3.1: Motivate why you chose 1° resolution here. I suspect the CMIP6 models are coarser, so how exactly did you interpolate to a higher resolution?

  There are models with low resolution, not sufficient to have many grid points to compute the index of the Atlantic equatorial mode over the ATL3 region (20°W-10°E,3°S-3°N). We, therefore, use a bilinear interpolation method with the climate data operator routine (CDO) to interpolate all the sea surface temperature datasets to a common grid of 1° of resolution. We added this information to the revised manuscript. Such a manipulation of the data has been applied in Worou et al., 2022 and Kucharski et al., 2017.

- Line 175-176: Do you mean "….averaged over the JAS season **for each year**"? It would also be informative if the authors could calculate and report on the standard deviation over each time slice interval you evaluate over to see how its variability is changing.

  Thank you for this remark. In the revised manuscript, we improved the description of the method in section 2.4.2.

  Moreover, we added a box-whisker plot of the changes in the mean and standard. We showed the distribution of the changes in the standard deviation over three future periods, instead of their climatology over the four periods, which would be informative. Figure R2.2 shows the mean changes in the standard deviation of the extreme indices over Guinea Coast:

[Figure]

*Figure R2. 2 Near-term (2021-2040), mid-term (2041-2060) and long-term (2080-2099) mean changes (relative to 1995-2014) in the standard deviation of the timeseries of JAS extreme rainfall indices averaged over Guinea Coast. Each box-whisker plot represents the changes in 24 different GCMs, in the SSP5-8.5 scenario relative to the historical simulations.*

- Line 188 – 190: "performs better" in terms of what exactly? Can you clarify what you mean herebetter? Also – do you still intend to evaluate individual models to identify which produce more realistic distributions/frequency/intensities of extreme rainfall events? This would be important toinclude I would think. Furthermore, this information could be used to eliminate inclusion of specific models that perform "poorly" for a given index, and thus could potentially increase the accuracy/realism of your ensemble mean by eliminating them from consideration.

  By "performs better", we mean that the multimodel ensemble mean values are closer the observations than each individual model. This can be seen in Fig. R2.1, where we show the multimodel ensemble median instead of the multimodel mean. We improved the text accordingly.
  Moreover, we showed the skills of each model, compared to the different observations.
  Moreover, we do not stratify models into "poor" or "good" models in the analysis of the AEM impact of the extreme indices over Guinea Coast, to have a more focused analysis on the overall changes in the AEM impact on the extreme indices over Guinea Coast.

- Figure 2 – again how does this change when a different target other than CHIRPS is used? If it isthe same, you could just comment on it in the text without adding additional figures.

     If it is different, it may be useful to include/expand a figure showing the changes if a different target is used.

     We updated our results by using additional observed rainfall data in the revised manuscript.

- Line 460: Suggest expanding this to include discussion in terms of the uncertainty that exists in the observations by using more than 1 rainfall dataset to evaluate the CMIP6 models against (see prior comment earlier).

     We updated in the revised manuscript discussion of uncertainties related to the observations.

- Lines 464 – 465: Comment on how the ensemble mean relates to the spread of the individual members that are used to determine the ensemble mean. Are most models close to the ensemble mean, or are most outliers and they average out to the mean?

     We discuss the spread in both models and observations in the revised manuscript.

- Lines 483 – 485: So how well can we expect the CMIP6 coupled models to replicate the AEM, and what are the implications for this on your findings here?

     The CMIP6 models can replicate the SST pattern associated with the AEM over the tropical Atlantic. However, they show some difficulties in replicating the rainfall responses over the tropical Atlantic and the Guinea Coast. We mentioned these points in the revised manuscript. This work, however, extends the application to the representation of the AEM and extreme rainfall indices relationships.     This has been done in section 4.1.

- On all spatial map figures in the manuscript it would be helpful if the country outlines were included in each panel.

     Thank you for this comment. We added country contours on the different maps.

References

Worou, K., Goosse, H., Fichefet, T., and Kucharski, F.: Weakened impact of the Atlantic Niño on the future equatorial Atlantic and Guinea Coast rainfall, Earth Syst. Dynam., 13, 231–249, https://doi.org/10.5194/esd-13-231-2022, 2022.

Kucharski, F. and Joshi, M.K. (2017), Influence of tropical South Atlantic sea-surface temperatures on the Indian summer monsoon in CMIP5 models. Q.J.R. Meteorol. Soc, 143: 1351-1363. https://doi.org/10.1002/qj.3009

**Review of "Future changes in the mean and variability of the extreme rainfall indices over the Guinea coast and role of the Atlantic equatorial mode" by KoffiWorou, Thierry Fichefet, and Hugues Goosse.**

The authors assess the future evolution of precipitation characteristics (*e.g.*, extreme rainfall events) and the effects of changes in the Atlantic equatorial mode on Guinea coast precipitation variability and on rainfall indices. The topic is relevant for the scientific community and could help understanding better future changes in precipitation over West Africa.

I have several concerns that should be assessed before considering the study for publication.

- The novelty of the work is unclear. All results on changes in precipitation extremes have already been shown in the literature, and the section on the effects of the Atlantic Equatorial mode is not convincing. Please explain better what is the novelty of the study.

  Thank you for this comment. The motivation of this study is based on the fact that over the 20th century, the Atlantic equatorial mode of variability has been the main oceanic driver of the interannual rainfall variability over Guinea Coast, as shown in many studies (Giannini et al.,2003; Rodríguez-Fonseca et al.,2011,2015; Losada et al.,2012, Lübbecke et al.,2018; Worou et al.,2020). Other studies pointed out that this oceanic internal mode of variability could also impact extreme rainfall events over West Africa (Diatta et al,2020). An extraordinary high impact rainfall event occurs in Burkina Faso, on the 1st of September 2009, coinciding with below normal sea surface temperature (SST) conditions in the eastern equatorial Atlantic. This motivates us to study future changes in the SST variability in the eastern equatorial Atlantic an the impacts on the occurrence of extreme rainfall events over the Guinea Coast. This approach, is new, because it studies a well-known couple ocean-atmosphere phenomenon, the Atlantic equatorial mode, which is part of the numerous internal climate modes of variability. We improved our motivation in the revised manuscript, accordingly.

- The introduction is too long, and its structure could be revised, with the information streamlined to provide a summary of the literature. For instance, information from Bichet and Diedhiou (2018), Odoulami and Akinsanola (2017), and Kpanou et al. (2018) is for different seasons (*g.*, AMJ, JAS), for different periods, and different metrics. Information seems also contradictory, with Odoulami and Akinsanola (2017) stating that there is a negative trend in extreme events (relative to the 95th percentile)over the Guinea coast while Kpanou et al. (2018) state that the number of such eventshas increased over some of the West African countries (Ivory coast, Togo, and Benin).
- The authors have used monthly SSTs to compute the AEM index. Please explain how monthly variabilities in SSTs could lead to changes in the variability of rainfall extremes.

  We would like your attention on the fact that we computed seasonal SST indices of the Atlantic equatorial mode (AEM) from monthly SST anomalies (linearly detrended) averaged over July, August, and September (JAS). The impact of the AEM is evaluated on the seasonal rainfall characteristics.
  Moreover, we did not show the dynamics at play in our current study, and we will provide in the new version of the article, a few statements about the mechanism. Warm phases of the AEM during the boreal summer lead to strong low-level convergence and rising motion of humid air over the eastern equatorial Atlantic. The land-sea surface pressure gradient weakens, and shifts southward the penetration of the monsoon flow, compared to the climatology. Then, humid air is advected by the low-level circulation from the equatorial Atlantic to the coastal areas of West Africa. This provides favorable conditions

to an increase in rainfall occurrence as well as extreme rainfall events over Guinea Coast. The physics behind the connection of the AEM to West Africa and the tropical regions is well known and can be read in detail in Losada et al,. 2010; Lübbecke et al.,2018; Worou et al., 2020,2022, among others. We recalled the mechanisms in the introduction of the revised manuscript.

We also reduce the introduction, summarize the main informations and better structure the review of the works on the extreme rainfall events over Guinea Coast, with a focus on the boreal summer season.
It is hard to compare Odoulami and Akinsanola (2017) and Kpanou et al (2018), because the former study is focused on June-September (1998-2013), while the latter is on an annual basis (1981-2015). Thank you for this meaningful comment.

- Method: The authors defined an anomaly as robust when 50% of the models present a significant regression coefficient. However, this low value could be eventually obtained by chance and would not show robustness in the results. The authors should revise this threshold, using the two-thirds threshold used for the sign of the EnsMean. Data have been linearly detrended. However, it is shown that anthropogenic aerosols have strong effects on West African precipitation and have driven a part of the 1970s-1980s drought and of the precipitation recovery (*e.g.*, Herman et al., 2020, Hirasawa et al. 2020; Monerie et al. 2022). Removing a linear trend will thus not allow considering the full effect of the anthropogenic activity on West African precipitation. The authors should check the robustness of the results using other methods, such as estimating the forced response using the ensemble mean (see for instance Ting et al. 2009).

We really like this comment, thank you for the suggestions.

We revised the robustness metric for the change in the regression patterns related to the AEM. We only consider the two-third threshold for the sign-agreement among the 24 models.

Our aim is not to assess the impact of the aerosols on the extreme rainfall trend over the Guinea Coast, which is why we remove any drift from the anomalies computed for each index. We will better motivate the detrending of the data in the methods. However, we agree that anthropogenic aerosol forcing can have a long-term impact on the climate of different regions around the world. Over the past decades, changes in the anthropogenic aerosol forcing have led to differences in the sea surface temperature between the North Atlantic and the global tropics, which led to different hydrological conditions over the Sahel (A Giannini et al., 2013). You showed (Monerie et al.,2022) that the AA induces a negative trend in the rainfall over West Africa during 1950-1980, and a positive trend afterward. These aerosol emissions have also induced a north-to-south SST gradient trend in the tropical Atlantic, which led to a weakening of the AEM and its impact on the rainfall over the Guinea Coast. We stated that will mention the impact of different forcing on the West African climate, and highlight the role of aerosol in our introduction. This is no longer the case, as we shorten the revised introduction. As above-mentioned, we are not interested in the aerosol impact on the rainfall trend over Guinea Coast. Rather, we want to assess changes in the year-to-year variability around the trend.

For the last point, we computed a signal-to-noise ratio to compare the forced response to the internal variability. We compared the IPCC AR6 methodology to th one used in Monerie et al 2017 (Fig. S4 of the revised manuscript):

$$SNR = mean(\Delta X)/\sigma(\Delta X)$$

Where for a variable $X$,
- $\Delta X$ is the change for a given model,
- $mean(\Delta X)$ is the forced change signal, which averaged over the 24 GCMs
- $\sigma(\Delta X)$ is the intermodel spread of the change

- The authors argue that future changes in the variability of the Atlantic equatorial mode

would have significant effects on future variability in precipitation extremes. However, the authors should that there is no relationship between the Atlantic equatorial mode and the precipitation indices over land (Figure 5). How could it then be possible that changes in the Atlantic equatorial mode could impact precipitation extremes over land?Please explain. Here I strongly disagree with the comments on lines 326-327, and 336-337, which are not supported by the results. Would averaging precipitation overthe box be useful to extract a significant signal in the relationship with the Atlantic equatorial mode?

We agree that according to the metric that we used to verify if a signal is robust or not, it seems that there is no significant point over Guinea Coast. One should bear in mind, however, that this metric is subjective, and the robustness could be assessed in different ways, such as the two-third metric or another level of penalization that we want. Moreover, we have shown in Worou et al.2022 that there as some groups of models that show statistically significant rainfall regression coefficients over the Guinea Coast, and other models fail also show differences in their atmospheric responses, which could explain the lack of significance in some of the responses. One would like to show a robust signal across all the models, but unfortunately, this is not the case all the time. In the revised manuscript, according to the two-third metric that you suggest, Fig. S23 indicates a robust extreme response to the AEM over the four different periods, and for the different indices. We rely on the average over the Guinea Coast box to give information on the overall changes in the extreme rainfall index responses to the AEM. We take care to specify heterogeneity in the spatial changes as well as the lack of significant changes in the regression patterns.

It is highlighted in the conclusion that "extreme rainfall anomalies related to one standard deviation of the AEM under the present-day conditions are barely significant over the Guinea coast". Please note that those anomalies are not significant, not barelysignificant.

We modified our statement according to the different modifications that are planned in the revised manuscript.

In the abstract, it is stated that "the decreased variability of the AEM in a warmer climate leads to a reduced magnitude of the rainfall extreme response associated with AEM". (i) This will be more about a weaker effect of the AEM than because of a reducedvariability of the AEM, because the AEM index is standardized, and Figure 8 shows reduced effects of the AEM for one standard deviation.

Thank you for the comment. We tested our results in Worou et al 2022 by using composite analyses instead of linear regressions and we obtained similar results. We are confident in our statement.

(ii) It is argued in the introduction that the EAM effect on rainfall is stationary. This is here contradictory to the comments of the authors, please comment. The authors show the change in the regression patterns of the JAS extreme rainfall indices associated with the standardizedJAS AEM SST index (Figure 8). It would be best to also know if the regression coefficient is significant over the 2080-2099 period (*i.e.*, as for figure 5 but for the period 2080-2099). This would help understand the results of the authors.

 We are talking about the stationary relationship between the AEM index and the Guinea Coast rainfall (GCR)over the last century. By stationarity, we mean no change in the significant positive correlation between the AEM index and the GCR index in the observation. This term has been widely used in the literature.

We also showed the long-term future regression maps of the extreme indices onto the standardized AEM index in Fig. S23 (revised manuscript), for the four different periods of our study. It clearly shows long-term weaker responses of the extremes indices to AEM, compared to the present-day responses.

Ting, M., Kushnir, Y., Seager, R., & Li, C. (2009). Forced and Internal Twentieth-Century SST Trends in the North Atlantic, Journal of Climate, 22(6), 1469-1481.

Retrieved Nov 4, 2022, from
https://journals.ametsoc.org/view/journals/clim/22/6/2008jcli2561.1.xml

Herman, R.J., Giannini, A., Biasutti, M. *et al.* The effects of anthropogenic and volcanic aerosols and greenhouse gases on twentieth century Sahel precipitation. *Sci Rep* **10**, 12203 (2020). https://doi.org/10.1038/s41598-020-68356-w

Monerie, P., Wilcox, L. J., & Turner, A. G. (2022). Effects of Anthropogenic Aerosol and Greenhouse Gas Emissions on Northern Hemisphere Monsoon Precipitation: Mechanisms and Uncertainty, Journal of Climate, 35(8), 2305-2326. Retrieved Nov 4, 2022, from https://journals.ametsoc.org/view/journals/clim/35/8/JCLI-D-21-0412.1.xml

Hirasawa, H., Kushner, P. J., Sigmond, M., Fyfe, J., & Deser, C. (2020). Anthropogenic Aerosols Dominate Forced Multidecadal Sahel Precipitation Change through Distinct Atmospheric and Oceanic Drivers, Journal of Climate, 33(23), 10187-10204. Retrieved Nov 4, 2022, from https://journals.ametsoc.org/view/journals/clim/33/23/jcliD190829.xml

**Additional comments**

Lines 33-35 are about future changes in heavy precipitation trends, but the following part of the paragraph (lines 35-39) is about the total wet day rainfall and rx5day. Thereis therefore no rationale for the "for instance" of line 36. Lines 38-39: What "could be" mean here in terms of confidence?

Lines 33-35 are about the past changes in the heavy precipitation over West Africa during the last decades (and not future changes as you stated). The following sentence is therefore acceptable. Moreover, I believe that a proper detection and attribution study is needed for a strong statement that attributes a flooding event as a consequence of an extreme rainfall event. It could also result from compounding events. As this has not been done in their article, we believe that Dike et al, 2020 prefer to use "could be". This section has been strongly modified in the new version of the manuscript.

Lines 40-47: What the authors are trying to demonstrate is not clear. Is there a spatial inhomogeneity in changes in rainfall indices, or is it about the complexity of changes in precipitation characteristics, that will be rainfall indices-dependent? Please rephrase thetext to show the main point of the paragraph more clearly.

In the revised manuscript, this section has been rewritten, with a focus on a few common extreme indices in the literature. The complexity of this paragraph is in part due to different indices computed from different data sources, different periods, and different seasons. That did not help to recap easily the main information.  We mentioned this point.

Line 48: Is this information obtained from observations?

This information is removed from the revised manuscript.

Lines 61-69: Do the authors note a relationship between bias in the different rainfall indices and bias in seasonal mean precipitation?

We reduced the number of indices an we did not evaluate this link.

Line 71 and Line 72: "anthropogenic emission of greenhouse gases", and "the shared socioeconomic pathway scenarios". Please name the scenarios

Line 71: This is a forward statement, and we believe that it is no need to go too much into detail (as for the different seasons in my introduction, different indices, and so on). Rind et al, 1989;  Mearns et al. 1995 used a doubling CO2 simulation, Hegerl et al. 2015 and Diedhiou et al. 2018 discussed RCP2.6,RCP4.5,RCP6.0,RCP8.5 projections, Akinsanola et al. 2020 evaluated the RCP4.5 and RCP4.8 scenarios. Van der Wiel et al, 2021 perform specific simulations (present-day, 2°C, 3°C, and outputs from CMIP5 RCP8.5 GCMs simulations). We stop the inventory here, as the aim is not to check in detail the differences in the simulations. However, for specific results that we mentioned, we provide information about the scenarios.

We reduce the revised introduction, and modify this point.

Line 74: The sentence is about RX1day and RX5day while the previous sentences are about extreme events. Please be clearer.

The previous sentence that you mention talks about "Extreme rainfall events" in general. These events can be characterized in terms of frequency, duration intensity. The index RX1day describes the maximum rainfall quantity in 1 day (so an intensity). The RX5day describes the maximum of 5 consecutive days of rainfall, which is a measure of duration and intensity. RX1day in removed from the new version of the manuscript.

Line 77: "RCM-CMIP5" Please define and explain.

We rewrote this expression in the revised manuscript: "Regional climate models (RCM)" forced with outputs from CMIP5 models". Thank you for the comment.

Lines 75-79: Are the results also model-dependent?

I think that there are noticeable differences between GCMs projections and those from RCMs, in terms of rainfall characteristics. Moreover, within the RCMs simulations, there is still a spread in the magnitude of the changes, and even in the sign of the change depending on the variable of interest. This can be seen in Figure 8 of Akinsanola and Zhou (2019).

Line 80: Does "These simulations" refer to Akinsanola and Zhou (2019)? Please be more specific.

Yes, we clarify this point by referring to Akinsanola and Zhou (2019) at the beginning of the paragraph. Thank you for the remark.

Line 89: both enhanced.

Thank you for the remark, this section has been restructured in the revised manuscript.

Lines 94-96: The increase in air moisture following Clausius Clapeyron explains a partof the seasonal mean increase in water vapor. I am puzzled about how the increase in water vapor, following Clausius Clapeyron could lead to a change in precipitation variability. Do the authors mean that it would be due to a change in variability of the temperature (SSTs) that would lead to different changes in air moisture?

This is an interesting point. Unfortunately, the authors did make such a statement and computed a synthetic change in the rainfall variability du the enhanced water vapor in the future. As the obtained change in variability exceeds the simulated change in the rainfall variability, and as they expect a decrease in the contribution of the dynamic, they conclude that the increase in the variability is due to thermodynamic changes.   We won't add anything to the statement in Lines 94-96. We leave the discussion to the community.

Line 103: Please replace "More" with ", where"

The corrections are applied. Thank you

Line 107: "warming and cooling", is that following a north/south dipole? Please be more specific.

No, we were not talking about a dipole. Rather, we were referring to the different positive and negative phases of the AEM, which are characterized by the above positive and negative SST anomalies in the eastern equatorial Atlantic, respectively.

Lines 113-114 could be shortened, removing "The first mode…indicates a strong", and removing ", and", in line 115.

We keep the sentence. Thank you

Line 115: Is it the "total variability" in Guinea coast rainfall? What is the time scale considered for the variability? (*e.g.*, daily, interannual?).

This is not the variability over the Guinea Coast, but the covariability between West Africa and the tropical Atlantic. This covariability between the Guinea Coast and the eastern equatorial Atlantic is dominant on the interannual timescale.

Line 116: Is it about the wind convergence?

Yes, this is about a low-level wind convergence. We added "wind" to the revised text.

Line 124: "variability of the AEM". Is it the daily variability? Please be more specific throughout the text.

It is a variability on the interannual timescale. This information is added in the revised manuscript. Thank you for the remark.

Lines 205-215: The authors could add sentences to explain briefly why the authors are using these metrics.

There are numerous performance metrics. We just choose few common used metrics in the literature, and we specify it in the specified section.

Lines 225-230: Are the changes in rainfall indices following the changes in seasonal mean precipitation (sign and pattern)?

There are some similarities between some indices and this help to reduce the number of indices in our revised manuscript.

Line 225: "simulated by climate models". The authors are not showing the results for each model individually in Figure 1. Please change it to "shown by the CMIP6 ensemblemean".

Good remark. This section is modified. Thank you.

Line 230: "observations". Shown where?

Figure A1.g is no longer in the revised manuscript.

Lines 236-246: Are these results model-dependent?

Yes, they are model and observation dependent. This has been highlighted throughout the revised manuscript.

Figure 1: It would be helpful to have the observation with contours.

In the revised manuscript, we showed in color the multimodel ensemble median, and, in contours, the ensemble median of the 6 rainfall observation datasets. Thank you for the suggestion.

Lines 261: There are plenty of references on the change in seasonal mean precipitation over West Africa and over the Sahel. Please acknowledge the literature.

We acknowledged the literature as mentioned. Thank you.

Lines 272-273: Is the change in RX1day consistent with the shortening of the rainy

season over the western Sahel?

Change in the frequency of wet days is more consistent with the shortening of the rainy season over the western Sahel (Fig. 3l of the old manuscript). The RX1day just tells you about the maximum intensity of a daily rainfall event, which is increasing over all the regions of West Africa (Fig. 3h of the old manuscript).

Figure 3: The pattern of R10mm is very similar to the pattern of PRCPTOT (Figure 3). The authors could comment on the possible strong role of R10mm in the total change in precipitation. Does R10mm provide a similar result for the number of rainy days?

The best way to make this link is to evaluate the contribution of the R10mm events (defined as days when more the 10mm of rainfall occurs) to the total rainfall. We do not compute this index, and we won't do it, as we have to reduce the number of indices in our article. This choice has been motivated by the two anonymous reviewers. Still, this would be a nice approach.

Line 285: What is the timescale used for computing the standard deviation here?

The standard deviation is simply computed over the 20 years period, without removing any trend or any frequency from the raw indices. The methodology is improved in the revised manuscript.

Lines 311-312: As for the precipitation indices, result sensitive to how the forced response was removed? (*e.g.*, a linear trend here)

We didn't test different ways to remove the trend, but in our previous work (Worou et al.2022), we test the impact of a linear trend and a quadratic trend on the AEM variability. Similar results were obtained in both cases. There are other methods, but we make things simple in our current study.

Line 314: "total wet-day precipitation index" does not show significant differences over land in Figure 5.

This index is removed from the revised manuscript.

Line 364: "weakened variability". What is the considered time scale?

We are considering the interannual timescale. This information is added to the revised manuscript.

Line 375-377: Are differences between periods significant? How would this be consistent with Figure 5 which shows no robust effects of the tropical Equatorial modeon precipitation extreme over land?

Spatial distribution of differences between periods are hardly significant. We mentioned it in the revised manuscript.

---

## Referee Report (RR1)

2nd review for "Future changes in the mean and variability of extreme rainfall indices over the Guinea Coast and role of the Atlantic equatorial mode" by Worou et al.

**Overview**

I commend the authors on a vastly improved iteration of their manuscript. From my side, the authors have satisfactorily answered and addressed all my concerns, the paper is close to a publishable state in my view. There are only a few very minor points left.

**General comments/questions**

- Even though the paper focuses on extreme rainfall, I think the pronounced mismatch between models and observations in CWD and FRQW (Fig. 1 e+f) deserves a bit more discussion. Clearly, there is a bimodal imprint in the observational data from the Guinea Coast region, which the models are just unable to resolve for present day scenarios. And this may or may not be of relevance for the characteristics of future daily rainfall, perhaps rather more so for multi-day precipitation extremes.

**Specific comments/questions**

L278:    "…are two orders of magnitude greater…". If this is related to 100%, then it is rather "…twice as high…".
Figure 3:    This is model minus observation, correct?
Figure 5:    This is long-term minus present day, correct? What do the blank areas without stippling/hatching/etc. indicate?
L452:    Typo for "averaged".
L475:    Do the authors mean the "fraction of explained variance"?

---

## Author Response (AR2)

**Response to the reviewer**

We would like to thank the reviewer for his meaningful comments.
Please, find below, in blue, our point-by-point responses to the questions raised by the reviewer (in black color).

**2nd review for "Future changes in the mean and variability of extreme rainfall indices over the Guinea Coast and role of the Atlantic equatorial mode" by Worou et al.**

**Overview**

I commend the authors on a vastly improved iteration of their manuscript. From my side, the authors have satisfactorily answered and addressed all my concerns, the paper is close to a publishable state in my view. There are only a few very minor points left.

**General comments/questions**

- Even though the paper focuses on extreme rainfall, I think the pronounced mismatch between models and observations in CWD and FRQW (Fig. 1 e+f) deserves a bit more discussion. Clearly, there is a bimodal imprint in the observational data from the Guinea Coast region, which the models are just unable to resolve for present day scenarios. And this may or may not be of relevance for the characteristics of future daily rainfall, perhaps rather more so for multi-day precipitation extremes.

  Figure R2.1 indicates that there are some models which present a bi-modal structure for the FRQW under present-day conditions (e.g. GFDL-ESM4, IPSL-CM6A-LR, MIROC6, NorESM2-LM and NorESM2-MM). These models keep the bimodality structure for the long-term future period. On the other hand, CanESM5 shows no bimodal structure in the FRQW under the present-day climate situation, while in the long-term future, the bimodality structure appears, due to a substantial decrease in the projected boreal summer FRQW values. The other GCMs conserve their annual cycle structures in both climates.

  Overall, there is no clear relation between the biases in the FRQW index and the projected long-term changes in FRQW (Fig. R2.2).

  The reviewer wonders if the biases in the FRQW could be relevant to the projected changes in the multi-day precipitation extremes. Considering the maximum rainfall over five consecutive days (Rx5day), we found no clear relationship between the biases in FRQW and the projected changes in Rx5day over Guinea Coast, during the boreal summer (Fig. R2.3).

  A similar result is obtained when we consider the link between the biases in the CWD index and the projected changes in the CWD and Rx5day indices (Fig. R2.4).

  We added the following sentence in the revised manuscript: "Noteworthy, we found no clear linkages between the JAS CWD and FRQW biases and the JAS long-term projected mean changes in the extreme rainfall indices (not shown)."

[Figure]

*Figure R2. 1 Annual cycle of the FRQW index averaged over the Guinea Coast, for the 1995-2014 (blue curves) and 2080-2099 (orange curves) periods, and for the 24 GCMs. Observations ensemble median is also indicated in dashed lines for the 1995-2014 period.*

[Figure]

*Figure R2. 2 Link between the FRQW biases (model minus the observations EnsMedian) and the FRQW long-term changes (2080-2099 minus 1995-2014) over Guinea Coast during July-September. Biases and changes are computed at a monthly level before being averaged over July-September.*

[Figure]

*Figure R2. 3 Link between the FRQW biases (model minus the observations EnsMedian) and the long-term changes (2080-2099 minus 1995-2014) in the RX5day over the Guinea Coast. Biases and changes are computed at a monthly level and averaged over July-September.*

[Figure]

*Figure R2. 4 Link between the CWD biases (model minus the observations EnsMedian) and the long-term changes (2080-2099 minus 1995-2014) in the RX5day over the Guinea Coast. Biases and changes are computed at a monthly level and averaged over July-September.*

**Specific comments/questions**

- L278: "…are two orders of magnitude greater…". If this is related to 100%, then it is rather "…twice as high…".
  Thank you for your suggestion. We take it into account.
  "This means, in other words, that the CWD values in the GCMs are twice as high as in the observations."

- Figure 3: This is model minus observation, correct?
  Yes, it is. We add this information in the captions of Fig. 3 and Fig. 8 of the revised manuscript.

- Figure 5: This is long-term minus present day, correct? What do the blank areas without stippling/hatching/etc. indicate?
  Yes, it is correct. We specified at the end of the sentence the difference performed (2080-2099 minus 1995-2014).

  Blank areas without stippling/hatching/and crossed lines are due to a non-interpolation between the three robustness categories. Their visibility is strengthened by the coarse grid resolution (2.8°x2.8°). We add this sentence to the figure in the revised article:

  Figure R2.5 includes blue dots in all the grid points over West Africa. These blue dots can help to visualize the delimitation between each robustness category.

[Figure]

*Figure R2. 5 Projected multi-model ensemble long-term median change in the JAS rainfall extreme indices over West Africa, relative to the present-day period (2080-2099 minus 1995-2014). The stippling indicates regions where the change robustly emerges from internal variability (at least 66% of the models show a change greater than the IAV and at least 80% of the models agree on the sign of change); hatching (\\) indicates regions where the change is non-robust (fewer than 66% of the models show change greater than the IAV); crossed lines (X) indicate conflicting signals where at least 66% of the models show change greater than the IAV, with less than 80% agreement on the sign of the change.* **Blue dots indicate grid points***.*

- L452: Typo for "averaged".
  Thank you for the correction, we take it into account.
- L475: Do the authors mean the "fraction of explained variance"?
  Yes, we replace the variance explained fraction (VEF) with the fraction of explained variance (FEV)